## ARTICLES

# Light-Seq: light-directed in situ barcoding of biomolecules in fixed cells and tissues for spatially indexed sequencing

Jocelyn Y. Kishi [1,2,7], Ninning Liu[1,7], Emma R. West[3,4,7], Kuanwei Sheng [1], Jack J. Jordanides[1], Matthew Serrata[1], Constance L. Cepko [3,4,5], Sinem K. Saka [1,2,6] and Peng Yin [1,2]

We present Light-Seq, an approach for multiplexed spatial indexing of intact biological samples using light-directed DNA barcoding in fixed cells and tissues followed by ex situ sequencing. Light-Seq combines spatially targeted, rapid photocrosslinking of DNA barcodes onto complementary DNAs in situ with a one-step DNA stitching reaction to create pooled, spatially indexed sequencing libraries. This light-directed barcoding enables in situ selection of multiple cell populations in intact fixed tissue samples for full-transcriptome sequencing based on location, morphology or protein stains, without cellular dissociation. Applying Light-Seq to mouse retinal sections, we recovered thousands of differentially enriched transcripts from three cellular layers and discovered biomarkers for a very rare neuronal subtype, dopaminergic amacrine cells, from only four to eight individual cells per section. Light-Seq provides an accessible workflow to combine in situ imaging and protein staining with next generation sequencing of the same cells, leaving the sample intact for further analysis post-sequencing.

To comprehensively understand the cellular states that drive biological function and pathology, it would be ideal to combine optical characterization and imaging-based screening of fixed cells and tissues with omics-level profiling by next generation sequencing (NGS).

Recent applications that focus on integrating imaging with transcriptomic measurements of the same individual cells rely largely on physical dissociation and sorting of cells using opto- or microfluidics before, after or during imaging[1–4]. Alternative methods rely on photoconversion of fluorescent proteins in selected live cells for later sorting[5–7], or photoactivation of messenger RNA capture moieties followed by physical aspiration of target live cells for transcriptomic analysis[8]. These approaches typically require live imaging of cells and extraction of a limited number of visual features before pooling or single-cell sorting of dissociated cells for sequencing. In these cases, the spatial information or the original location of the cell is typically lost or is not relevant.

In an effort to preserve spatial information and morphology, particularly in tissues, recent spatial transcriptomic methods aim to profile cells in situ[9,10]. Employing various modes of DNA barcoding, spatial omics approaches aim to spatially index either a two-dimensional capture surface (Slide-Seq[11,12], HDST[13], Seq-Scope[14], Stereo-Seq[15], Sci-Space[16]) or the biomolecules directly. In the latter case, barcode sequences can be incorporated onto biomolecules through in situ hybridization (ISH) of combinatorially barcoded probes, or enzymatically by reverse transcription (RT) or ligation (for example, after microfluidic delivery of the barcodes into defined sample positions as in DBIT-Seq[17]). These barcodes can then be read out in situ by iterative imaging (for example, SeqFISH[18] or MERFISH[19], HybISS[20]), by in situ sequencing (FISSEQ[21], CARTANA[22]) or by ex situ sequencing

after they are retrieved from the sample (DBIT-Seq[17], IGS[23]). Alternative approaches have used spatially restricted (1) iterative photo-cleavage and collection of ISH probe barcodes from target regions of interest (ROIs) (DSP[24]) or (2) collection of target cells themselves by selective immobilization[25], suction[26] or laser capture microdissection[27] for subsequent barcoding and sequencing.

Existing spatial profiling methods currently rely on one or several of expensive instrumentation (closed box systems costing $100,000–1,000,000, high-end custom microscopes or fluorescence sorters), complex multi-round optical deconvolution of barcoded arrays or barcode sequences in situ and custom microfluidics systems or arrays with rigid sample format restrictions. Many of these methods also use targeted ISH probes rather than whole transcriptome sequencing or are partially or completely destructive to the sample. This creates a high need for accessible and scalable visual selection methods that can directly link multi-dimensional and high-resolution cellular phenotypes (including morphology, protein markers, spatial organization) to transcriptomic profiles for diverse sample types. Recent methods in this direction have adopted an ultraviolet (UV)-uncaging approach to allow spatial barcoding of RNAs (PIC[28]) or whole cells (via DNA-barcoded antibodies or lipids as in ZipSeq[29]), but the connection to the sequencing output requires destruction of the samples.

Here, we present a different paradigm, named Light-Seq, for light-directed in situ spatial barcoding of target molecules in desired ROIs for ex situ NGS without sample destruction (Fig. 1). We achieve this by two innovations: (1) Building on our previous work of light-controlled rapid crosslinking of nucleotides[30], we utilize an ultrafast crosslinking chemistry[31] and parallelized photolithography[32] for light-controlled enzyme-free covalent attachment

[1]Wyss Institute for Biologically Inspired Engineering, Harvard University, Boston, MA, USA. [2]Department of Systems Biology, Harvard Medical School, Boston, MA, USA. [3]Department of Genetics, Blavatnik Institute, Harvard Medical School, Boston, MA, USA. [4]Howard Hughes Medical Institute, Chevy Chase, MD, USA. [5]Department of Ophthalmology, Harvard Medical School, Boston, MA, USA. [6]Present address: Genome Biology Unit, European Molecular Biology Laboratory (EMBL), Heidelberg, Germany. [7]These authors contributed equally: Jocelyn Y. Kishi, Ninning Liu, Emma R. West. ✉e-mail: jocelyn.y.kishi@gmail.com; cepko@genetics.med.harvard.edu; sinem.saka@embl.de; py@hms.harvard.edu

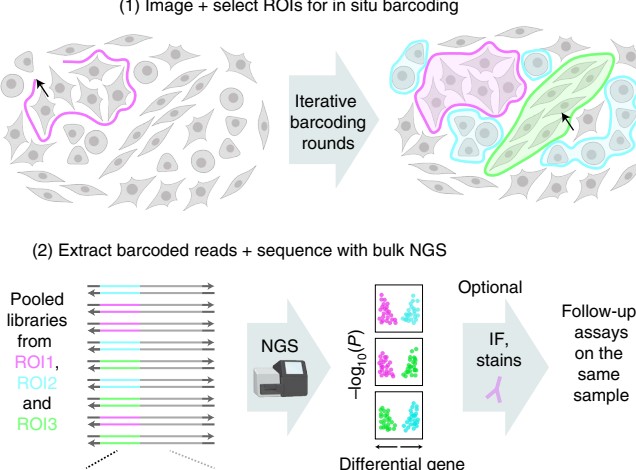

(1) Image + select ROIs for in situ barcoding

Iterative barcoding rounds

(2) Extract barcoded reads + sequence with bulk NGS

Pooled libraries from ROI1, ROI2 and ROI3

NGS

$-\log_{10}(P)$

Optional IF, stains

Follow-up assays on the same sample

Region ID

Sequence (for example, cDNA)

Differential gene expression

**Fig. 1 | Light-Seq overview.** Light-Seq enables selective barcoding of custom selected cells or tissue regions in situ for transcriptomic sequencing. Step (1): Target ROIs can be selected based on phenotypic factors including spatial location, morphology or protein biomarkers in automated or manual fashion after imaging. Custom selection allows large or small regions, and contiguous or disjointed cell groups to be flexibly labeled by photocrosslinking of DNA barcodes, which are then converted into sequenceable indices. For multiplexed targeting of different cell groups or regions, the process can be iterated using different barcode sets. Step (2): After light-directed labeling, barcoded cDNAs are released and prepared into pooled sequencing libraries which are read by standard NGS platforms. The obtained profiles can be analyzed to identify differentially expressed genes. Optionally, the same sample can be revisited after sequencing to perform follow-up assays, such as high-resolution imaging, morphology or protein labeling.

of pre-designed barcode sequences onto biomolecules in situ. (2) Developing a cross-junction synthesis reaction to integrate DNA barcodes onto biomolecules for ex situ sequencing. We combine this barcoding strategy with a nondestructive workflow to enable imaging and whole transcriptome sequencing of selected cells in fixed samples with the possibility to revisit the sample for further assays such as protein stainings. We benchmark and demonstrate the applicability of Light-Seq on mixed cell cultures and mouse retina tissue sections, and utilize the approach for rare cell transcriptomics, where we identify biomarkers for the very rare dopaminergic amacrine cells (DACs) in the mouse retina.

## Results

**Light-Seq overview and barcoding chemistry.** Light-Seq employs a light-controlled DNA barcode attachment strategy to enable custom indexing of ROIs in imaged samples. To achieve this capability, we use barcode strands that contain the ultrafast photocrosslinker 3-cyanovinylcarbazole nucleoside (CNVK)[31]. Hybridized CNVK can form an interstrand crosslink upon short UV illumination (Fig. 2a). Our general strategy is to hybridize CNVK-containing barcodes to complementary docking sequences and then direct UV light to an ROI to photocrosslink the barcode strands only in that area, and then wash away noncrosslinked barcode strands. This process can be iteratively performed to label multiple ROIs with orthogonal barcode strands.

To set up the Light-Seq platform, we optimized the crosslinking exposure time and light intensity and found that 1–10 s produced efficient crosslinking, similar to previous in vitro results[33]. To create custom photomasks in a parallelized manner, we use a

digital micromirror device (DMD)[34] attached to a standard widefield imaging setup. Using a ×10 objective, a single mirror in our DMD setup can yield a practical resolution <2 μm based on estimating the full-width at half-maximum on a dot array (Extended Data Fig. 1a,b). We first validated the barcoding chemistry in vitro on a glass surface coated with immobilized DNA strands. By adding fluorescently labeled barcode strands and using custom photomasks (for example, of a cat) to crosslink them to the surface, we were able to create patterns with a single barcode strand (Fig. 2b) or use sequential rounds of barcoding with unique strands to pattern multiple regions on the same slide, such as the three-color Penrose triangle (Fig. 2c). Although we primarily use 365-nm light-emitting diode (LED) epi-illumination with a DMD for the UV crosslinking step, we also demonstrated subcellular spatial labeling with a 405-nm laser[30] on a confocal point-scanning microscope (Extended Data Fig. 1c–g). The laser-scanning system offers higher resolution and contrast but is slower than the DMD and LED illumination.

Next, to read out the sequence of target DNAs with their corresponding crosslinked barcode sequences by NGS, the crosslinked bases must be addressed without loss of either the barcode identity or the barcoded sequence. To this end, we developed a cross-junction synthesis reaction to copy both the barcoded DNA sequence and barcode into a new single strand of DNA without a crosslink (Fig. 3a). For this we use a strategy similar to our previously developed Primer Exchange Reaction (PER)[35]. We use a primer with a strand displacing polymerase that copies a new strand until it is halted at the crosslink point. We designed the sequences around the crosslink to have an identical domain so that the extended primer can reach across the junction and be templated on the opposing strand through a branch migration[36–38] competition between two identical domains. The single-stranded DNA product of this cross-junction reaction can then be amplified and read out with standard NGS pipelines.

**Utilizing Light-Seq for whole transcriptome sequencing.** To use Light-Seq for whole transcriptome sequencing, we first perform in situ RT[39] on fixed and permeabilized cells or tissue sections to synthesize complementary DNA (cDNA) (Fig. 3b, step 1). To label RNAs regardless of polyadenylation, we use a degenerate primer with five N and three G bases on the 3′ end[40] and a barcode docking site on the 5′ end. After RT, the 3′ ends of the generated cDNAs are A-tailed to create a 3′ handle for ex situ primer binding. A CNVK- and unique molecular identifier (UMI)-containing barcode strand is then hybridized to the 5′ docking site on all cDNAs. We then direct UV light to the ROI to photocrosslink the barcode strands in that area and then wash away noncrosslinked barcode strands. This process can be iteratively performed to label multiple ROIs with orthogonal barcode strands (Fig. 3b, step 2).

After all ROIs have been barcoded, barcoded cDNAs are collected from the sample with a mild RNase H treatment and prepared for sequencing (Fig. 3b, step 3). We then apply our cross-junction synthesis reaction to stitch the cDNAs and barcode sequences together into a single readout strand with one enzymatic reaction step (Fig. 3b, step 4). This direct attachment of spatial barcodes onto transcriptomic sequences allows a straightforward transition to ex situ sequencing of the readout sequences after PCR amplification and NGS library preparation (Fig. 3b, steps 5–6). While developing the protocol, we made several critical design choices to minimize the potential artifacts of the in situ RT reaction that have been previously observed[41–43], as we discuss in detail in Supplementary Note 1.

To validate the capability to select and barcode cells based on phenotypic profiles, we performed a cell mixing experiment where mouse 3T3 cells were mixed with human HEK cells that stably express eGFP (Fig. 3c). We targeted each cell type with distinct barcode sequences (coupled to different fluorophores for visualization purposes) by manually selecting them based on eGFP expression

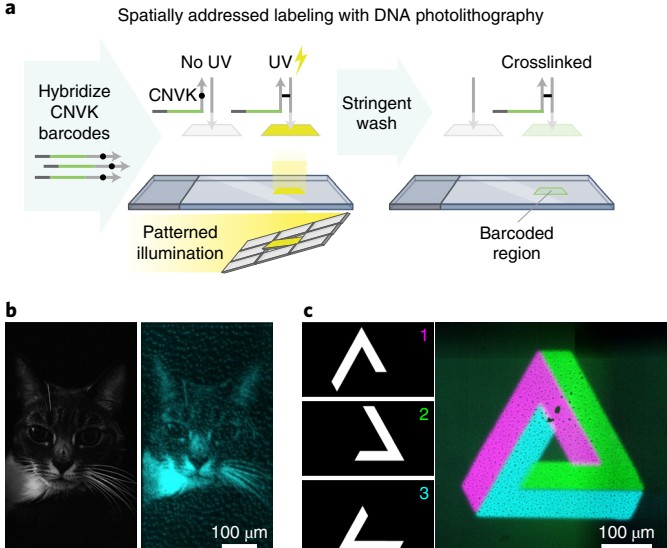

**Fig. 2 | Light-controlled DNA photocrosslinking. a**, Schematic for light-directed barcode attachment on glass slides. Biotinylated single-stranded DNA oligos are immobilized onto glass surfaces with biotin–streptavidin binding. Fluorescent barcode strands containing a CNVK moiety in the complementary domain are hybridized to these immobilized oligos. Target pixels corresponding to ROIs in the field of view are UV-illuminated in a parallelized fashion using a DMD to photocrosslink the barcodes in a photomask pattern. Uncrosslinked strands are removed by stringent washes, which reveals the encoded barcode pattern in fluorescence. **b**, Custom patterning (right) achieved by using a cat photo (left) to create a binary photomask and photocrosslinking the fluorescent CNVK-containing barcode strands onto a functionalized glass slide. **c**, Iterative photocrosslinking using three photomasks (left) that define three ROIs to attach three orthogonal barcode strands onto a DNA-coated glass slide, forming a Penrose triangle (right).

and cellular morphology (Supplementary Figs. 1 and 2). Out of a total of ~4,500 cells in the same well, we barcoded ~25 human and ~25 mouse cells and confirmed the targeted cell type was barcoded correctly by a fluorescent scan (Fig. 3d–f). Sequencing reads were mapped to a merged human and mouse genome, and unique maps were further analyzed. We were able to validate the barcode integration by cross-junction synthesis by matching the reads to our expected sequence output (Supplementary Fig. 3a–c). We observed a good discrimination ratio of mouse and human maps to their respective barcode sequences ($89.1 \pm 0.7\%$ of mouse reads, $87.3 \pm 0.7\%$ of human reads, $n = 3$ technical replicates, mean $\pm$ s.d.), and notably $93 \pm 0.5\%$ of eGFP reads were correctly attributed to the human-specific barcodes (Fig. 3g). After sequence extraction, we performed multiplexed immunofluorescence (IF) to validate the integrity of the sample for secondary assays (Fig. 3h).

Next, we estimated the abundance of transcripts that Light-Seq can capture. Because the barcoding area of Light-Seq can be arbitrarily set by the user, we chose to normalize the number of transcripts that can be captured with Light-Seq as the number of UMI sequences per 'unit area' that was roughly the size of a bead in Slide-Seq[11] or a barcoded square in DBIT-Seq[17], which we define as $10 \times 10\,\mu m^2$. We obtained the number of UMIs per unit area by estimating the total cell or tissue area that was subject to UV illumination for each barcode based on microscopy images of ROIs via segmentation (Extended Data Fig. 2a) and calculating the average UMI count (for correctly barcoded and uniquely mapped reads) for a 100-$\mu m^2$ barcoded area. At a sequencing depth of ~30 million reads per replicate and with only ~1/2 of the sample amplified for library preparation and sequencing, we observed an average of $1,959 \pm 453$ and $1,170 \pm 207$ UMIs per unit area for HEK and 3T3 cells, respectively (Supplementary Table 1, mean $\pm$ s.d.). We note that the read depth of ~30 million was subsaturating (Extended Data Fig. 2b), and a single ~200 million read dataset from half of the sample yielded 3,328 and 2,029 UMIs per unit area for HEK and 3T3 cells, respectively (Supplementary Table 1).

Normalized gene expression levels (expressed as log$_2$-transformed transcripts per kilobase per million reads (TPM)) displayed good correlation across technical replicates (Pearson correlation coefficient > 0.8). The top 200 expressed genes correlated highly across technical replicates (Pearson correlation coefficient > 0.9; Extended Data Fig. 2c,d) and comprised various protein coding genes, as well as short transfer RNAs and long noncoding RNAs (Supplementary Table 2 and source data), illustrating the range of transcripts that can be targeted with the barcoding strategy.

With Light-Seq, we were able to successfully recover species-specific transcriptomes of targeted cells, despite only selecting 1–2% of cells within the whole well. We hypothesized that the Light-Seq background, reflected here by the small portion of species-specific transcripts harboring the barcode corresponding to the other species (Fig. 3g), could arise from three potential sources: (1) diffusion of RNAs or cDNAs before photocrosslinking, (2) incomplete removal of uncrosslinked barcode strands during stringent washing and (3) light-scattering inducing out-of-ROI barcoding. To mitigate (1) and (2), we added blocking and crowding agents to our barcode hybridization step for all of the following experiments, which substantially improved signal-to-noise (Methods and Supplementary Note 2). To mitigate light-scattering effects, we seeded the cells below confluence. For higher density labeling (for example, in tissues), we suggest optimization of the optical setup and sample preparation and slight erosion of ROI boundaries (~1–3 $\mu m$) to account for scatter (Methods and Extended Data Fig. 1).

**Spatial sequencing with Light-Seq in tissue sections.** RNA sequencing of specific cell populations within tissue samples remains challenging, especially when target cells are rare or difficult to isolate. We therefore tested Light-Seq on fixed sections from the mouse retina to capture biomarkers from cell populations of interest based on in situ identification. First, in situ RT was performed in fixed 18 $\mu m$ retinal cryosections to synthesize cDNAs for spatial barcoding. We then manually selected three cellular layers of the retina,

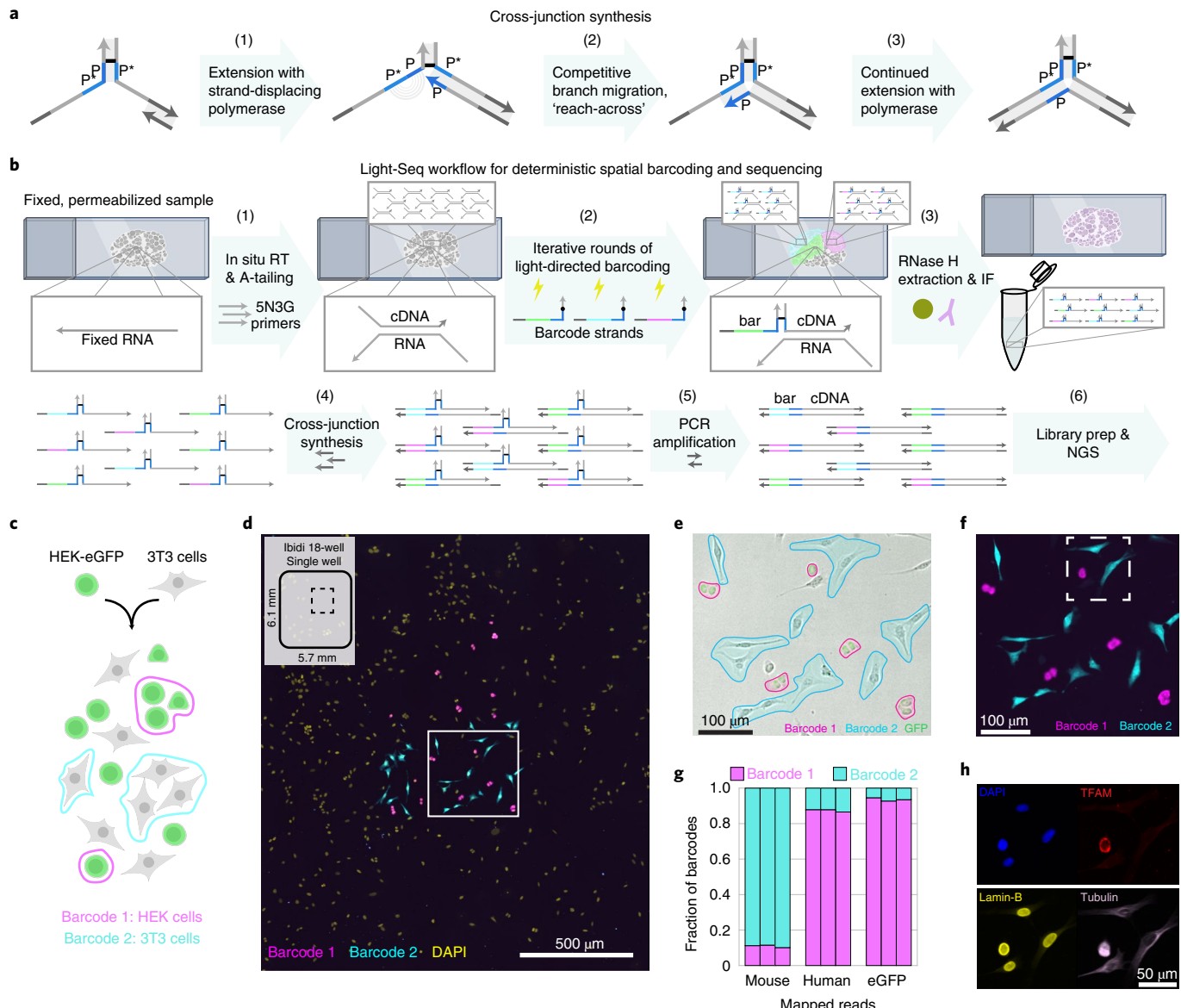

**Fig. 3 | Cross-junction synthesis and full in situ protocol with validation on cell mixtures. a**, Design of the cross-junction synthesis reaction. First, a primer extends the new strand until the stopper (step 1). Next, the extended primer P domain competes with the identical P domain on the opposite template through branch migration, similar to our previously developed Primer Exchange Reaction (PER)[35] (step 2). Once displaced, the synthesized P domain (blue) primer can bind across to form a three-way junction and then continue to be extended (step 3). The P domain is typically 7 nt, which may become 8 nt if the Bst polymerase A-tails. **b**, The Light-Seq workflow for in situ transcriptomic sequencing: (1) RT is performed with random primers containing a 5′ barcode dock site, followed by A-tailing of 3′ cDNA ends. (2) Within each ROI, a unique CNVK-modified DNA barcode strand is UV-crosslinked to the 5′ cDNA dock site. (3) Barcoded cDNAs are extracted using RNase H, which cleaves RNA in RNA–DNA hybrids. (4) The cross-junction synthesis reaction copies the barcode and cDNA sequences into a single strand for (5) PCR amplification and (6) sequencing library preparation and NGS. **c**, Cell mixing tests: eGFP-expressing HEK293 and mouse 3T3 cells were co-cultured and fixed, and cDNAs were labeled with Barcodes 1 and 2, respectively. **d**, A subset of ~25 3T3 (cyan) and ~25 HEK cells (magenta) were barcoded in the whole well of an 18-well chambered coverslip (rectangular area with the dashed line on the schematic marks the size of the stitched image shown in this panel with respect to the area of the whole well), each containing ~4,500 total cells (*n* = 3 technical replicates, representative image shown). All cells were stained with DAPI (yellow) after barcoding. **e**, Brightfield and GFP fluorescence overlaid with ROIs for labeling with Barcodes 1 and 2 (field-of-view is magnification of panel **d**, white square). **f**, Fluorescent image for panel **e** after photocrosslinking Barcodes 1 (magenta) and 2 (cyan). **g**, Portions of reads that mapped to human, mouse or eGFP sequences in a merged human and mouse reference genome, which were respectively labeled with Barcode 1 or 2 (*n* = 3 technical replicates). **h**, After barcoded sequence extraction, the same cells (white square from panel **f**) were stained by IF for Lamin-B (yellow), tubulin (violet) and TFAM (red, human epitope-specific).

each known to have unique function and cellular composition: the outer nuclear layer (ONL), containing rod and cone photoreceptors; the bipolar cell layer (BCL), containing bipolar cells, horizontal cells and Müller glia; and the ganglion cell layer (GCL), containing retinal ganglion cells and displaced amacrine cells (ACs) (Fig. 4a).

After photocrosslinking fluorescently labeled barcodes to all cDNAs within each layer, the targeted ROIs were imaged using a confocal microscope to verify layer-specific labeling (Fig. 4b and Extended Data Fig. 3a–c). Barcoded cDNAs were then extracted, pooled and prepared for NGS as described above, leaving the sample intact.

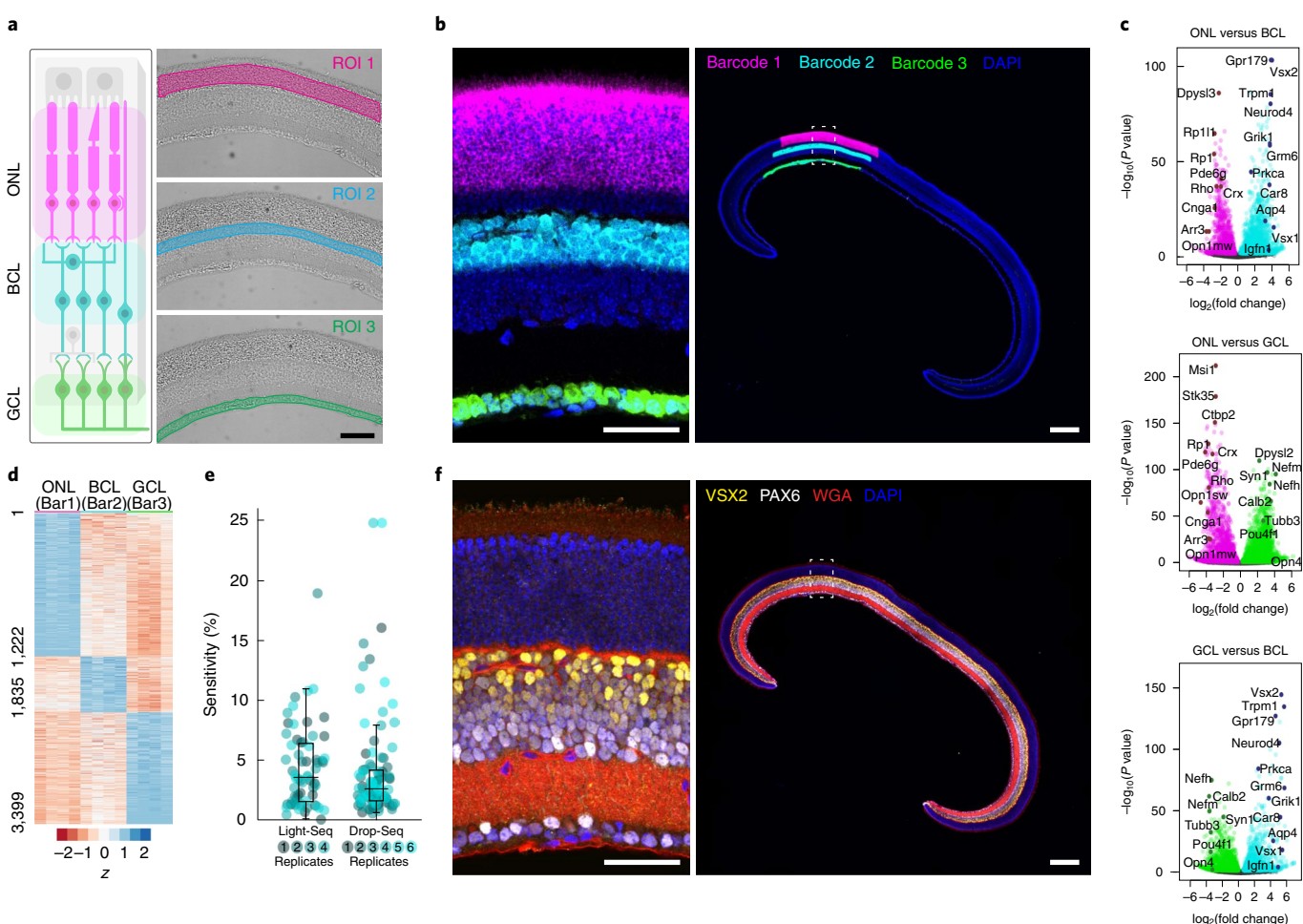

**Fig. 4 | Application of Light-Seq for spatial barcoding of three main retinal layers in fixed frozen mouse retina sections. a**, Three regions of the mouse retina were uniquely barcoded: the ONL with Barcode 1, the BCL with Barcode 2 and the GCL with Barcode 3. **b**, After barcoding, fluorescently labeled barcode strands were detected in the targeted cell layers: ONL (magenta, Barcode 1; 1,112 ± 199 cells, $n = 4$ sections), BCL (cyan, Barcode 2; 298 ± 29 cells, $n = 4$ sections) and GCL (green, Barcode 3; 91 ± 14 cells, $n = 4$ sections). **c**, Volcano plots of differentially expressed genes between the ONL and BCL (top), ONL and GCL (middle) and BCL and GCL (bottom), with select markers labeled. The $x$ and $y$ axes show the $\log_2$(fold change) and the $\log_{10}$($P$ value), respectively. **d**, Heatmap of $z$-scores for differentially expressed genes with enrichment in just one layer ($P_{adj} < 0.05$; see source data; two-sided Wald test with Benjamini–Hochberg adjustment for multiple hypothesis testing). **e**, Boxplot of estimated sensitivity of Light-Seq ($n = 4$ replicates, 16 genes) and Drop-Seq[46] ($n = 6$ replicates, 16 genes) compared with smFISH data for bipolar subtype marker genes with measured abundances based on quantitative smFISH[50]. Sensitivity is defined as (number of expected transcripts by smFISH)/(number of observed reads by Light-Seq or Drop-Seq). Midline marks the median and edges indicate the 25th and 75th percentiles. Whiskers extend to encompass all data not considered outliers (default threshold in MATLAB *boxplot* function; maximum whisker length is 1.5 × interquartile range). Dot color corresponds to replicate number. **f**, DAPI, WGA staining and IF for VSX2 and PAX6 proteins were performed on the same tissue section after extraction of barcoded cDNAs. Scale bars are 100 μm in **a**, and 50 μm (left) and 200 μm (right) in **b** and **f**.

Since the cellular composition of each retinal layer is well-established[44], we verified the specificity of spatial barcoding based on the levels of cell-type-specific markers associated with each barcode. We used DESeq2 (ref. [45]) to perform differential expression analysis on exon-mapped barcoded sequencing reads to look for layer-specific markers. Based on the size of barcoded areas, we labeled an estimated 1,112 ± 199, 298 ± 29 and 91 ± 14 (mean ± s.d., $n = 4$ technical replicates) cells for the ONL, BCL and GCL layers, respectively. UMI yields varied for different retinal layers depending on cell and RNA content of each layer, ranging from ~1,200 to 5,000 UMI's per 10 × 10 μm² unit area (Extended Data Fig. 3d and Supplementary Table 3). As expected, we observe that cells with larger cytoplasmic volumes correlate with higher UMIs per unit area (Extended Data Fig. 3d).

Technical replicates showed consistent read filtering throughout the sequence-processing pipeline (Extended Data Fig. 4a,b) and were well correlated based on principal component analysis for each layer (Extended Data Fig. 4c), and we discovered >3,400 genes with significant differential expression between pairs of barcoded populations (3,430 genes for ONL versus BCL; 3,434 for BCL versus GCL; 6,165 for ONL versus GCL, $P_{adj} < 0.05$; Fig. 4c,d and accompanying source data), including many known markers of rod and cone photoreceptors in the ONL, of bipolar cells and Müller glia in the BCL and of retinal ganglion cells in the GCL.

To further benchmark Light-Seq data, we simulated pseudo-bulk RNA sequencing of the retinal layers using published single-cell Drop-Seq data of ONL and BCL cells[46] (Methods). We saw strong correlation for the genes enriched between the ONL and BCL (98.6%

of genes significantly enriched in both assays were enriched in the same layer; Extended Data Fig. 4d). For all BCL-enriched genes, the ratios of Light-Seq to Drop-Seq reads show higher correlation and comparable sensitivity for longer genes and lower correlation and sensitivity for shorter genes (Extended Data Fig. 4e–g), consistent with Light-Seq's internal priming strategy versus Drop-Seq's polyA-targeted transcript capture.

Interestingly, more genes were significantly differential among the layers in Light-Seq data than Drop-Seq (3,430 compared with 1,524) despite similar total numbers of genes detected (24,460 compared with 24,904) across technical replicates, in line with previous studies that suggest targeted bulk transcriptome measurements can provide better statistics for discovery of moderate to lowly expressed biomarkers than single-cell sequencing[47].

To assess the sensitivity of Light-Seq for detecting mRNAs, we compared both Light-Seq and Drop-Seq directly with single molecule FISH (smFISH[48]) data. Our previous work used quantitative multiplexed SABER-FISH[49] to co-detect 16 mRNA markers of bipolar interneuron subtypes in age-matched mouse retinas[50]. From these data, we estimated the average number of transcripts per cell within the pooled bipolar cell population. Multiplying this by the number of cells within the barcoded Light-Seq BCL (ROI2) and pseudo-bulked Drop-Seq data, we estimated the expected number of detectable transcripts for each gene in the sequenced populations. Relative to smFISH, we find that the sensitivity of Light-Seq was $4.29 \pm 3.39\%$ (mean $\pm$ s.d., $n = 16$ genes, 4 replicates) and sensitivity of Drop-Seq was $3.97 \pm 4.38\%$ (mean $\pm$ s.d., $n = 16$ genes, 6 replicates) (Fig. 4e and Extended Data Fig. 4h–j).

After Light-Seq, the same sections were then stained with DAPI, wheat germ agglutinin (WGA) and antibodies targeting PAX6 and VSX2 proteins, demonstrating that cellular DNA, oligosaccharides on cell membranes and proteins remained detectable after extraction of barcoded cDNAs for sequencing (Fig. 4f and Extended Data Fig. 3c).

Due to internal priming during the RT step, Light-Seq shows read coverage spanning gene bodies (Extended Data Fig. 5a) and consistent reads per kilobase per million mapped reads (RPKM) across transcripts of different lengths (Extended Data Fig. 5b,c). This strategy, combined with membrane permeabilization in situ, enables Light-Seq to capture a wide variety of RNA species including nonpolyadenylated and mitochondrial transcripts (see source data for Fig. 4). When intronic sequences are included for transcriptome mapping, ~21–26% of reads map to introns, indicating capture of nuclear RNAs at ratios consistent with similar methods using internal RNA priming (Supplementary Note 3)[51], suggesting that our workflow can detect both nuclear and cytoplasmic RNAs.

**Rare cell transcriptomics with Light-Seq.** For applications where only a small number of cells of interest are present and/or where spatial context is critical for their identification, capturing

transcriptomes remains a major challenge. To test Light-Seq's utility for imaging, barcoding and sequencing of rare cells, we targeted the extremely rare and difficult to isolate DACs in the mouse retina. DACs comprise $\leq 0.01\%$ of retinal cells[52] and are interspersed among diverse types of neurons. Previous works aimed at profiling DACs with single-cell RNA-seq[53] or microarray capture after dissociation[54] have seen limited success. To find subtype-specific biomarkers of DACs, we fixed and sectioned mouse retinas and first performed in situ RT (Fig. 5a, step 1). Then, IF was done to detect tyrosine hydroxylase (TH), a known marker of DACs, to locate them for barcoding (step 2). Next, two rounds of barcoding were guided by the TH IF signal (Fig. 5b and Extended Data Fig. 6a), to barcode cDNAs within TH⁻ DACs with FITC-labeled barcode strands (roughly 500 cells per section) and cell bodies of all TH⁺ DACs with Cy3-labeled barcode strands (4–8 cells per section, $n = 5$ section replicates) (step 3). Finally, barcoded cDNAs were displaced from the tissue and prepared for sequencing, with the sample remaining intact for post-sequencing stains (step 4).

Light-directed barcoding permitted precise labeling of individual TH⁺ DACs within the dense tissue environment (for notes on optical system, see Methods and Supplementary Note 4). Although some light-scattering can induce out-of-ROI crosslinking, this effect was mitigated by drawing photomasks slightly smaller than the intended ROIs: for barcoding TH⁺ DACs, photomasks were drawn 1–3 μm inside the cellular boundary (Extended Data Fig. 6a–c). Use of a laser-based point-scanning microscope is slower but offers higher barcoding resolution and may be used in place of a DMD (Extended Data Fig. 1e).

Sequencing at a subsaturating sequencing depth of 3.5–5 million reads per replicate (one pooled MiSeq run for five successfully amplified section replicates; Extended Data Fig. 6d) yielded 6,000–10,000 UMIs per $10 \times 10$-μm² unit area for TH⁺ DACs, with an average of 7,800 UMIs per cell (Supplementary Table 4). We again observed good gene body coverage and consistent RPKM across replicates (Extended Data Fig. 7a–c). Differential expression analysis revealed 36 significantly enriched genes in the TH⁺ population ($P_{adj} < 0.05$, $\log_2$(fold enrichment) > 1), including the known markers TH ($P_{adj} = 1.32 \times 10^{-24}$) and CARTPT ($P_{adj} = 4.98 \times 10^{-65}$)[54] (Fig. 5c and accompanying source data). To validate the top biomarker (*Cartpt*) after sequencing, we revisited the stored samples and performed IF for CARTPT on the same cells, revealing specific labeling (Fig. 5d,e).

For further validation, RNA-FISH was performed in new retinal sections to detect the top differentially expressed genes ($P_{adj} < 0.05$ and $\log_2$(fold enrichment) > 3). In all cases, the RNA-FISH confirmed the Light-Seq results, showing marker enrichment in TH⁺ DACs relative to neighboring TH⁻ ACs (Fig. 5f). As controls, we detected *Gad1* mRNA, with known expression in both TH⁺ DACs and TH⁻ ACs, and *Vsx2* mRNA, which is not expressed in either

**Fig. 5 | Rare cell transcriptomics by Light-Seq. a**, Workflow for performing Light-Seq on the rare TH⁺ AC subtype, DACs: (1) Mouse retinas were fixed, frozen and cryosectioned. (2) After in situ RT, sections were stained with an antibody targeting the TH protein to label DACs (orange). (3) Barcoding of TH⁻ ACs with FITC-barcode strands (Bar1) and TH⁺ DACs with Cy3-barcode strands (Bar2) was performed in two rounds of light-directed barcoding, guided by the antibody stain. (4) After barcoding, cDNAs were displaced for sequencing, leaving the sample intact for further stains on the same cells. **b**, Representative image ($n = 5$ replicates) of one section replicate, stained with anti-TH antibody (orange) and DAPI (blue) before barcoding. For each replicate, only four to eight individual TH⁺ DACs were identified and their cell bodies were barcoded with Bar2 (magenta), together representing 0.01–0.02% of all cells in each section, and ~300 TH⁻ ACs were barcoded with Bar1 (green). Scale bars are 200 μm. **c**, Differential expression analysis revealed 36 transcripts enriched in DACs ($P_{adj} < 0.05$; two-sided Wald test with Benjamini–Hochberg adjustment for multiple hypothesis testing; genes with $\log_2$(fold change) > 1 are shown; see source data) for $n = 5$ technical replicates. *Marker genes selected for further validation ($\log_2$(fold change) > 3 and $P_{adj} < 0.05$). **d**, Fluorescently labeled barcodes (Bar1, Bar2) reveal the location of barcoded cDNAs, relative to the TH IF. Scale bars are 10 μm ($n = 5$ replicates, each with 4–8 TH⁺ cells per section). **e**, After cDNAs were displaced and sequenced, the same intact sections were stained for a membrane label (WGA) and a known marker of DACs via IF (CARTPT, cyan), in addition to the original TH IF and DAPI labels. **f**, Markers with $\log_2$(fold change) > 3 and $P_{adj} < 0.05$ were validated using TH IF and RNA-FISH in new samples. Nondifferential controls, *Gad1* and *Vsx2*, were also detected to demonstrate FISH labeling in TH⁻ ACs and other retinal cells. Top row shows overlay of RNA detection with TH IF, and bottom row shows single RNA-FISH channel. Scale bars are 10 μm. Representative images of $n = 3$–4 section replicates per marker.

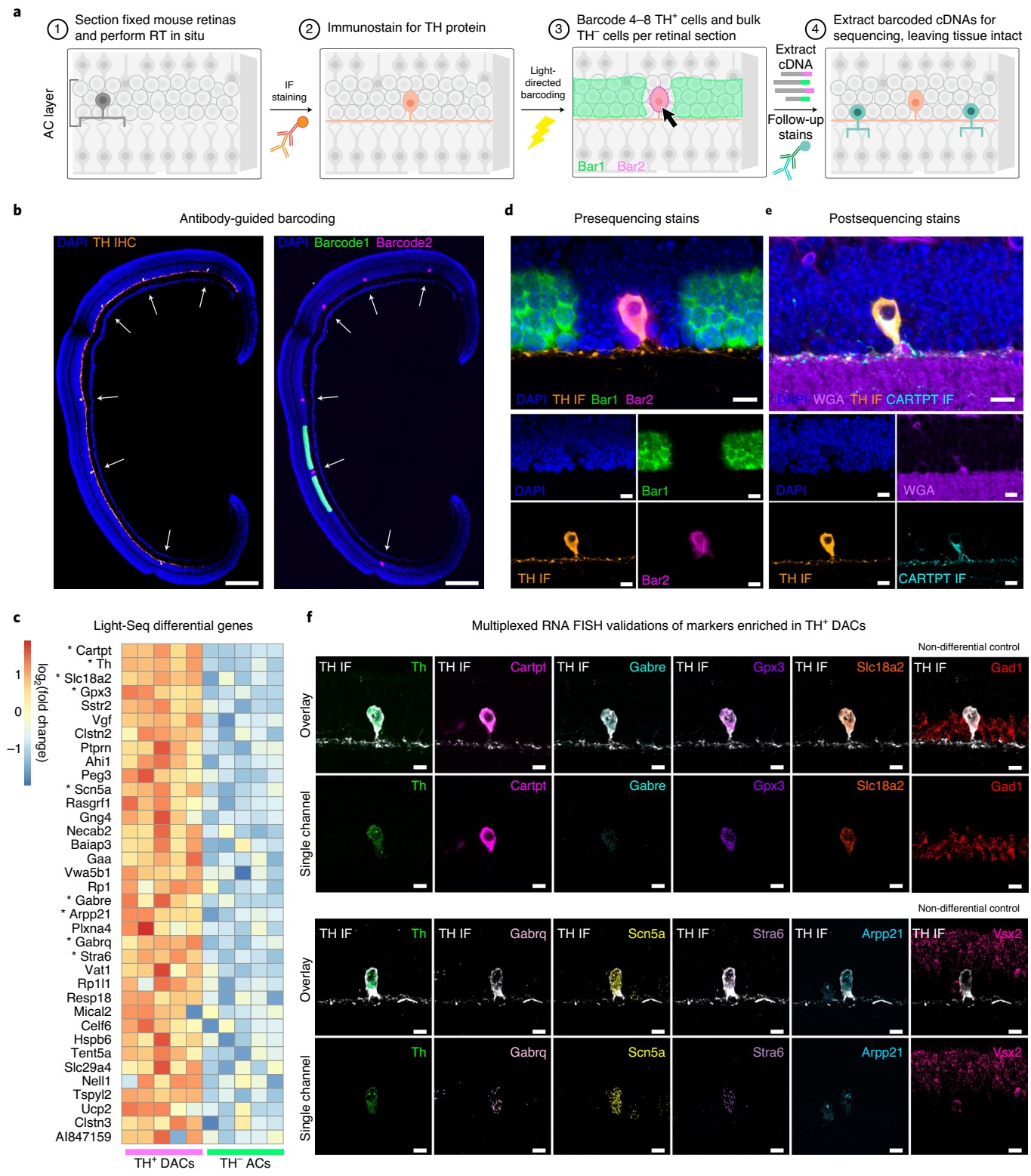

**a**
① Section fixed mouse retinas and perform RT in situ
② Immunostain for TH protein
③ Barcode 4–8 TH⁺ cells and bulk TH⁻ cells per retinal section
④ Extract barcoded cDNAs for sequencing, leaving tissue intact

**b** Antibody-guided barcoding

**c** Light-Seq differential genes

**d** Presequencing stains

**e** Postsequencing stains

**f** Multiplexed RNA FISH validations of markers enriched in TH⁺ DACs

population. Importantly, FISH data suggest that some markers had high expression levels (*Cartpt*, *Slc18a2*), while others were expressed more moderately (*Gabre*, *Gabrq*, *Stra6*), demonstrating that Light-Seq can accurately detect moderately expressed RNAs.

Several of the Light-Seq enriched markers have been previously reported in murine TH⁺ DACs: *Cartpt*[54], *Slc18a2* (*VMAT2*)[55,56] and gamma-aminobutyric acid (GABA) A receptor subunits *Gabre* and *Gabrq*[57]. Other transcripts, including *Stra6*, *Gpx3*, *Arpp21*, *Scn5a*

and most biomarkers identified here, have, to our knowledge, not yet been reported for this subtype.

## Discussion
Here, we present Light-Seq, a method to attach sequenceable spatial indices onto biomolecules in intact samples using light. By directly integrating two powerful domains, microscopy and NGS, Light-Seq enables linking of morphological and spatial parameters of target

cells to their transcriptomic profiles. Capture of morphology, tissue context, transcriptome and protein expression in the same cells provides a more comprehensive measurement of the state of cells and their interactions.

We demonstrated that Light-Seq can be used for full-transcriptome profiling of populations of 4–1,000+ cells within fixed tissue sections, with sensitivity similar to existing methods. Light-Seq produced UMI yields of 1,000–10,000 per $10 \times 10$-$\mu m^2$ unit area depending on the target cell type, comparable to DBIT-Seq (~5,000 UMIs)[17] and Slide-SeqV2 (500–1,000 UMIs)[12] for tissue areas of the same size (Supplementary Tables 3 and 4). Light-Seq sensitivity for transcript detection ranged between 1% and 10% for individual genes when compared with smFISH measurements, with a mean of $4.29 \pm 3.39\%$, similar to single-cell RNA sequencing sensitivity for the same set of genes (Fig. 4e and Extended Data Fig. 4h–j). Even with our conservative pipeline which includes only reads that mapped uniquely to the genome and exonic features, the sensitivity of Light-Seq is in line with existing spatial sequencing methods (0.005–15.5% (refs. [17,21,58,59]), as reviewed before[9]). We note that both sensitivity and UMIs per unit area measurements are generally highly impacted by the particular cell and tissue type, genes assayed and sequence-processing pipelines, which makes direct comparisons across technologies and applications imperfect.

We expect that Light-Seq's sensitivity can be further improved with optimization of the in situ RT and barcoding, such as by protease treatment, antigen retrieval or changes to fixation/permeabilization conditions[21,28,39,60], use of targeted ISH probes[61] and targeted ribosomal RNA depletion. These improvements, combined with the flexibility of custom photomasks, could ultimately enable profiling of single cells or subcellular compartments with higher efficiency. We will continue to update detailed protocols for applying Light-Seq at *lightseq.io*, including suggestions for optimization in different types of tissues.

Despite using different methods for RNA detection, we observed strong agreement between differentially enriched genes in single-cell Drop-Seq and Light-Seq data from the same cell types, such that 98.6% of significantly differential genes were enriched in the same retinal cell population in both datasets (Extended Data Fig. 4d). While the Drop-Seq data originated from dissociated cells among many retinas, Light-Seq requires far less cellular input (selected cells from only four 18 μm sections) and is not subject to loss of cells upon dissociation and the selection biases that this can produce.

Many existing spatial transcriptomics methods are complex and expensive to implement, regardless of the biological question of interest. Using standard NGS, Light-Seq circumvents many challenges associated with in situ sequencing and FISH approaches, which are limited by tissue autofluorescence and image deconvolution. The sequencing output of Light-Seq enables detecting not only RNA presence, but precise sequence information such as single nucleotide polymorphisms and splice isoforms, offering a major advantage over most ISH approaches (Supplementary Note 3). This approach could be particularly useful for tracking mutations and clonality in cancers or resolving microbial species in tissue samples. The cost for RT, A-tailing, three rounds of barcoding, displacement, cross-junction synthesis and PCR, on each section, is only ~$34.50 (Supplementary Table 5) per section. Photocrosslinking is feasible to perform with any standard optical imaging setup that can focus UV light on specified areas (for example, a microscope with a 365-nm UV LED and a DMD attachment, or a 405-nm laser on a confocal scanning or even light-sheet microscope for three-dimensional applications).

The DSP platform offers similar flexibility for spatial targeting by using iterative UV-cleavage and microcapillary collection of released barcodes[24], but this platform requires expensive equipment and targeted hybridization-based barcoding of mRNAs, and currently offers lower sensitivity, requiring a minimum of 20–300 cells

for sequencing. Another approach, PIC, similarly takes advantage of light-directed ROI targeting and shows higher sensitivity with single-cell barcoding, but is single-plex and destructive to the sample. These methods include differences in sample preparation and UMI recovery protocols which may be useful to consider for future variations of Light-Seq.

By spatially restricting the barcoding to targeted cDNAs and selectively amplifying them for library generation, Light-Seq allows the sequencing reads to be focused on cells of interest. Thus, unlike surface capture or microfluidic channel-based methods, sequencing depth and cost can be flexibly optimized, particularly for experiments targeting very few cells. This advantage is evident with our rare cell experiment (Fig. 5) where we used a single pooled MiSeq run (20 million reads) for all replicates. Experimentally decoupling the imaging and sequencing also makes the workflow highly flexible, and the light-directed barcoding allows addressing ROIs of different scales (from subcellular structures to large super-cellular regions) with the same reagents and experimental strategy.

We demonstrated how Light-Seq offers a simple and customizable workflow for studying very rare cell populations. We discovered and validated previously unknown biomarkers of TH+ DACs, which have not been captured by previous attempts to transcriptionally profile this population[53,54]. Our data confirm previously known markers, such as *Th*, *Cartpt* and *Slc18a2*, but also provide new leads for understanding the biology of dopaminergic retinal neurons. Several of these, such as *Arpp21* (ref. [62]), *Vgff*[63] and *Gpx3* (ref. [64]), are suggested to play a role in dopaminergic neurons or related diseases elsewhere in the nervous system, but many remain unstudied. Among the novel markers is *Stra6*, encoding a transmembrane vitamin A transporter that bidirectionally traffics retinol. Interestingly, this gene is additionally expressed in the retinal pigment epithelium and has been implicated in retinal disease, but the role of TH+ DACs in disease phenotypes remains unexplored[65,66]. The discovery of the tetrodotoxin-resistant voltage-gated sodium channel, *Scn5a*, is surprising and interesting. *Scn5a* encodes the main cardiac sodium channel, *Nav1.5*, and has been detected very rarely in neuronal populations[67,68]. Here, we report that *Scn5a* is highly expressed in TH+ DACs, but its function in these cells remains to be shown.

The nondestructive nature of Light-Seq leaves the sample intact after sequencing, potentiating multi-omic measurements from the same cells. We also envision that Light-Seq could be adopted for landmark-based transcriptomics (such as APEX-Seq[69]), without genetic intervention and sample destruction. While the combination of sensitivity, spatial resolution and ease of adoption of Light-Seq is collectively advantageous over existing technologies, the number of addressable regions is currently limited compared with other spatial methods. We expect that increasing the multiplexing would be highly feasible, as other published methods (for example, SABER[49,70], CycIF[71] or CODEX[72]) perform much longer (up to 60) serial cycles with repeated labeling, imaging and dehybridization/bleaching, and provide strong preceding evidence for good tissue preservation across more barcoding rounds. Scaling to combinatorial barcode construction and to labeling of other biomolecules (for example, proteome, epigenome) in future applications could support high-throughput barcoding and screening of hundreds to thousands of cells or regions with minimal increase in cost and is of great interest for further development.

## Online content

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

## Methods

**Oligo design and preparation.** CNVK-containing barcode sequences were screened with NUPACK[73] to have minimal secondary structure and were also checked against the mouse and human genomes using the BLAST[74] and BLAT[75] online tools. For in situ transcriptomic barcoding, an RT primer ending in five Ns and three Gs[40] was designed to contain a docking sequence complementary to a shared region in the barcode sequences (Supplementary Fig. 3). Barcode oligos were ordered from Gene Link, and all remaining oligos were ordered from Integrated DNA Technologies. Oligo stocks of 100 µM were stored at −20 °C, and working stocks of 10 µM in IDTE (Integrated DNA Technologies cat. no. 11-01-02-02) were prepared for most frequently used oligos. All sequence, purification and vendor information is listed in Supplementary Tables 5 and 6.

**Cell culture.** A stable HEK293-GFP cell line (SC001) that constitutively expresses eGFP under a CMV promoter was purchased from GenTarget and cultured in high-glucose D-MEM with GlutaMax supplemented with 10% fetal bovine serum, 0.1 mM MEM nonessential amino acids, 1% Pen-Strep (Thermo cat. no. 15140122) and 10 µg ml⁻¹ blasticidin (Thermo cat. no. J67216). A mouse 3T3 cell line was purchased from ATCC (CRL-1658) and cultured in D-MEM with GlutaMax (Thermo cat. no.10569-010) supplemented with 10% calf bovine serum (ATCC 30-2030) and 1% Pen-Strep. For the cell mixing experiment, cells were seeded overnight in D-MEM with GlutaMax supplemented with 10% calf bovine serum, 1% Pen-Strep and 0.1 mM MEM nonessential amino acids (Thermo cat. no 11140-050).

**Tissues.** All animal experiments were conducted in compliance with protocol IS00001679, approved by the Institutional Animal Care and Use Committee at Harvard University. Experiments were performed on tissue collected from postnatal day 18 wild-type CD1 IGS mice (Charles River, Strain Code 022).

**Barcoding setup.** Barcoding was performed on an inverted Nikon Eclipse Ti-E microscope with an attached Mightex Polygon 400 DP DMD. A Mightex BLS-series high-power liquid light guide-coupled LED source, 365 nm, 50-W emitter, was applied at 10% power for 10 s through a CFI plan fluor 10× objective per selected ROI per barcoding round (with the exception of Fig. 2b where at 10% power was applied for 5 s). For the glass surface or cell culture crosslinking experiments, focus was set to z-position corresponding to the top surface of the glass coverslip. For the retina layers experiment and barcoding of TH⁻ ACs, focus was set at 10 µm above the glass surface. For the TH⁺ DAC population, the focal plane of the TH-antibody stain was found for each cell, and barcoding was performed on the plane 5 µm above, to target the middle of the cells. Photocrosslinking was done on a single z-plane. Photomasks for the cell mixing and retina experiments were hand-drawn using the Bezier ROI tool and set as stimulation regions using the Nikon Elements (v.4.51) Polygon 400 module user interface. Refer to our protocols at lightseq.io and Supplementary Note 4 for detailed barcoding details, including calibrating and optimizing the optical setup.

**Retina tissue barcoding and immunofluorescence.** Neural retinas were dissected from postnatal day 18 mice in 1× PBS (Invitrogen AM9625, diluted in ultrapure water, Invitrogen cat. no. 10977) and immediately fixed for 25 min at room temperature in 1× PBS with 4% paraformaldehyde (diluted from 16% solution, Thermo Scientific cat. no. 28908) and 0.25% Triton X-100 (Sigma Aldrich T8787). Retinas were then washed 3 × 5 min in 1× PBS and once for 10 min in 7% sucrose in 1 × PBS (40-µm filter-sterilized), before getting embedded in a 1:1 solution (v:v) of OCT (Tissue-Tek 4583) and 30% sucrose in 1× PBS for freezing in cryomolds and subsequent storage at −80 °C. Cryosectioning was performed to cut 18 µm sections onto poly-L-lysine-coated 18-well ibidi chamber slides (ibidi custom order cat. no. 81814, #1.5 polymer). For comparing retinal layers (Fig. 4), four technical replicates were prepared by cutting four distinct retinal sections from the same animal into different wells. To promote tissue adhesion, chamber slides were coated before cryosectioning with an additional layer of poly-D-lysine (PDL) (Sigma P6407) dissolved at 0.3 mg ml⁻¹ in 2× Borate Buffer (diluted in water from Thermo Scientific PI28341, aliquoted and stored −20 °C). PDL coating was performed by covering the chambers with sterile PDL solution for 2 hours at 4 °C, removing the solution and drying completely and washing once with UltraPure water. After cryosectioning, retinas were dried briefly (~10 min) and washed three times in 1× PBS with 0.1% Tween-20 (vol/vol, Sigma Aldrich cat. no. P9416-50ML) (PBST).

The buffer was then replaced with an RT mix composed of 300 µM dNTPs (NEB N0447S), 0.5% Triton X-100 (vol/vol), 6 mM RNaseOUT (Invitrogen cat. no. 10777019), 1 µM RT primer and 8 U µl⁻¹ Maxima RT H Minus enzyme (Thermo Scientific cat. no. FEREP0753) in 1× RT buffer and incubated on a flat-top thermocycler (Mastercycler Nexus Flat, Eppendorf cat. no. 6335000020) with the following program: 30 min at 22 °C, followed by a 12-cycle ramp program of 8 °C for 30 s, 15 °C for 30 s, 25 °C for 30 s, 30 °C for 1 min, 37 °C for 1 min and 42 °C for 2 min. After a final 42 °C incubation for 30 min, samples were held at 4 °C temporarily. Following RT, samples were washed in PBST with 60% deionized formamide (vol/vol, Thermo Scientific cat. no. AM9342) for 3 × 5 min, in PBST with 1 M NaCl (Invitrogen AM9760G) for 2 × 2 min and in PBST for 2 × 2 min. Samples were kept in PBST until the buffer was exchanged with an A-tailing

master mix consisting of 1× ThermoPol Buffer (NEB cat. no. B9004S), 1 mM dATP (NEB N0446S), 25 µM ddATP (Sigma GE27-2051-01) and 1,000 U ml⁻¹ terminal transferase enzyme (NEB M0315L) for a 45-min incubation at 37 °C. After A-tailing, samples were washed in PBST for 3 × 1 min and stored overnight at 4 °C in PBST. Next day, before barcoding, the PBST was removed and fresh PBST with 1 M NaCl was added. For barcode hybridization, a barcoding solution consisting of PBST with 2 mg ml⁻¹ sheared salmon sperm DNA (Invitrogen AM9680), 10% dextran sulfate (wt/vol, Sigma Aldrich cat. no. S4030), 250 nM Barcode 1 strand (GATE.D12.B1) and 500 mM NaCl was applied for 30 min. Samples were then washed with PBST with 1 M NaCl (3 × 1 min). Last wash buffer was replaced with fresh PBST with 1 M NaCl before proceeding to the barcoding. Slide was then transferred to the microscope for light-directed barcoding. Desired regions were visually identified in the brightfield images and hand-drawn masks were set as photostimulation regions (see the Barcoding setup section). After photostimulation in the regions of interest, the chamber was removed from the microscope for washing. Samples were then washed with PBST with 60% deionized formamide (vol/vol) 8 times (four cycles of two buffer exchanges with 5-min incubation in between) and with PBST with 1 M NaCl (2 × 2 min). Last wash buffer was replaced with fresh PBST with 1 M NaCl for the next barcoding round. The same tissue area was then manually found on the microscope, and two additional barcoding rounds were performed with Barcodes 2 (GATE.D12.B2) and 3 (GATE.D12.B3). After the last barcoding round and washes, buffer was replaced with PBST.

Barcoded retinas were stained with DAPI for 30 min (0.5 µg ml⁻¹ in PBST) and imaged (Fig. 4 and Extended Data Fig. 3). After imaging, samples were treated with 67.5 µl of displacement mix consisting of 1× ThermoPol buffer and 250 U ml⁻¹ RNase H (NEB cat. no. M0297L) and incubated at 37 °C for 45 min. To maximize yield of recovery, a low-retention pipette tip was first coated with primer by pipetting up and down in a 20 nM solution, and then the same empty tip was used to pipette the displacement mix up and down several times within the well before collection. Each eluate was then transferred to a tube containing 1.6 µl of 1 µM cross-junction synthesis primer (GATC.20T), mixed and heat inactivated at 75 °C for 20 min in a PCR machine (Eppendorf Mastercycler Nexus Gradient). Then, 10.9 µl of cross-junction synthesis mix containing 1.15× ThermoPol buffer, 734 µM dNTPs and 5,872 U ml⁻¹ BST LF polymerase (NEB cat. no. M0275L) was added to each eluate. Samples were incubated in a PCR machine at 37 °C for 30 min followed by heat inactivation at 80 °C for 20 min.

After extraction, cryosections were washed twice in PBST and kept in fresh PBST at 4 °C until further analysis. Retinas were incubated with Tissue Blocking Solution (1× PBS, 0.3% Triton X-100 and 5% Normal Donkey Serum (Jackson ImmunoResearch cat. no. 017-000-121, RRID:AB_2337258) and filter-sterilized using a 40-µm syringe filter) for 1 hour at room temperature before antibody staining. The following primary antibodies were prepared in the Tissue Blocking Solution: sheep anti-CHX10 (Exalpha X1180P, RRID:AB_2314191, diluted 1:500) and rabbit anti-PAX6 (Abcam cat. no. ab195045, RRID:AB_2750924, diluted 1:300). Primary antibodies were incubated overnight at 4 °C and then washed 5 × 5 min with PBST. Secondary antibodies (donkey anti-sheep-Alexa647, Jackson ImmunoResearch cat. no. 713-605-147, RRID:AB_2340751; donkey anti-rabbit-Cy3, Jackson ImmunoResearch cat. no. 711-165-152, RRID:AB_2307443) were incubated overnight at 4 °C in Blocking Solution (both antibodies diluted 1:500 from 50% glycerol stock) and washed 5 × 5 min with PBST. Wheat germ agglutinin (WGA, Biotium cat. no. 29022-1) diluted 1:100 in PBST from a 1 mg ml⁻¹ stock solution was applied for 1 hour at room temperature, followed by 30 min of staining with DAPI in PBST (0.5 µg ml⁻¹).

After cross-junction synthesis, the extracted sequences were quantified and then bulk amplified as follows. First, 5-µl volumes of reactions were combined in a 1:1 ratio with a PCR mix consisting of Sybr Green I (1×, Invitrogen S7563), Kapa HiFi Buffer (2×, from Roche KK2502), forward and reverse primers (600 nM each, GATE and GATC sequences; see Supplementary Table 6), dNTPs (600 µM, from Roche KK2502) and Kapa HiFi Hot Start Polymerase (0.04 U µl⁻¹, from Roche KK2502). Samples were amplified on a quantitative PCR machine (Biorad CFX Connect Real-Time System) with the following program: 98 °C for 3 min, followed by 30 cycles of 98 °C for 20 s, 60 °C for 30 s and 72 °C for 2 min. Samples were then incubated for a final 5 min at 72 °C, and a melt curve measurement was performed. Based on the quantification, bulk amplification was performed under the same conditions for 20 cycles using all of the remaining reaction material.

The extracted material for each of the four technical replicates was prepared separately, and samples were later pooled for sequencing as below (Library preparation and sequencing). Of the total tissue area of the sections, 7–11% (Supplementary Table 3) was estimated to be labeled with barcodes based on the size of the barcoded area over the total retina tissue area. Barcoded area was measured using the area of the photomask, and total retina tissue area was measured by the free-hand area tool in FIJI[76].

**Barcoding TH⁺ DACs.** Retinas were prepared and sectioned as described above for six section replicates. Light-Seq was performed as described for the retinal layers in Fig. 4, with few changes: After in situ RT and A-tailing and before barcoding, sections were stained with an anti-TH antibody (Millipore cat. no. AB152, RRID:AB_390204) for 1 hour at room temperature. The antibody was diluted 1:500 in Blocking Solution, which was made of 1× PBS, 0.1% Tween-20

and 1% molecular grade BSA (GeminiBio, 700-106 P) and was filter-sterilized using a 20-μm syringe filter. Samples were washed 5 × 5 min with PBST. Donkey anti-rabbit-Alexa647 secondary antibody (Jackson ImmunoResearch cat. no. 711-605-152, RRID:AB_2492288) was added at 1:250 dilution in the same Blocking Solution for 30 min at room temperature and then washed 5 × 3 min with PBST.

Barcoding was then performed as described above with minor changes, listed here. The first round targeted the FITC-barcode strand (Barcode 3, GATE.D12.B3 sequence, annotated as Barcode 1 in Fig. 5a) to TH⁻ ACs, and the second round targeted the Cy3-barcode strand (Barcode 2, GATE.D12.B2) to TH⁺ DACs (see Supplementary Table 6 for barcode sequences). The z focal plane for barcoding TH⁺ DACs was guided by the anti-TH antibody signal with an additional 5-μm adjustment away from the well surface. Stringent washes after light-directed barcoding in PBST with 60% formamide were doubled in number (four cycles of 5-min incubations, where each cycle consisted of four buffer exchanges, rather than two buffer exchanges as done for the retinal layers). All TH⁺ DACs in the same section were addressed individually across multiple fields of view and were crosslinked with Barcode 2 after manual selection. Of the six section replicates prepared, five yielded enough material for sequencing during the PCR amplification of the extracted sequences. For all replicates, the number of TH⁺ DACs was between four and eight, and therefore the TH⁺ DAC transcriptomes came from a pool of four to eight cells per technical replicate.

**Immunostaining after sequencing TH⁺/⁻ ACs.** After sequencing, we revisited the same barcoded cells from two of the original sections to validate the enriched expression of the top hit (lowest $P_{adj}$), $Cartpt$, at the protein level. The samples were stored in PBST at 4 °C for >10 days while the sequencing was run and analyzed. For staining, samples were incubated with a goat anti-CARTPT antibody (Thermo Fisher Scientific cat. no. PA5-47170, RRID:AB_2607700) at 1:20 in Blocking Solution for 1.5 hours at room temperature and then washed 5 × 5 min with PBST. Then, a donkey anti-goat-Alexa488 secondary antibody (Jackson ImmunoResearch cat. no. 705-545-003, RRID:AB_2340428) was added at 1:250 in the same Blocking Solution for 30 min at room temperature and then washed 5 × 3 min with PBST. WGA-Rhodamine (Vector Labs cat. no. RL-1022-5, 1:100 from 1 mg ml⁻¹ stock) and DAPI were added to the secondary antibody mixture to label cell membranes and nuclei.

**RNA-FISH validation of TH⁺ DAC markers.** For the top ten markers (with $P_{adj} < 0.05$, $\log_2(\text{fold change}) > 3$), we validated using RNA-FISH. We used serial SABER-FISH to detect the markers in new 25 μm retinal sections as previously described[49]. FISH validations were done on four new sections from a mouse littermate of the source animal used for Light-Seq experiments. SABER-FISH probes were designed using the 'RNA Probe Design' feature of the PaintSHOP tool (https://oligo.shinyapps.io/paintshop/)[77]. For each gene, all probes were appended with a common gene-specific SABER-FISH primer sequence for orthogonal detection of multiple genes in the same cells. See Supplementary Table 7 for probe sequences (with their attached primer) targeting each gene. Previously described probe sets were used for $Gad1.26$ and $Vsx2.25$ (ref. [49]). SABER-FISH probe preparation and RNA detection were performed as described before (by Kishi et al., 2019, 'User-friendly protocol: Retina Tissue Sections RNA-FISH' section of the Supplemental Protocols[49]). The $Sstr2.25$ SABER-FISH probe set failed to extend during the probe synthesis reaction (before being applied to the tissue samples) and therefore was excluded from the validation experiments.

Probes were split into two groups for multiplexed detection, with three rounds of fluorescent detection done to capture six RNAs total in each sample (representative images shown as rows in Fig. 5f). Each round detected two different genes using ATTO565 and Alexa647 fluorescent oligos, as described before[49] (sequences included in Supplementary Table 7). The antibody stain for TH was performed after the first round of fluorescent detection, as described in the section above, and therefore was present during all rounds of sequential imaging. WGA-405S (Biotium cat. no. 29022-1) was added during the secondary antibody incubation at 1:100 for membrane staining.

**Barcoding and immunofluorescence on cultured cells.** Eighteen-well poly-L-lysine-coated ibidi chambers (ibidi custom order cat. no. 81814) were coated with sterile PDL (Sigma Aldrich cat. no. P6407) at 0.3 mg ml⁻¹ overnight at 4 °C. Afterwards, the chamber was dried for 1 hour, washed with UltraPure water (Invitrogen cat. no. 10977) and dried again before cell seeding. Chambers were then seeded with ~4,000 HEK293 and ~5,000 NIH/3T3 cells per well and placed in an incubator (37 °C with 5% CO₂) overnight. Samples were gently washed with DPBS (pre-warmed to 37 °C, Gibco cat. no. 14190-144) and fixed in 4% formaldehyde (wt/vol, Thermo cat. no. 28908) in 1× PBS (Invitrogen cat. no. AM9625) for 10 min at room temperature. Then, samples were washed twice with 1× PBS and permeabilized with 0.25% Triton X-100 (vol/vol) for 10 min. Samples were then washed twice with 1× PBS, and the in situ RT step was done following the same protocol described for retina samples above. Following RT, samples were washed in PBST and 60% deionized formamide for 3 × 2 min, in PBST with 1 M NaCl for 2 × 2 min and in PBST for 2 × 2 min. Samples were kept in PBST at 4 °C (as a pausing point in the protocol). To proceed further, the PBST buffer was exchanged with the A-tailing master mix (as described above for tissues) and incubated at 37 °C for 30 min.

After A-tailing, samples were washed in PBST for 3 × 1 min and left in PBST with 1 M NaCl until barcoding. For barcode hybridization, a barcoding solution consisting of PBST with 250 nM Barcode 1 strand (GATE.D12.B1) and 500 mM NaCl was applied onto the samples for 15 min and excess strands were washed with PBST with 1 M NaCl for 3 × 1 min. Last wash buffer was replaced with fresh PBST with 1 M NaCl, and slide was then transferred to the microscope for light-directed barcoding. Selected HEK cells were then photocrosslinked using hand-drawn photomasks (see Barcoding setup for more details). Samples were then washed with PBST with 60% deionized formamide (vol/vol) eight times (four cycles of two buffer exchanges with 2-min incubations in between) and followed by 2 × 2 min washes with PBST with 1 M NaCl. The last wash buffer was replaced with fresh PBST with 1 M NaCl before the next barcoding round with Barcode 2 (GATE.D12.B2, for 3T3 cells), which was performed with identical barcode incubation and washing protocol, except after all washes, the liquid in the wells was replaced with PBST. See Supplementary Note 2 for suggested protocol updates to reduce the nonspecific background, and see *lightseq.io* for the latest suggested protocols.

After barcoding, samples were treated with 67.5 μl of displacement mix consisting of 1× ThermoPol buffer and 250 U ml⁻¹ RNase H (NEB M0297L) and incubated at 37 °C for 30 min. cDNA collection, RNase H heat inactivation and cross-junction synthesis were done as described above for retina samples.

After extraction of cDNAs, cells were washed twice in PBST and then kept in fresh PBST at 4 °C until further analysis. For the cell mixing experiment, multiplexed IF was performed on one well. All antibodies were spun down at 10,000 g for 10 min at 4 °C before use. Cells were incubated with a primary antibody mix containing goat anti-Lamin-B (sc-6216, RRID:AB_648156), mouse anti-TFAM (MA5-16148, RRID:AB_11157422) and rat anti-alpha Tubulin (MA1-80017, RRID:AB_2210201) diluted 1:75 in 1× PBS, 0.3% Triton X-100 and 5% BSA (Jackson ImmunoResearch cat. no. 001-000-162) for 1 hour. After 2 × 1 min washes in 1× PBS, cells were incubated with secondary antibodies anti-mouse-Alexa647 (Jackson ImmunoResearch 715-605-150, RRID:AB_2340862), anti-goat-Cy3 (VWR 102649-368) and anti-rat- Alexa488 (Jackson ImmunoResearch 712-545-150, RRID:AB_2340683) diluted 1:150 (from 50% glycerol stocks) in 1× PBS with 0.3% Triton X-100 and 5% BSA for 1 hour. Cells were washed in 1× PBS 2 × 1 min and then incubated with 4 μg ml⁻¹ DAPI (Invitrogen cat. no. D1306) in 1× PBS for 5 min, and washed 2 × 1 min with 1× PBS, followed by imaging in fresh 1× PBS.

After cross-junction synthesis, extracted sequences were quantified as in the retina tissue barcoding and immunofluorescence section above. Bulk amplification was then performed with the same conditions for 24 cycles but starting with about half (40 μl) of the starting material. The other half of the sample material was kept to test alternative PCR kits (all the results shown in figures in this work were obtained with the Kapa HiFi kit, which we chose as our standard amplification method).

For cell mixing experiments, on average, ~25 cells of each type were pooled from a population of ~4,500 cells per well for each of the three technical replicates. Extracts were separately prepared from each replicate and pooled for sequencing as below. The total cell number was estimated from a manual count of ~130 cells in a 1-mm² area of a single well and extrapolating to the total surface area.

**Library preparation and sequencing.** Library preparation and sequencing were performed the same for both the cell mixing and retina tissue experiments. After PCR, samples were stored at −20 °C then purified with a 1.2× ratio of Ampure XP Beads (Beckman A63881) and eluted in water. Next, tagmentation was performed with a Nextera XT Library Preparation Kit (Illumina cat. no. FC-131-1096), but using custom primers for the i5 end. Tagmentation was performed using the standard manufacturer protocols and reagents (TD buffer, ATM, NT buffer, NPM PCR master mix) on 2 ng of sample in 20 μl reactions containing 10 μl of TD buffer and 5 μl of ATM for 5 min and at 55 °C, and reactions were stopped with 5 μl of NT buffer and held on ice. To each tube, 6.5 μl of water, 1.75 μl of Nextera i7 primer, 1.75 μl of custom i5 primer (GATE*.P5* in Supplementary Table 6) and 15 μl of NPM PCR master mix were added. Reactions were incubated at 72 °C for 3 min, 95 °C for 15 s, then 12 cycles of: 95 °C for 15 s, 55 °C for 15 s and 72 °C for 40 s. After a final incubation at 72 °C for 1 min, samples were held at 10 °C. Reactions were then purified with 0.9× Ampure XP Beads and eluted in water. Samples were stored at −20 °C until sequencing.

Sequencing was performed either by GeneWiz on an Illumina HiSeq machine or by the Biopolymers Facility at Harvard Medical School on an Illumina NovaSeq machine using custom Read 1 and i5 primers and 30% Phi-X spike-in. For the cell mixing data, all replicate sequences were pooled on a single lane of a HiSeq 4000 flowcell, although one replicate was also sequenced on its own lane. For the retina layers experiment (Fig. 4), all replicate sequences were pooled together and sequenced in both lanes of a NovaSeq 6000 flowcell. For the retina amacrine experiment (Fig. 5), all replicates were pooled together and sequenced with a single Illumina MiSeq run.

**In vitro surface barcoding.** An eight-well ibidi ibiTreat chamber (ibidi cat. no. 80826) was functionalized with BSA-biotin and streptavidin for in vitro surface barcoding tests. BSA-biotin (Sigma A8549) and streptavidin (Invitrogen S-888) solutions were diluted to 1 mg ml⁻¹ and 0.5 mg ml⁻¹ in 1× PBS, respectively. Then, 200 μl of the BSA-biotin solution was pipetted onto an empty ibidi well and

incubated for 5 min at room temperature. Afterwards, the BSA-biotin solution was aspirated out of the well and washed twice with 1× PBS. The streptavidin solution was then added to the well and also incubated for 5 min at room temperature, followed by three 1× PBS washes.

The functionalized surface was then incubated with a biotinylated strand (Supplementary Table 6) at 1 μM in 1× PBS for 5 min at room temperature, followed by three 1× PBS washes. Hybridization solutions were made up of a 200 nM CNVK barcoding strand in 1× PBS with 1 M NaCl.

Photomasks for the cat photo and Penrose triangle in Fig. 2b,c were generated as binary .tif files which were then uploaded and mapped onto the DMD chip. The binary image of the cat was generated using Adobe Photoshop (v.2021) diffusion dither function on a picture taken with a personal camera. Photomasks for the individual portions of the Penrose triangle were hand-drawn in Adobe illustrator (v.2021) and saved as an 8-bit grayscale .tif file.

Hybridization of the barcoding strands proceeded sequentially with each photomask corresponding to a unique barcoding strand and fluorophore (the cat photomask crosslinking was done as a single round (using the Barcode 2 strand) and with crosslinking). Each barcoding round introduced a barcoding strand in hybridization solution followed by a 5-min incubation at room temperature and UV photocrosslinking as described in the Barcoding setup section above. Afterwards, noncrosslinked strands were removed with a stringent wash with 40% formamide in 1× PBS for 2 × 2 min. The formamide was then removed with 2 × 1-min washes in 1× PBS with 1 M NaCl to prepare the chamber for the next round of barcoding.

**Imaging setups.** In control experiments to monitor the signal after RT, after barcoding and after cDNA extraction, we scanned the samples with an ImageXpress Micro-4 system (Molecular Devices) equipped with a custom 5-mm liquid light guide Gen III Spectra LED-based light engine (solid-state 377/54, 438/29, 475/28, 511/16, 555/28, 576/23, 635/22, 730/40), Semrock filters (Zero Pixel Shift Filter Cubes: for DAPI and 750/Cy7 LED-Da/Fi/Tr/Cy5.Cy/5x-A Penta Band; for 488/GFP LED-FITC-A Single Band; for 565/Cy3 LED-TRITC-A Single Band; for 647/Cy5 LED-Cy5-A Single Band) and an Andor Zyla 4.2 camera controlled with the MetaXpress software (v.6.5.3.427). Tiling was performed with 10% overlap.

A fully motorized Nikon Ti-2 inverted microscope was used to image fluorescent retina samples. This confocal microscope was equipped with a Yokogawa CSU-W1 single spinning disk (50-μm pinhole size) and a Nikon linear-encoded motorized stage with Mad City Labs 500-μm range Nano-Drive Z piezo insert, and an Andor Zyla 4.2 plus (6.5-mm photodiode size) sCMOS camera using a Nikon Apo λS LWD 40×/1.1 DICN2 water immersion objective lens with Zeiss Immersol W 2010. Fluorescence was acquired from 405 nm, 488 nm, 550 nm and 640 nm by sample illumination with directly modulated solid-state lasers 405-nm diode 100-mW (at the fiber tip) laser line, 488-nm diode 100-mW laser line, 561-nm 100-mW diode-pumped solid-state laser line and 640-nm diode 70-mW laser line, in a Toptica iChrome MLE laser combiner, respectively. For all channels, a hard-coated Semrock Di01-T405/488/568/647 multi-bandpass dichroic mirror was used. Images were captured with 16-bit Dual Gain (high dynamic range camera mode). Nikon Elements AR 5.02 software was used during acquisition. Z-stacks were acquired using a Piezo Z-device (shutter closed during axial movement). Data were saved and exported as ND2 files.

For Extended Data Fig. 1c–g, a Leica SP5 X MP inverted laser-scanning confocal microscope was used to create custom scan regions using the 'FRAP' module. Point-scanning was performed through an HCX PL APO CS ×63.0 1.20 water objective with a 405-nm diode laser set to the highest power. Scan speed was set to the slowest rate of 10 Hz and scanned twice across the ROIs.

**Image processing and analysis.** For images from the cell mixing experiment, multi-channel images from each round were registered and stitched using ASHLAR[78] (v.1.12.0). Brightfield images for the cell mixing and retina images were manually contrasted for best visibility. Hand-drawn ROIs for the cell mixing and retina experiments were saved as binary .tif files and converted to a vector image with Adobe Illustrator's image trace function and then overlaid onto the brightfield images. Multi-channel fluorescent images for the cell mixing were scaled and overlaid based on the following: Each image channel's pixel values were separated as foreground or background pixels with an Otsu threshold. A linear normalization from 0 to 1 was then applied to each image, with the maximum pixel value of 1 set to the 95th percentile of foreground pixel values. Images were then false-colored and blended into an overlay image with a custom Python script that is equivalent to Adobe Photoshop's screen blend function. The cat and Penrose triangle fluorescent images were manually contrasted for best visibility. The stitched overlay image in Fig. 3d was prepared in OMERO[79] (v.5.4.6.21).

Retinal images in Fig. 4b,f are from a single Z-plane, extracted from composite multi-channel images in FIJI[76] (v.2.0.0-rc-69/1.52n). The minimum and maximum intensity levels were manually chosen to linearly scale the pixel intensities for optimal display and adjusted using the *Image → Adjust → Brightness/Contrast* window in ImageJ. Single-channel images in Extended Data Fig. 3b,c were adjusted using the 'Auto' setting under *Image → Adjust → Brightness/Contrast*, and the minimum intensity value was then set to zero.

Figures were assembled in Adobe Illustrator (v.2021 and 2022).

**Sequencing data processing and differential gene expression analysis.** Parsing and mapping of sequencing data were performed on the Harvard Medical School O2 cluster (Kernel 3.10.0) with Python (3.7.5), PyTables (3.6.1), samtools (1.9 and 1.12), pysam (0.17.0), numpy (1.21.4), pandas (1.3.4), Biopython (1.79) and scikit-bio (0.5.6). See the Code availability statement for the GitHub repository containing all code and virtual environment parameters. The pipeline for sequencing analysis is outlined in Extended Data Fig. 4a, with a breakdown of read distributions for each replicate depicted in Extended Data Fig. 4b. Barcode, UMI and cDNA mapping sequences (up to 40 nt) were extracted from Read 1 (R1) reads using the UMI-tools (v.1.1.1) package[80]. Sequences were then mapped to the appropriate genome (Human v38 or Mouse vM27) or a merged genome using the STAR aligner[81] (v.2.7.9a), with multimapped alignments discarded in all cases.

After mapping, the featureCounts[82] tool was used to assign genomic mappings to genes, and then reads were deduplicated (per gene) with the UMI-tools dedup command. For mouse transcript mapping, reads corresponding to two genes (ENSMUSG00000119584.1 and ENSMUSG00000064337.1, a ribosomal RNA and a mitochondrial rRNA, respectively) were discarded as the number of mapped sequences came close to or exceeded the number of possible UMIs and caused the deduping process to stall. For the human and cell mixing experiment, reads that did not map were then compared with the eGFP transcript sequence with the Striped Smith Waterman algorithm[83] and were considered eGFP reads if they had a score of at least 40 and the UMI was unique. For the mouse and human cell mixing experiment, discrimination values were calculated based on mapping to a merged human and mouse genome, and UMI counts were estimated based on mapping separately the human-barcoded reads to the human genome and mouse-barcoded reads to the mouse genome.

After gene assignment and UMI deduplication, reads were parsed out by their DNA barcode sequence by exact matches only with custom Python scripts. Normalized expression levels for the cell mixing experiment were calculated as $\log_2$-transformed TPM ($\log_2(\text{TPM} + 1)$). Gene enrichment analysis for the retina tissue experiment was performed in R with the DESeq2 package[45,84]. Genes were considered enriched if their adjusted P values ($P_{adj}$, with the Benjamini–Hochberg method) were under 0.05. For Fig. 4d, markers for each layer were the set of genes that were enriched relative to both other layers, plotted in R 3.6.1 with pheatmap function (v.1.0.12). For Fig. 4c, genes with adjusted $P > 0.05$ are plotted in black (at the bottom of the graphs), while genes that were significantly enriched in the pairwise comparisons are colored based on their layer enrichment (e.g., magenta, enriched in ONL, Barcode 1).

To compare with existing Drop-Seq data[46] for Extended Data Fig. 4e–g, pseudo-bulk RNA sequencing data were modeled based on single-cell counts by pooling the counts from all cells of each cell type that are known to localize within the targeted layers. Single cells from the Drop-Seq dataset were classified as cell types as outlined in the GitHub markdown (at https://github.com/broadinstitute/BipolarCell2016). The pseudo-ONL was constructed by pooling together the rod and cone photoreceptor cells, and the pseudo-BCL was constructed by pooling all bipolar subtype clusters and the Müller glia. Number of transcripts per cell (Supplementary Table 3) was estimated by dividing by the number of cells pooled for each layer.

For Light-Seq, the number of barcoded cells per region was estimated based on the area of the barcoded region (Extended Data Fig. 3) and manual counting of the number of DAPI-stained nuclei within three-dimensional confocal images of the barcoded regions. For each layer, the number of DAPI-stained nuclei was counted within a subset of the barcoded area (~5,000 μm²), and the total number was estimated by linearly scaling based on the precise size of the full area. Since the BCL contains both bipolar cells and Müller glia, the number of bipolar cells within the BCL was estimated as 72% of the total DAPI-stained nuclei within the layer[85]. In Fig. 4c, markers that were in enriched in one of the ONL or BCL layers in either assay were compared, and in Fig. 4d, markers that were enriched relative to both other layers were plotted based on their DESeq2 z score[45]. In Extended Data Fig. 4f, markers that were enriched in the BCL in both assays were plotted.

Scripts used for sequencing data processing, mapping and analysis have been posted to GitHub, along with cell count estimates for the pseudo-bulk comparisons and step-by-step instructions for running the code.

**Sensitivity estimation.** To estimate the sensitivity of Light-Seq, we compared sequencing read counts with published smFISH data for a set of 16 bipolar cell markers, which were captured in the BCL Light-Seq data[50]. From this previous publication, we used the smFISH puncta-per-cell counts to estimate the expected number of total transcripts detected, based on the number of bipolar cells captured in the Light-Seq BCL area and the known average transcript per cell counts within the BCL. To estimate the number of bipolar cells captured within the Light-Seq BCL area, we overlaid the ROI outline with the DAPI-stained barcode image. First, the full arclength of the region, $\text{arc}_{total}$, was measured in FIJI using the *Segmented Line* tool followed by *Analyze → Measure*. Then, a small portion of the arclength, $\text{arc}_{small}$, was measured (~200 μm) in the same way, and the number of DAPI-stained nuclei within the ROI were counted for the $\text{arc}_{small}$ area ($\text{cells}_{small}$). The total number of nuclei in the ROI was predicted by scaling: (no. of total cells) = ($\text{arc}_{total}$ × $\text{cells}_{small}$)/$\text{arc}_{small}$ (Supplementary Table 3).

With an estimate of the number of bipolar cells within the BCL, we then used the smFISH data for the 16 marker genes with single-cell expression counts published in West et al. 2022 (ref. [50]) to predict how many transcripts should be present within a bipolar cell population of the measured size. Since the published data were single-cell transcript (smFISH puncta) counts for each of the 16 marker genes for all bipolar subtypes (at their measured ratios), we chose to exclude all cells of type BC1B from the smFISH data and the Drop-Seq data because their cell bodies are located within the AC layer and would not be within the Light-Seq BCL (as shown in previous work, particularly in Fig. 3 of Shekhar et al. 2016 (ref. [46])). To exclude BC1B from the smFISH single-cell gene expression matrix from West et al. 2022 (ref. [50]), we removed all rows from 'Retina1.csv' (available on https://github.com/ewest11/Bipolar-Serial-SABER-FISH-Analysis) with 'Subtype' = 2. For Drop-Seq data, the cells belonging to the BC1B cluster were similarly removed.

With the remaining gene expression counts across all other bipolar subtypes, we averaged across all cells to obtain an 'average transcript per cell count' for the BCL. This average was then scaled by our estimated cell numbers (listed in Supplementary Table 3 for each replicate) to obtain the expected number of transcripts per Light-Seq BCL replicate and per Drop-Seq replicate. Sensitivity was plotted with MATLAB 2018a's *boxplot* function, with default settings.

We note that the mice in West et al. 2022 (ref. [50]) were injected with EdU and BrdU for cellular birth dating and the Drop-Seq cells were dissociated for sequencing, which should be considered as potential sources of variance.

**Chimeric read analysis.** R1 and R2 reads for each replicate from the TH⁺ DAC experiment were mapped and deduplicated separately. The R1 and R2 deduplicated read files were then merged, sorted by read name and iterated through to identify pairs where both R1 and R2 mapped to transcripts. The numbers of pairs that mapped to the same transcript versus different transcripts are reported in Supplementary Note 1. For this analysis, Python v.3.10.4 was used on a MacBook Pro (2021) with macOS Monterey (v.12.2.1).

**Intron analysis.** To analyze intronic reads, the -t gene flag of featureCounts[82] was used to map to genes rather than just exons. Then, the RSeQC[86] (v.4.0.0) read_distribution.py program was used to profile the numbers of reads from UTR exonic, CDS exonic and 10-kb regions upstream and downstream for each replicate. Further details and counts can be found in Supplementary Note 3. For this analysis, Python v.3.10.4 was used on a MacBook Pro (2021) with macOS Monterey (v.12.2.1).

**Gene length bias analysis.** Gene lengths as reported by featureCounts[82] (v.2.0.1) were used to profile read counts from transcripts of different lengths for each barcode for each replicate. Histograms were generated to show the distribution of transcript counts across different transcript lengths. Box plots were then generated to show transcript counts for transcripts within different length ranges, with bins chosen based on mouse embryonic stem cells full-length comparisons[87] in R (v.4.1.3). RPKM values were calculated for the same bins by dividing the read counts by the length (in kilobases) of the transcript and dividing again by a scaling factor calculated as the number of reads from the condition divided by 1 million.

**Gene body coverage analysis.** Files containing aligned, deduplicated reads in BAM file format were input to the RSeQC[86] geneBody_coverage.py program to generate the gene body coverage plots. The reference BED file input to RseQC was generated by converting the comprehensive gene annotation GFF3 file (vM27) to a 12-column BED file using conversion utilities hosted by the UCSC Genome Browser[88] (specifically, gff3ToGenePred and genePredToBed (v.1.04.00)). These data were visualized in IGV (v.2.12.3)[89].

**Reporting summary.** Further information on research design is available in the Nature Research Reporting Summary linked to this article.

## Data availability
Detailed protocols for barcoding experiments are accessible online on the protocols.io platform, and up-to-date protocols and resources can be found at *lightseq.io*. These protocols cover the following: (1) in situ reverse transcription and A-tailing, (2) in situ spatial barcoding, (3) displacement and extraction of barcoded cDNA sequences, (4) cross-junction synthesis, (5) PCR amplification and (6) library preparation via tagmentation. Raw sequencing data are available online in NCBI's Gene Expression Omnibus and are accessible through GEO Series accession number GSE208650. The gene mappings and counts for human–mouse cell mixing experiment replicates are provided in the source data for Extended Data Fig. 2. The full lists of differentially enriched genes enriched between layers in the retina tissue experiment are provided in the source data for Fig. 4. The full list of differentially enriched genes enriched between TH⁺ and TH⁻ cells in the rare retinal AC tissue experiment is provided in the source data of Fig. 5. All SABER-FISH probe sequences are provided in Supplementary Table 7. Correspondence and requests for materials should be addressed to J.Y.K., S.K.S., P.Y. or C.L.C.

## Code availability
Code is available on GitHub at https://github.com/Harvard-MolSys-Lab/Light-Seq-Nature-Methods-2022. This includes the code for image analysis, sequence analysis and differential gene expression analysis and plotting.

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

## Acknowledgements
The authors acknowledge funding from the Wyss Institute (through their Validation Project Program for J.Y.K., N.L., P.Y. and S.K.S.; the Molecular Robotics Initiative; and a Technology Development Fellowship to J.Y.K.), the National Institutes of Health (under grant nos. UG3HL145600, UH3CA255133, DP1GM133052, R01GM124401, RF1MH124606 and RF1MH128861 to P.Y.), the Office of Naval Research (under grant no. N00014–18–1–2549 to P.Y.), the Chan Zuckerberg Initiative (under grant no. 2019-02433 to P.Y.), the European Molecular Biology Laboratory (S.K.S.) and Howard Hughes Medical Institute (E.R.W., C.L.C.). The authors acknowledge the MicRoN Core at Harvard and P. Montero Llopis for imaging equipment and expertise. We thank Kylie the cat for being our feline model for photomask generation. We thank I. Goldaracena for assistance with cell experiments, J. Immen for help with graphic design and T. Ferrante, J. Rosenberg, J. McDonough and B. Bedell for discussion and feedback. Several figure panels (Figs. 1, 2a and 3b–c) were partially created with BioRender.com.

## Author contributions
J.Y.K. conceived and led the study, designed and performed experiments, wrote software, analyzed data and wrote the manuscript. N.L. conceived the study, designed and performed experiments, wrote software, analyzed data and wrote the manuscript. E.R.W. designed and performed experiments, wrote software, analyzed data and wrote the manuscript. K.S. contributed to experimental design, method optimization and data analysis. J.J.J. performed experiments and contributed to optimization of the method and writing of the manuscript. M.S. provided experimental assistance. C.L.C. contributed expertise and supervision, experimental design and data analysis. S.K.S. conceived the study, provided scientific and technical guidance, designed and performed experiments, analyzed the results and wrote the manuscript. P.Y. conceived and supervised the study, designed the experiments and wrote the manuscript. All authors edited and approved the manuscript.

## Competing interests
J.Y.K., N.L., P.Y. and S.K.S. are inventors on patent applications covering the method. Multiple authors are involved in commercialization of the technique and engage with

Digital Biology, Inc. (J.Y.K. and E.R.W. are co-founders and employees; P.Y. is co-founder, equity holder, director and consultant; S.K.S. is anticipated to be a consulting scientific co-founder; N.L. is a consulting founding scientist; J.J.J. is an employee.) P.Y. is also a co-founder, equity holder, director and consultant of Ultivue, Inc. The remaining authors declare no competing interests.

## Additional information

**Extended data** are available for this paper at https://doi.org/10.1038/s41592-022-01604-1.

**Correspondence and requests for materials** should be addressed to Jocelyn Y. Kishi, Constance L. Cepko, Sinem K. Saka or Peng Yin.

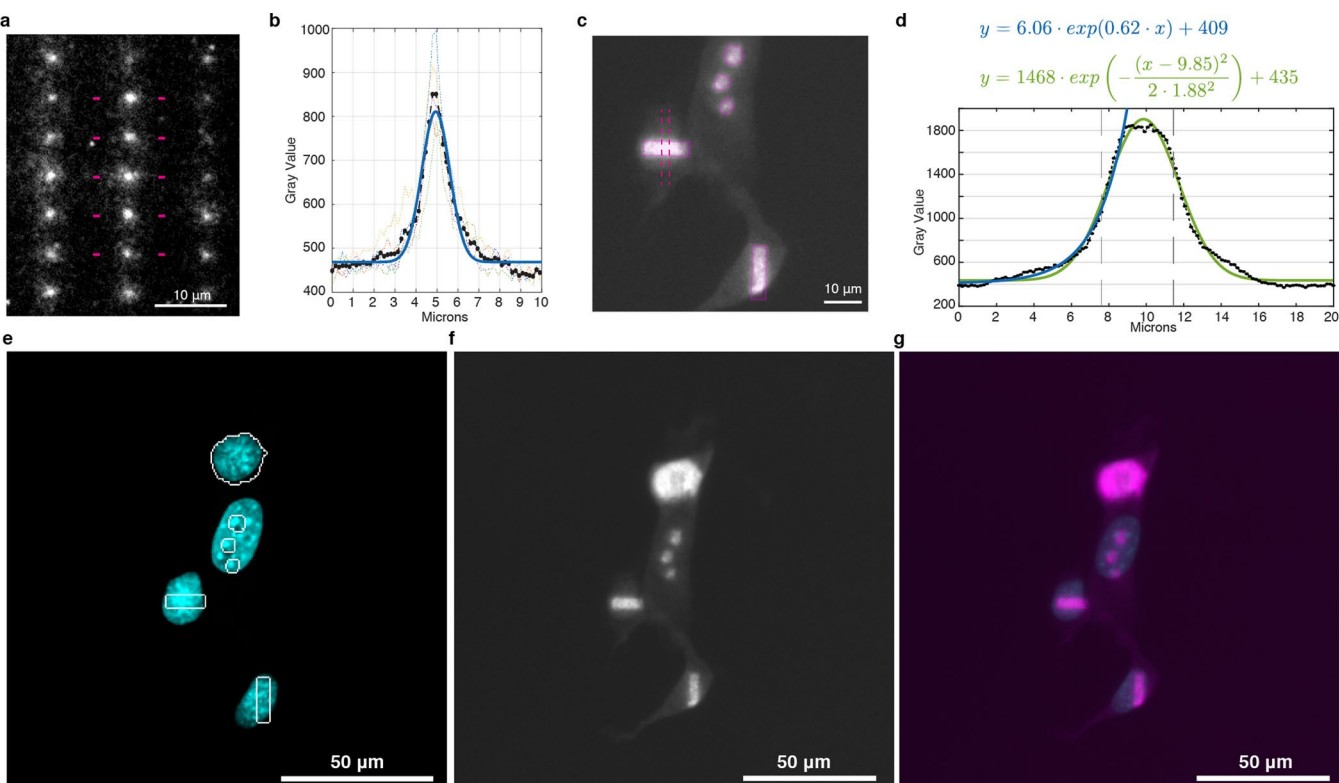

**Extended Data Fig. 1 | High resolution light-directed DNA barcoding. (a)** Fluorescent image of a dot array printed onto a glass slide functionalized with DNA docking sequences. Dots were printed through targeted photocrosslinking of fluorescent DNA barcode strands to complementary docking sites (see also Fig. 2a). Five dots were chosen for a profile scan of gray values (magenta dashes), pixel contrast set to 450–800. **(b)** Linescans from panel **a** (dotted colored lines) were averaged into a single linescan (black dots with dashes). Averaged linescan was fit to a Gaussian curve (blue). A single dot corresponds to a single activated DMD mirror, estimated to illuminate a 0.76 µm diameter area. FWHM from the fit was ~1.56 µm. **(c)** Subcellular labeling of 3T3 cells with a 405 nm laser on a point-scanning confocal microscope ($n = 4$ cells from a single field of view). The photomask used for crosslinking was scaled to the size of the fluorescent image and manually overlaid (magenta) to aid in visualization. A profile scan was performed on the rectangular area between the magenta dashed lines. **(d)** Intensity profile of the dotted box from panel **c**, data was fit to a Gaussian curve (green) with a measured FWHM of 4.4 µm. A second exponential decay was fitted to one-half of the profile scan (blue) to calculate a 84–16% criterion, the distance across which the signal drops from 84 to 16% of the maximum value. Distance of the 84–16% drop was calculated from the exponential fit to be 2.67 µm. Vertical dashed lines indicate the estimated ROI boundary from the photomask. Width of the photomask was estimated to be 4 µm. **(e-g)** The single field of view in panel **c** imaged on a confocal scanning microscope. Nuclear signal (cyan) with the ROI selection (white lines) overlaid **(e)**, fluorescent Cy3 barcode after stringent washes of non-crosslinked strands **(f)**, overlay with lower contrast display of nuclear signal to enable visualization of the overlapping Cy3 signal **(g)**.

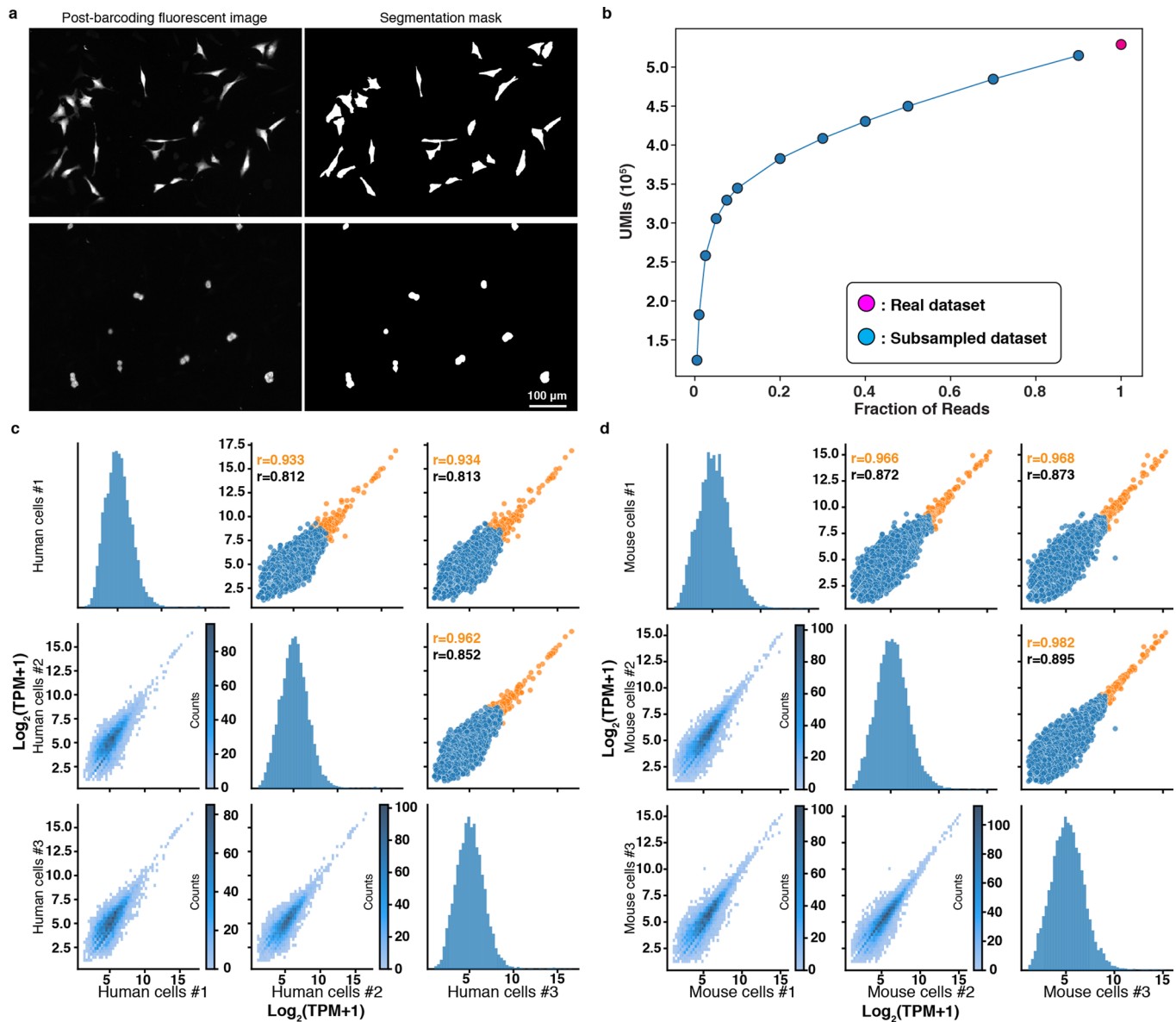

**Extended Data Fig. 2 | Cell segmentation and read counts of 3T3 and HEK cells. (a)** Representative segmentation results from the fluorescent signal of the mouse (top row) and human (bottom row) barcoded cells (representative from $n = 3$ technical replicates). Masks were used to calculate barcoded area (also see Methods and Supplementary Table 1). **(b)** A single ~200 million read dataset from the cell mixing experiment (magenta) was mapped to a merged genome and subsampled by fraction of reads without replacement and processed with the UMI deduplication pipeline. Average number of UMIs from 5 simulated datasets are shown (cyan). **(c-d)** Scatterplots and histograms of normalized expression level ($\log_2(\text{TPM}+1)$) between the three technical replicates for cells. We only considered the genes detected across all replicates ($\log_2(\text{TPM}+1)$ cutoff of $\geq 1$). Highlighted data points (orange) indicate top 200 genes, remaining genes are colored blue. Pearson correlation for all genes (black) and top 200 genes (orange) reported for **(c)** human cells and **(d)** mouse cells. Histograms of $\log_2(\text{TPM}+1)$ distributions (excluding top 200 genes) for each replicate are plotted on the diagonals. Full list of gene mappings and counts is provided as Source Data Table 1.

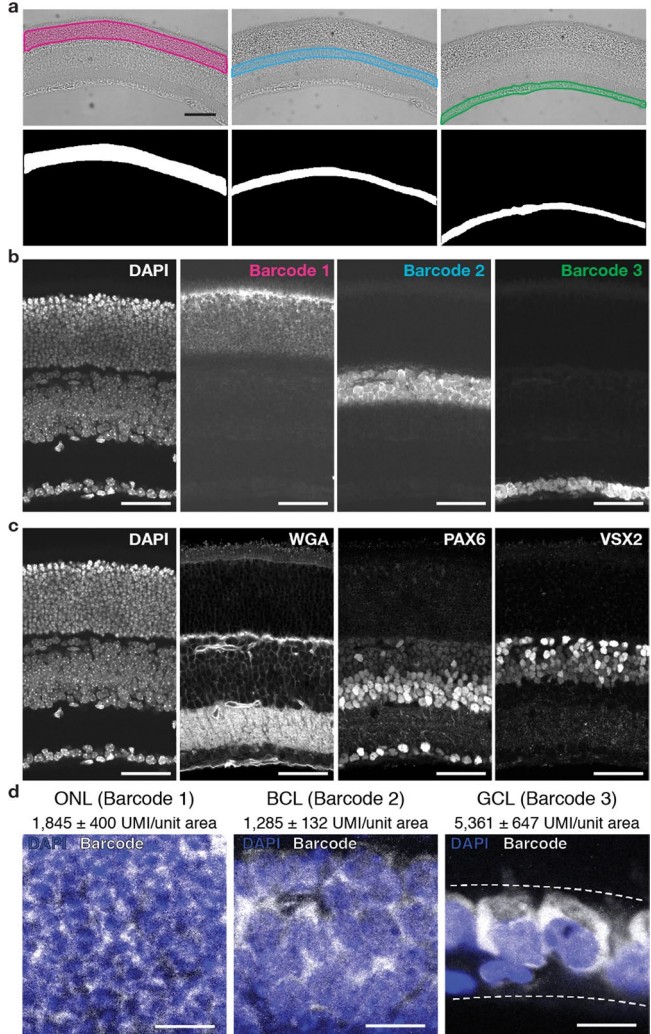

**Extended Data Fig. 3 | Barcoding of retinal layers. (a)** Brightfield images of a mouse retina cryosection with the barcoded area overlaid. From left to right, Outer Nuclear Layer (ONL), Bipolar Cell Layer (BCL), Ganglion Cell Layer (GCL). Scale bar is 100 μm. Binary images show the selected ROIs that were used as barcoding photomasks. Pixel size is 1.6 μm/pixel. **(b)** Single Z-plane spinning disc confocal images taken after barcode crosslinking for DAPI and the fluorescent barcodes 1–3 (labeled with Cy5, Cy3 and Fluorescein, respectively). **(c)** Single Z-plane images of DAPI, WGA, and immunofluorescence for PAX6 and VSX2 proteins in the same barcoded cells after recovery of barcoded cDNAs for sequencing. Images in **(b)** and **(c)** are displayed with auto scaling (with minimum set to zero). Scale bars are 50 μm. **(d)** Single Z-plane spinning disk confocal images of barcode fluorescence (white) overlaid with DAPI (blue) within each barcoded layer, displaying the different cellular morphologies with differences in cell size, cytoplasmic area, RNA density for each cellular layer that is comprised of different cell types. Scale bars are 10 μm. UMI counts per unit area (10 μm x 10 μm) are listed for each barcoded layer. Panels **a-d** are representative images from n = 4 technical replicates.

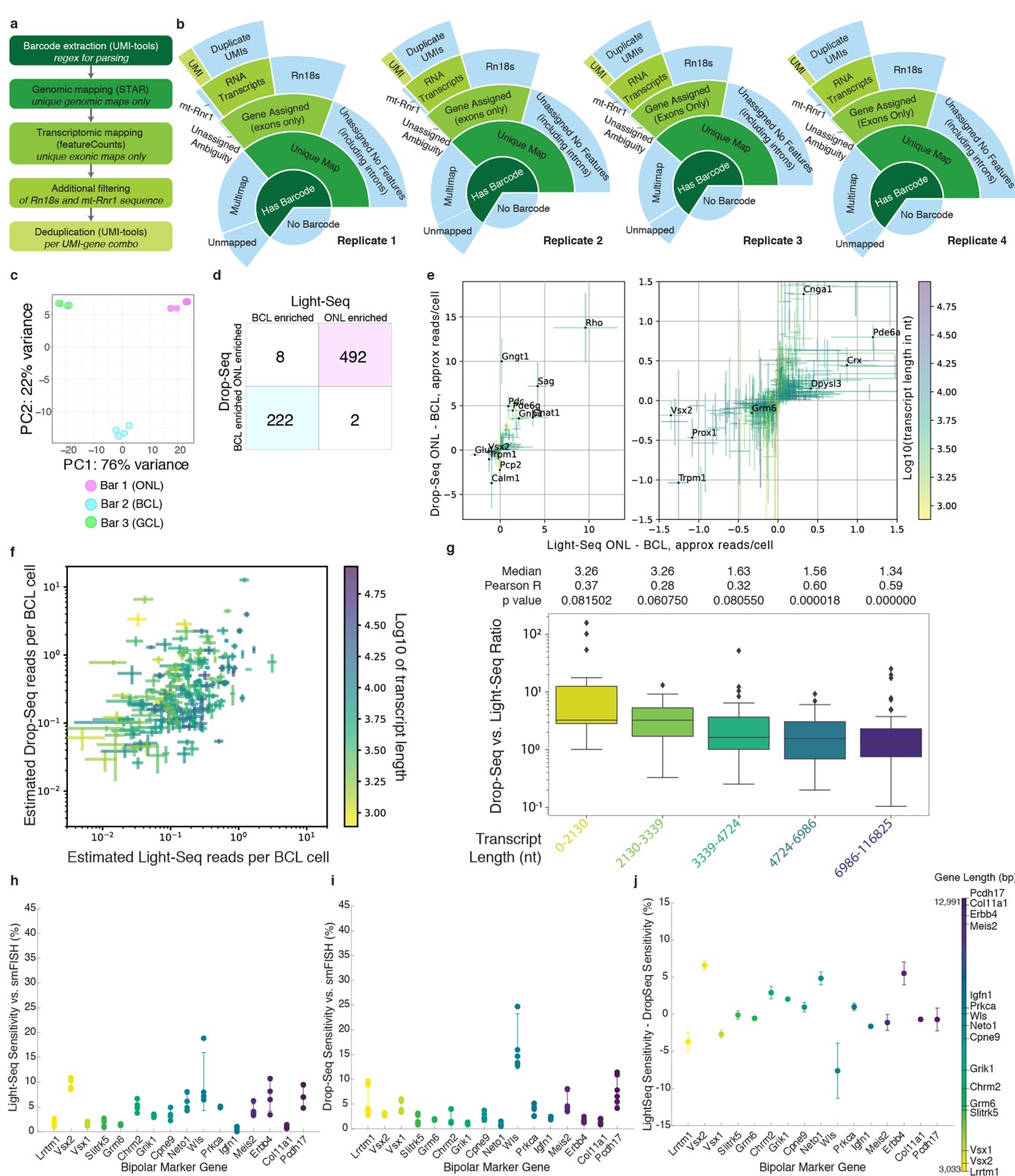

**Extended Data Fig. 4 | See next page for caption.**

**Extended Data Fig. 4 | Sequencing metrics and sensitivity of Light-Seq for retinal layer experiment. (a)** Sequence processing pipeline. **(b)** Sunburst plots depicting fractions of reads filtered at each processing step. **(c)** PCA plot of Light-Seq replicates (n = 4 technical replicates per layer). **(d)** Correlation matrix of genes enriched in ONL versus BCL in Light-Seq and Drop-Seq data[46] ($p_{adj} < 0.05$, two-sided Wald test with Benjamini-Hochberg adjustment for multiple hypothesis testing). **(e)** Subtraction-based enrichment of approximate difference in Light-Seq transcripts per cell between ONL and BCL to simulated difference Drop-Seq transcripts per cell between the ONL and BCL. Genes significantly enriched in either the ONL or BCL in both assays are plotted ($p_{adj} < 0.05$, two-sided Wald test with Benjamini-Hochberg adjustment for multiple hypothesis testing). Zoom of the plot shown on right. **(f)** Estimated reads per cell in Drop-Seq (simulated BCL) versus Light-Seq BCL data (barcode 2) for genes enriched in the BCL in both assays ($p_{adj} < 0.05$). **(g)** Boxplots of mean Drop-Seq (n = 6 sample replicates) vs Light-Seq (n = 4 section replicates) counts per gene per for different transcript lengths. Pearson R and median ratio shown. Median line and quartiles bound the box, with whiskers marking 1.5× the interquartile range. **(h)** Sensitivity of Light-Seq relative to smFISH for 16 BCL marker genes with published single-cell smFISH data[50]. Based on the number of cells within the BCL, sensitivity calculated as [# expected transcripts by smFISH]/[# observed Light-Seq reads]. Dots represent the sensitivity of a single replicate (n = 4 replicates). Error bars show standard deviation, centered at the mean. **(i)** Sensitivity of Drop-Seq relative to smFISH. Based on the number of cells in the pooled bipolar clusters in Shekhar et al., 2016[46], sensitivity calculated as [# of expected transcripts by smFISH]/[number of observed Drop-Seq reads] (see Methods). Dots reflect sensitivity of a single replicate/gene (n = 6 sample replicates). Error bars reflect standard deviation centered around the mean. **(j)** Difference between mean Light-Seq and Drop-Seq sensitivity per gene from **(h)** and **(i)**. Error bars show standard error for the difference of means. For panels **h-j**, genes are arranged and colored by ascending gene length.

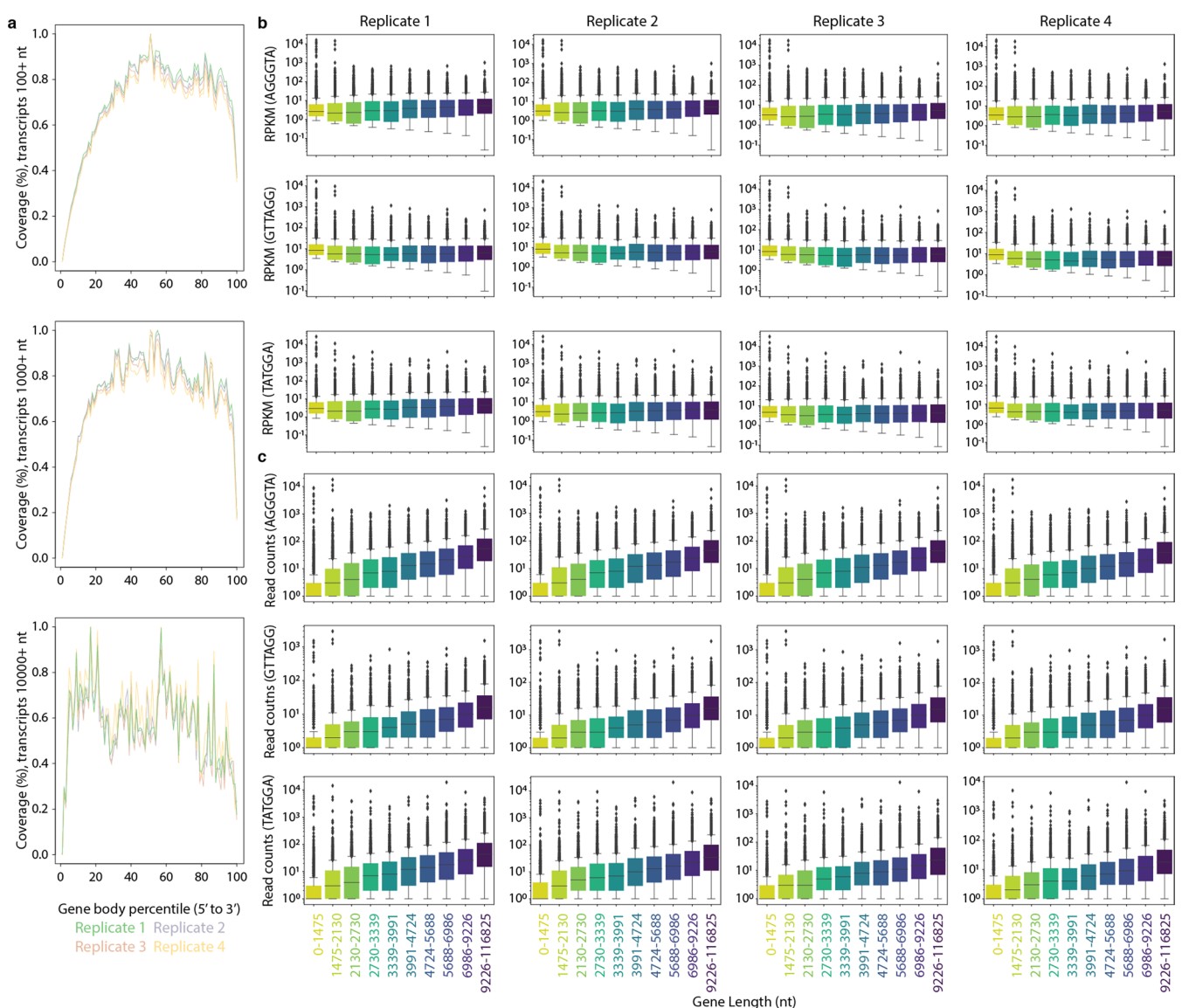

**Extended Data Fig. 5 | Gene body coverage and gene length read distributions for retinal layers. (a)** Gene body coverage for transcripts 100 nt and up (top), 1000 nt and up (middle), 10000 nt and up (bottom). **(b-c)** Reads Per Kilobase of transcript, per Million mapped reads (RPKM) (panel **b**) and read counts (panel **c**) for all barcode-replicate conditions across different transcript length bins (bins based on Phipson et al., 2017[87]). All box plots show median line and quartiles bounding the box, with whiskers marking 1.5× the interquartile range. $n=4$ technical replicates.

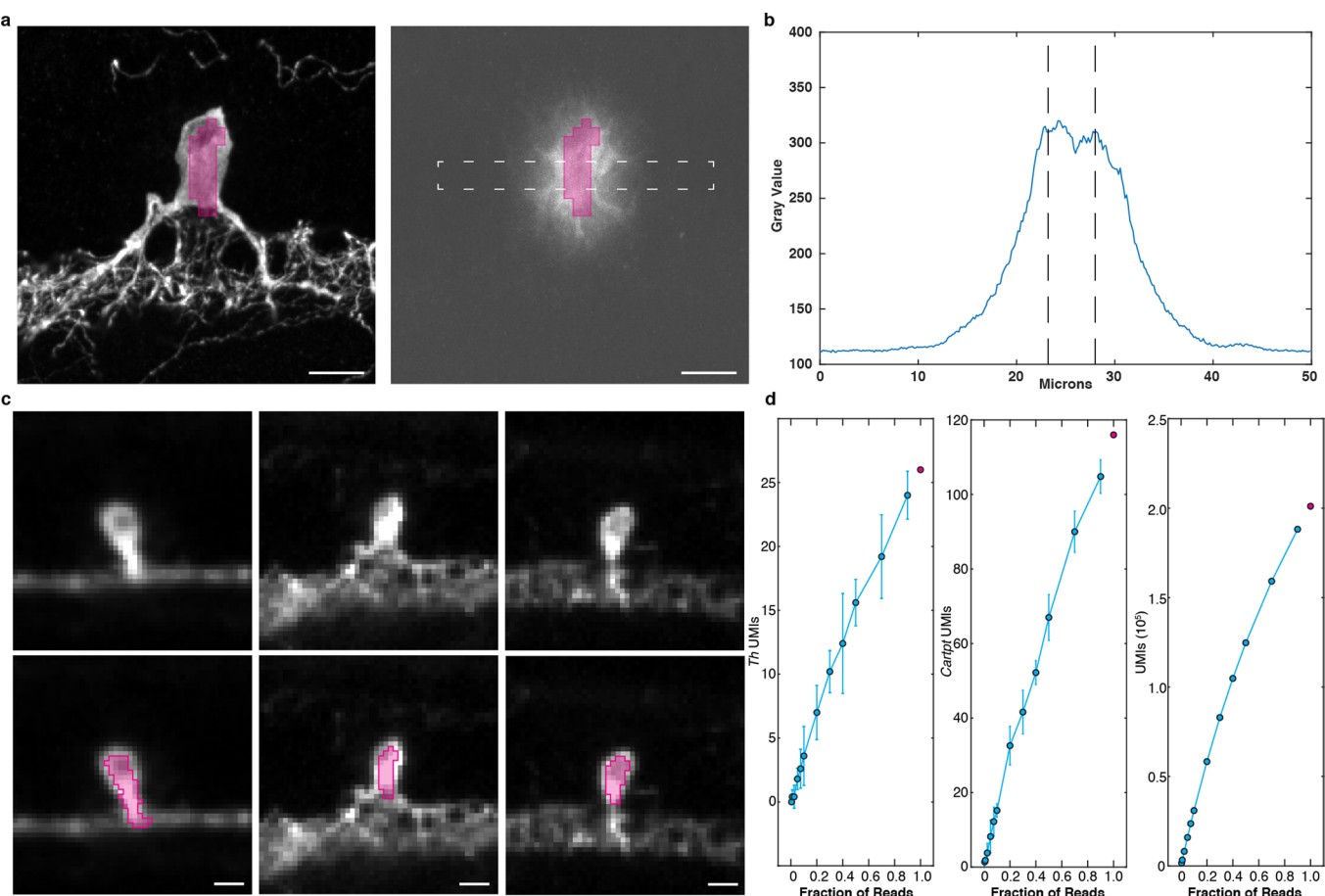

**Extended Data Fig. 6 | TH+ AC ROI selection and signal. (a)** TH IF of a TH+ AC imaged on a confocal microscope with a 40× objective with barcoding ROI overlaid (magenta, left). Fluorescent image of the barcoded region (right). Pixel value set to 0–400. **(b)** Profile scan performed on the 50×5 μm rectangle across the fluorescent barcode signal (dotted box in panel **a**). Dashed vertical lines indicate ROI boundary. **(c)** Selected images of single TH+ amacrine cells stained with IF (top), with the ROIs overlaid (magenta, bottom). ROIs were drawn slightly inside the cell bodies to account for light-scattering at the ROI boundary. Scale bars are 10 μm. Panels **(a)** and **(c)** are representative images from n=5 technical replicates. **(d)** A single sequencing run depth at 3.7 million read depth from a representative experimental condition was selected for subsampling without replacement illustrating UMI scaling by fraction of reads for the genes *Th*, *Cartpt*, and total UMIs. Mean +/- standard deviation of 5 simulations (cyan) are plotted, with full dataset represented as a single point (magenta).

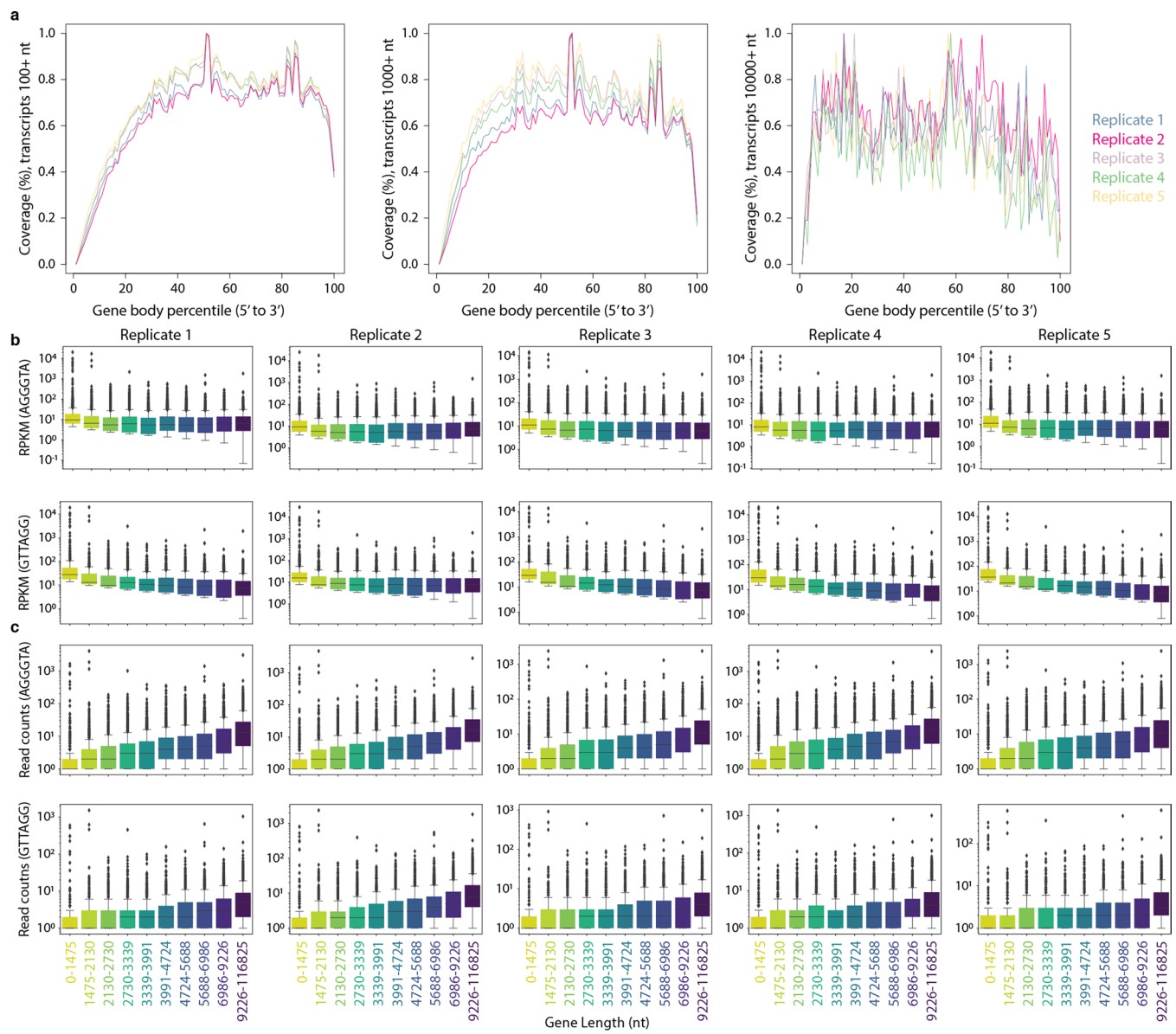

**Extended Data Fig. 7 | Gene body coverage and gene length read distributions for amacrine cell experiment. (a)** Gene body coverage for transcripts 100 nt and up (left), 1000 nt and up (middle), 10000 nt and up (right). **(b-c)** Reads Per Kilobase of transcript, per Million mapped reads (RPKM) (in **b**) and read counts (in **c**) for all barcode-replicate conditions across different transcript length bins. All box plots show median line and quartiles bounding the box, with whiskers marking 1.5× the interquartile range. *n* = 5 technical replicates.

# Reporting Summary

## Statistics

For all statistical analyses, confirm that the following items are present in the figure legend, table legend, main text, or Methods section.

| n/a | Confirmed | |
|---|---|---|
| ☐ | ☒ | The exact sample size (*n*) for each experimental group/condition, given as a discrete number and unit of measurement |
| ☐ | ☒ | A statement on whether measurements were taken from distinct samples or whether the same sample was measured repeatedly |
| ☐ | ☒ | The statistical test(s) used AND whether they are one- or two-sided *Only common tests should be described solely by name; describe more complex techniques in the Methods section.* |
| ☒ | ☐ | A description of all covariates tested |
| ☒ | ☐ | A description of any assumptions or corrections, such as tests of normality and adjustment for multiple comparisons |
| ☐ | ☒ | A full description of the statistical parameters including central tendency (e.g. means) or other basic estimates (e.g. regression coefficient) AND variation (e.g. standard deviation) or associated estimates of uncertainty (e.g. confidence intervals) |
| ☐ | ☒ | For null hypothesis testing, the test statistic (e.g. *F*, *t*, *r*) with confidence intervals, effect sizes, degrees of freedom and *P* value noted *Give P values as exact values whenever suitable.* |
| ☒ | ☐ | For Bayesian analysis, information on the choice of priors and Markov chain Monte Carlo settings |
| ☒ | ☐ | For hierarchical and complex designs, identification of the appropriate level for tests and full reporting of outcomes |
| ☐ | ☒ | Estimates of effect sizes (e.g. Cohen's *d*, Pearson's *r*), indicating how they were calculated |

*Our web collection on statistics for biologists contains articles on many of the points above.*

## Software and code

Policy information about availability of computer code

| Data collection | Microscope control: MetaXpress software (version 6.5.3.427) Nikon Elements AR 5.02 software Leica LAS AF (for Leica SP5) Nikon elements (version 4.51) polygon 400 module user interface for DMD control |
|---|---|
| Data analysis | Image processing: Cell Profiler 4.2.1 FIJI (version 2.0.0-rc-69/1.52n) HMS OMERO (version 5.4.6.21) ASHLAR 1.12.0 scikit-image 0.19.0 Adobe illustrator (v. 2021, 2022)<br><br>Sequencing data processing and analysis: HMS O2 cluster - Kernel 3.10.0 STAR - 2.7.9a Python 3.7.5 PyTables 3.6.1 UMI-tools 1.1.1 samtools 1.9, 1.12 featureCounts (subread) 2.0.1 pysam 0.17.0 numpy 1.21.4 |

```
pandas 1.3.4
Biopython 1.79
scikit-bio 0.5.6
R 3.6.1
DESeq2 1.26.0
pheatmap 1.0.12

Plotting:
seaborn 0.11.2
matplotlib 3.5.0
ggplot2 (R 3.6.1)
Matlab 2018a

Visualization:
IGV 2.12.3

Gene body coverage and chimeric read analysis:
MacBook Pro (2021) with macOS Monterey (v12.2.1)
RSeQC v4.0.0
Python 3.10.4
R 4.1.3
UCSC gff3ToGenePred and genePredToBed scripts
GNU sed 4.2.2

A Github repository with all code may be found at: https://github.com/Harvard-MolSys-Lab/Light-Seq-Nature-Methods-2022
```

For manuscripts utilizing custom algorithms or software that are central to the research but not yet described in published literature, software must be made available to editors and reviewers. We strongly encourage code deposition in a community repository (e.g. GitHub). See the Nature Portfolio guidelines for submitting code & software for further information.

## Data

Policy information about availability of data

All manuscripts must include a data availability statement. This statement should provide the following information, where applicable:
- Accession codes, unique identifiers, or web links for publicly available datasets
- A description of any restrictions on data availability
- For clinical datasets or third party data, please ensure that the statement adheres to our policy

Code for image analysis, sequence analysis, and differential gene analysis is accessible on GitHub at https://github.com/Harvard-MolSys-Lab/Light-Seq-Nature-Methods-2022.

Raw sequencing data is deposited online (GEO Series accession number GSE208650). Genes identified in the human-mouse cell mixing experiment are listed in Source Data Table 1 (for Extended Data Fig. 2 and Supplementary Table 2). Differentially enriched genes enriched across layers in the retina tissue experiment are listed in Source Data Table 2 (for Fig. 4).  The full list of differentially enriched genes enriched between TH+ and TH- cells in the rare retinal amacrine cell tissue experiment is provided in Source Data Table 3 (for Fig. 5).

# Field-specific reporting

Please select the one below that is the best fit for your research. If you are not sure, read the appropriate sections before making your selection.

☒ Life sciences          ☐ Behavioural & social sciences          ☐ Ecological, evolutionary & environmental sciences

For a reference copy of the document with all sections, see nature.com/documents/nr-reporting-summary-flat.pdf

# Life sciences study design

All studies must disclose on these points even when the disclosure is negative.

Sample size | For cell mixing experiments ~25-30 cells were pooled from a population of ~4,500 cells per well for each of the technical replicates. The sample size was not pre-determined, but the results were validated by determining the species-specific mapping of transcriptomes from each targeted cell population. For the retinal layer experiments, 4 tissue sections were prepared from retinas dissected from 1 mouse. This number of replicates was not statistically pre-determined, but was chosen to try to recover statistically significant gene expression profiles from a small amount of starting material. 10-15% of the cells in each section were in situ barcoded for RNA-sequencing. These sample sizes are sufficient for the conclusions drawn, as the conclusions were limited to the genes which were found to be statistically differential between cell populations. For amacrine cell experiments, 4-8 TH+ cells were barcoded and pooled per section replicate (5 successful replicates in total, 1 replicate which was unsuccessful). This number was not statistically determined, but was taken as the total number of TH+ cells present within each section. Since this number fell between 4 and 8 for all sections, we ended up with a range of 4-8 TH+ cells. TH- amacrine cells transcriptomes came from a pool of roughly 500 cells per replicate (5 in total). This number was similar to the region size chosen in the prior retinal layer experiment, where we recovered consistent gene expression profiles across 4 replicates from the other cellular layers, and therefore we hypothesized that this sample size would be sufficient to find interesting and robust biological insights. These sample sizes were sufficient for the conclusions drawn, as the conclusions were limited to genes which were statistically differential between the TH+ and TH- cell populations. Of course, it is possible that larger sample sizes would result in more statistically differential genes across these populations,

but our findings suggest that these sample sizes produced gene expression signatures consistent with orthogonal validation experiments, as covered in Figures 4 and 5.

| | |
|---|---|
| Data exclusions | For amacrine cell experiments, 6 section replicates were prepared for barcoding, but only 5 yielded enough material during the PCR amplification of the extracted sequences and  was hence sent for sequencing. Hence the sequencing data comes from 5 technical section replicates. |
| Replication | The cell-mixing experiment was done with 3 technical replicates where for each replicate, cells in a separate well were labeled and separately prepared and pooled for sequencing. All attempts at replication were successful. In the case of the mouse retina experiments, 4 technical replicates were performed where for each replicate a different section from the same source tissue was used to barcode and image 3 retinal layers and the extracted material for each replicates was prepared separately and later pooled for the sequencing run. All attempts at replication were successful. For evaluation and quantification of our method, multiple biological replicates were not accumulated to avoid unnecessary use of animal tissue material. For amacrine cell experiments, 6 section replicates were prepared for barcoding, but only 5 yielded enough material during the PCR amplification of the extracted sequences and was hence sent for sequencing. <br><br>For amacrine cell RNA-FISH validation experiments, 4 technical replicates were prepared per probe (one littermate mouse). 2 of the originally barcoded sections were revisited for validation antibody stainings after barcoding. All attempts at replication were successful. |
| Randomization | Randomization was not necessary for this study, since each experiment was designed to have technical replicates from the same biological specimens. For the cell culture experiment, multiple wells of the same chamber slide were simultaneously plated with human and mouse cells and processed in the same way. For the retina experiments, the sections were adjacent sections from the same retina, placed into distinct chambers on the same chamber slide, and therefore randomization was not necessary. For smFISH validation of the TH+ amacrine cell markers, the markers chosen for validation were chosen based on defined criteria (log2Fold enrichment >3 and padj<0.05) and all markers fitting this criteria were validated (with the exception of the single marker for which enough ISH probes could not be designed). These validations were performed in new animals. |
| Blinding | Blinding was not relevant to this study, since all technical replicates for each experiment were the same. Each replicate had multiple internal conditions (e.g. human vs. mouse cells in the same well, different cell layers in the retina), but these needed to be known to correctly and deterministically barcode each population. For analysis, blinding was also not relevant, since the sequences were separated by their spatially indexed DNA barcode for the statistical analyses. |

# Reporting for specific materials, systems and methods

We require information from authors about some types of materials, experimental systems and methods used in many studies. Here, indicate whether each material, system or method listed is relevant to your study. If you are not sure if a list item applies to your research, read the appropriate section before selecting a response.

## Materials & experimental systems

| n/a | Involved in the study |
|---|---|
| ☐ | ☒ Antibodies |
| ☐ | ☒ Eukaryotic cell lines |
| ☒ | ☐ Palaeontology and archaeology |
| ☐ | ☒ Animals and other organisms |
| ☒ | ☐ Human research participants |
| ☒ | ☐ Clinical data |
| ☒ | ☐ Dual use research of concern |

## Methods

| n/a | Involved in the study |
|---|---|
| ☒ | ☐ ChIP-seq |
| ☒ | ☐ Flow cytometry |
| ☒ | ☐ MRI-based neuroimaging |

## Antibodies

| | |
|---|---|
| Antibodies used | Primary antibodies used for cell samples: anti-Lamin B (sc-6216, RRID:AB_648156), mouse anti-TFAM (MA5-16148, RRID:AB_11157422), and rat anti-alpha Tubulin (MA1-80017, RRID:AB_2210201). All at 1:75 dilution from vendor-supplied stocks. Secondary antibodies (all used at 1:150 dilution, from 50% glycerol stocks prepared by 1:2 dilution of the vendor antibody with 100% glycerol): Alexa 647 anti-Mouse (Jackson ImmunoResearch 715-605-150, RRID:AB_2340862), Cy3 anti-Goat (VWR 102649-368), and Alexa 488 anti-Rat (Jackson ImmunoResearch 712-545-150, RRID:AB_2340683). <br><br>For mouse retina tissues: sheep anti-CHX10 (Exalpha X1180P, RRID:AB_2314191, diluted 1:500,  rabbit anti-PAX6 (Abcam ab195045, RRID: AB_2750924, diluted 1:300), anti-TH antibody (Millipore Cat# AB152, RRID:AB_390204, diluted 1:500) and goat anti-CARTPT antibody (Thermo Fisher Scientific #PA5-47170, RRID: AB_2607700, diluted 1:20). <br><br>Secondary antibodies: <br>Donkey anti-Sheep-Alexa647, Jackson ImmunoResearch #713-605-147,RRID: AB_2340751, diluted 1:500 from 50% glycerol stock) <br>Donkey anti-Rabbit-Cy3, Jackson ImmunoResearch #711-165-152, RRID: AB_2307443, diluted 1:500 from 50% glycerol stock) <br>Donkey anti-Goat Alexa 488 secondary antibody (Jackson ImmunoResearch #705-545-003, RRID:AB_2340428, diluted 1:250) <br>Donkey anti-Rabbit Alexa647 secondary antibody (Jackson ImmunoResearch #711-605-152, RRID:AB_2492288, diluted 1:250) |
| Validation | All antibodies used are from commercial sources as described. <br><br>Anti-Lamin B (sc-6216, RRID:AB_648156): This antibody has been cited in 213 publications, linked at the manufacturer's website. The manufacturer shows several Western blot validations for Lamin B detection in cell lysates from multiple cell types, showing 66 kDa bands, consistent with the size of Lamin B protein. |

Mouse anti-TFAM (MA5-16148, RRID:AB_11157422): According to the manufacturer's website, this antibody was validated by knockdown via siRNA, followed by Western blot.

Rat anti-alpha Tubulin (MA1-80017, RRID:AB_2210201): This antibody has been validated by the manufacturer by Western Blot in many cell types, showing 52 kDa bands corresponding to alpha Tubulin. It has also been validated for IF by the manufacturer and has been in many publications, which are sited on the product page.

Anti-TH antibody (Millipore Cat# AB152, RRID:AB_390204): According to the manufacturer's website, this antibody has been published and validated for use in ELISA, IF, IH, IH(P), IP and WB. They cite 341 publications using this antibody for detection of TH protein on their website: https://www.emdmillipore.com/US/en/product/Anti-Tyrosine-Hydroxylase-Antibody,MM_NF-AB152#anchor_REF.

Anti-CARTPT (Thermo Fisher Scientific #PA5-47170, RRID: AB_2607700): There was not validation for this antibody on the manufacturer's website, but we propose that our work in Figure 5 serves as validation of this antibody, as we showed CARTPT mRNA recovered from the TH+ population transcriptomes, and CARTPT signal by IF in the exact same cells. For further validation, we also performed RNA-FISH for Cartpt and Th in different animals and saw co-localization of CARTPT RNA with TH protein and Th RNA. Validation of this antibody is also supported by prior work from other groups demonstrating co-localization of CARTPT with TH+ amacrine cells in the retina:

S. Anna Sargsyan, P. Michael Iuvone; Cocaine- and amphetamine-regulated transcript (CART): a novel retinal neuropeptide. Invest. Ophthalmol. Vis. Sci. 2014;55(13):2641.

Gustincich S, Contini M, Gariboldi M, et al. Gene discovery in genetically labeled single dopaminergic neurons of the retina. Proc Natl Acad Sci U S A. 2004; 101: 5069–5074.

Sheep anti-CHX10 (Exalpha X1180P, RRID:AB_2314191): On the manufacturer's website, they validate specificity of this antibody to a 46 kDa protein within rat retina tissue lysate and mouse retina tissue lysate and lack of binding within rat and mouse liver. This antibody has been used in many studies where it has been validated through FISH, mouse lines, and RNA sequencing. These publications are listed on the manufacturer's website, but one such publication is:
Shekhar, K. et al. Comprehensive Classification of Retinal Bipolar Neurons by Single-Cell Transcriptomics. Cell 166, 1308-1323.e30 (2016).

Rabbit anti-PAX6 (Abcam ab195045, RRID: AB_2750924): On the manufacturer's website, they validate specificity of this antibody to detect the multiple known variants of PAX6 within mouse eyeball lysates by Western blot. The 47 kDa band is consistent with the full-length PAX6 protein while the 32 and 33 kDa bands are consistent with PAX6p32 and PAX6p33 truncations. They further show image validation in human brains, pancreas, and retina.

## Eukaryotic cell lines

Policy information about cell lines

| | |
|---|---|
| Cell line source(s) | A stable HEK293-GFP cell line (SC001) that constitutively expresses high-levels of GFP under a CMV promoter was purchased from GenTarget Inc. A mouse 3T3 cell line was purchased from ATCC (CRL-1658). |
| Authentication | Both cell lines had morphology and gene expression consistent with what was described for these cell lines in public databases (based on microscopy and RNA-Seq), but a formal authentication was not performed. |
| Mycoplasma contamination | Cell lines were not tested for mycoplasma contamination. |
| Commonly misidentified lines (See ICLAC register) | No commonly misidentified lines were used in this study. |

## Animals and other organisms

Policy information about studies involving animals; ARRIVE guidelines recommended for reporting animal research

| | |
|---|---|
| Laboratory animals | All retina experiments were performed on tissue collected from postnatal day (P) 18 male wild-type CD1 IGS mice (Charles River). Mice were housed at Harvard Medical School at ambient temperature and humidity and a 12-hour alternating light-dark cycle. |
| Wild animals | No wild animals were used in this study. |
| Field-collected samples | No field-collected samples were used in this study. |
| Ethics oversight | All animal experiments were conducted in compliance with protocol IS00001679, approved by the Institutional Animal Care and Use Committee at Harvard University. |

Note that full information on the approval of the study protocol must also be provided in the manuscript.

