## [Peer Review File · Nature Methods]

Peer Review Information

Journal: *Nature Methods*

Manuscript Title: Light-Seq: Light-directed in situ barcoding of biomolecules in fixed cells and tissues for spatially indexed sequencing

Corresponding author name(s): Jocelyn Y. Kishi, Constance L. Cepko, Sinem K. Saka, Peng Yin

Editorial Notes:

Redactions –
unpublished data

Reviewer Comments & Decisions:

Decision Letter, initial version:
--

Dear Sinem,

Thank you for submitting your manuscript entitled "Light-Seq: Light-directed in situ barcoding of biomolecules in fixed cells and tissues for spatially indexed sequencing". We have given the paper our careful consideration but we regret that we cannot publish it in *Nature Methods* in its current form.

We read your paper with interest and think it represents a potentially important step forward for spatial 'omics experiments. However, while we liked the demonstration on the retina, we thought that to succeed in peer review you would have to demonstrate single-cell resolution scRNA-seq on a tissue sample.

Should future experimental data allow you to address this concern, we would be happy to look at a revised manuscript (unless, of course, something similar has by then been accepted at *Nature Methods* or appeared elsewhere). This includes submission or publication of a portion of this work somewhere else. In the case of eventual publication, the received date would be that of the revised paper.

Of course, a revised version could be an Article, rather than a Brief Communication.

If you are interested in submitting a suitably revised manuscript in the future or if you have any questions, please contact me.

Thank you for your interest in Nature Methods. I am sorry that on this occasion we cannot be more positive.

Sincerely,
Rita

Rita Strack, Ph.D.
Senior Editor
Nature Methods

Author Rebuttal to Initial comments

Dear Rita,

Thank you for the opportunity to have our updated version reconsidered by you and the editorial team. Please find our updated manuscript, which includes a new demonstration (Figure 2, Supplementary Fig. 7, Supplementary Table 3 and Supplementary Data Tables 2-3) where we apply Light-Seq for light-directed in situ barcoding and RNA-sequencing of rare, disjointed cells (as few as 4 cells per tissue section replicate) in intact tissues, identified based on a protein biomarker specifically expressed in this very rare dopaminergic amacrine cell subtype of mouse retina, which has been challenging to study with single-cell approaches. For comparison, we have also differentially barcoded the neighbouring non-dopaminergic amacrine cells in the same section and analysed the differential gene expression for the target cell subtype versus other amacrine cells. This experiment confirmed the known markers for these cells and also yielded previously uncharacterized markers, which we validated by secondary immunofluorescence or multiplexed SABER-FISH experiments.

We hope that with these new experiments and datasets we can convincingly demonstrate a real-life application of the method which underlines its unique suitability for such challenging use cases, and address the editorial board's previous comments. We are very happy with the performance and sensitivity of the method for barcoding of such few and scattered cells across whole tissue sections and appreciate your feedback, which we believe made the manuscript a lot stronger. Please let us know your thoughts and further comments.

The full set of files are accessible here:

<https://www.dropbox.com/sh/c52to7skrsdztrs/AABqBXsQ2BSUngezAUSaM9DHa?dl=0>

Thank you,

Sinem

Decision Letter, first revision:

Dear Sinem,

Your Brief Communication, "Light-Seq: Light-directed in situ barcoding of biomolecules in fixed cells and tissues for spatially indexed sequencing", has now been seen by three reviewers. As you will see from their comments below, although the reviewers find your work of considerable potential interest, they have raised a number of concerns. We are interested in the possibility of publishing your paper in Nature Methods, but would like to consider your response to these concerns before we reach a final decision on publication.

We therefore invite you to revise your manuscript to address these concerns. When you do, we ask that you expand the paper to an Article (up to 3,000 words and 6 display items), which we hope will give you the additional space to place your work in the appropriate context in terms of referencing, as mentioned by referee 1. Referee 2 warns against citing too many preprints. While we are fine with citing preprints, we do ask that you do so judiciously if you are doing so to support claims made in your paper that lack alternative evidence.

In terms of experiments, we ask that you provide the additional validation experiments requested by the referees to convincingly show the robustness of the approach.

- * include a point-by-point response to the reviewers and to any editorial suggestions
- * please underline/highlight any additions to the text or areas with other significant changes to facilitate review of the revised manuscript
- * address the points listed described below to conform to our open science requirements
- * ensure it complies with our general format requirements as set out in our guide to authors at www.nature.com/naturemethods

* resubmit all the necessary files electronically by using the link below to access your home page

[Redacted] This URL links to your confidential home page and associated information about manuscripts you may have submitted, or that you are reviewing for us. If you wish to forward this email to co-authors, please delete the link to your homepage.

We hope to receive your revised paper within XX weeks [**ED TO CUSTOMIZE AS NEEDED**]. If you cannot send it within this time, please let us know. In this event, we will still be happy to reconsider your paper at a later date so long as nothing similar has been accepted for publication at Nature Methods or published elsewhere.

OPEN SCIENCE REQUIREMENTS

REPORTING SUMMARY AND EDITORIAL POLICY CHECKLISTS

IMAGE INTEGRITY

DATA AVAILABILITY

All novel DNA and RNA sequencing data, protein sequences, genetic polymorphisms, linked genotype and phenotype data, gene expression data, macromolecular structures, and proteomics data must be deposited in a publicly accessible database, and accession codes and associated hyperlinks must be provided in the “Data Availability” section.

Please include a “Data availability” subsection in the Online Methods. This section should inform readers about the availability of the data used to support the conclusions of your study, including accession codes to public repositories, references to source data that may be published alongside the paper, unique identifiers such as URLs to data repository entries, or data set DOIs, and any other statement about data availability. At a minimum, you should include the following statement: “The data that support the findings of this study are available from the corresponding author upon request”, describing which data is available upon request and mentioning any restrictions on availability. If DOIs are provided, please include these in the Reference list (authors, title, publisher (repository name), identifier, year). For more guidance on how to write this section please see: <http://www.nature.com/authors/policies/data/data-availability-statements-data-citations.pdf>

CODE AVAILABILITY

Please include a “Code Availability” subsection in the Online Methods which details how your custom code is made available. Only in rare cases (where code is not central to the main conclusions of the paper) is the statement “available upon request” allowed (and reasons should be specified).

For more information on our code sharing policy and requirements, please see: <https://www.nature.com/nature-research/editorial-policies/reporting-standards#availability-of-computer-code>

MATERIALS AVAILABILITY

SUPPLEMENTARY PROTOCOL

To help facilitate reproducibility and uptake of your method, we ask you to prepare a step-by-step Supplementary Protocol for the method described in this paper. We [encourage authors to share their step-by-step experimental protocols](https://www.nature.com/nature-research/editorial-policies/reporting-standards#protocols) on a protocol sharing platform of their choice and report the protocol DOI in the reference list. Nature Research's Protocol Exchange is a free-to-use and open resource for protocols; protocols deposited in Protocol Exchange are citable and can be linked from the published article. More details can found at www.nature.com/protocolexchange/about.

ORCID

Sincerely,
Rita

Rita Strack, Ph.D.
Senior Editor
Nature Methods

Reviewers' Comments:

Reviewer #1:

Remarks to the Author:

The authors have described an interesting method for ex situ RNA seq while preserving spatial identity of cells in a fixed biological sample through one-step photocrosslinking of barcoding guide primers that permit barcodes to be added to the in situ transcribed cDNA. Potential advantages of using this method are cost-effectiveness, preservation of tissue context for transcriptomics, dispensable upstream processing of samples like dissociation of cells and sorting, to name a few. The authors use this approach to assess the partial transcriptomes from various brain and retina cellular regions including groups of identified TH+ cells. While the methodology is potentially useful, there are concerns that the authors should address to increase confidence that the technology is working as envisioned:

Comments:

1. To know the sensitivity of the methodology, it is important to know the efficiency of the barcode attachment reaction. One way to do this would be to take 10 different in vitro transcribed RNAs and to mix them in various ratios and abundances and then perform the reaction procedure and assess the resultant ratio of products. This ideally would be performed on RNAs that are immobilized such as on beads, embedded in agar or on nitrocellulose.
2. Likewise, it is important to know the efficiency of the “switch RT” reaction (what % of queried mRNAs are detected). This is particularly difficult but may be optimizable if the original efficiency can be assessed.
3. With regard to the switch RT reaction, is there sequencing evidence for “jumping” of the RT to other mRNAs at this step? This happens in standard RT reactions and likely would happen more frequently for this reaction. Does this confound the interpretation of results?
4. There seems to be wide variation in the number of mapped sequenced reads from various samples. For example, on lines 140-143, 1959 (HEK cells) and 1170 (3T3) UMIs were reported per unit area. What does unit area mean? Is this per irradiated unit (multiple cells) or averaged to be per single cell? This is said to be subsaturating even though it was from 30 million reads, and that it almost doubles when going to 200 million reads. This suggests that there is a large amount of reads that didn't map to the genome (what percentage of the reads did map?) and that there are few reads/cell that can be detected. More clarification of this would be helpful.
5. The TH cell data appears to have many more UMIs (Supplementary Table 3, 8,428-to 10,446) than other cells or tissue areas that were examined in the study. Discussion of what accounts for these differences would be helpful in insuring that the technology is robust and working as envisioned.

6. The authors only present exon data as their output, yet given the experimental procedure where whole cells are irradiated there must also be nuclear intron containing RNA detected in their sequencing runs. It would be informative to know how much nuclear RNA vs cytoplasmic RNA is present in the sample. Indeed, it might be that the nuclear RNA would be a relatively high fraction of the reads since it is possible that the nuclear RNA is less “fixed” than cytoplasmic RNA. Or alternatively there may be less due to less accessibility of the reagents. The authors should report these numbers and discuss the implications.

7. Have the authors explored methods for removing RNA binding proteins or RNA structure to increase the accessibility of the RNA to the reaction reagents including reverse transcriptase? Such data may expand the usefulness of the methodology to other types of fixatives or differentially prepared tissue sections especially pathological tissue specimens.

8. ROI dimensions and the number of cells in an ROI is estimated by the size of ROI is unclear. Authors should add this number to their respective figure legends.

9. While the barcoding strategy is light directed and can be specific, authors should show/comment/discuss about the resolving power of light-seq barcoding to distinguish between adjacent cells.

10. Spatial transcriptomics is advancing fast with newer assays and hence a close comparison of this method with other spatial methods (10X visium, APEX-seq, slide-seq) becomes important. The Discussion section should be expanded to present a more detailed comparison of the data (not cost analysis) generated with this method as compared with other currently used spatial transcriptomic assays.

11. Referencing in the paper was haphazard. There are many background papers that should be referenced as the particular subtechniques were previously described. For example, the in situ cDNA synthesis step should reference Tecott et al. “In Situ Transcription: Specific synthesis of cDNA in fixed tissue sections.” Science 1988. There are many other examples as well and the authors are encouraged to be more appropriate and thorough in their referencing.

12. There is a wealth of useful supplemental data but it is not well integrated into the body of the manuscript. While it is referenced, much of the supplemental data is not discussed. Also, if the data isn't directly relevant to the manuscript then it should be removed, e.g. the cost analysis table is superfluous and not representative of actual costs since most investigators get institutional or quantity discounts making the price comparisons unrealistic.

Reviewer #2:

Remarks to the Author:

In this manuscript, Kishi et al. describe Light-Seq, a novel technique for in situ barcoding of cDNA followed by next-generation sequencing that allows for the characterization of transcriptomes from a defined group of cells or a single rare cell type. They provide excellent examples for the specificity and sensitivity of Light-Seq, sequencing transcriptomes from both cells in culture and fixed tissue sections. The ability of Light-Seq to resolve the transcriptome of a very rare cell type in the mouse retina is impressive. This kind of information is often not available from single-cell RNAseq experiments. The relative simplicity and affordability of the Light-Seq method are also impressive.

This manuscript is well written, the data presented is of high quality, and the experiments are well controlled. The authors provide more than sufficient evidence to support the applicability and the unique properties of their method.

I add only one cautionary note. The authors cite several papers posted on BioRxiv, which have not yet been peer-reviewed. I would suggest refraining from citing non-peer-reviewed papers.

Reviewer #3:

Remarks to the Author:

The Light_Seq technology is quite interesting and can be useful to the applications to study rare cell populations as demonstrated by the manuscript. Due to the limited throughput, it may not be able to be applied to study a large number of ROIs with high spatial resolution. In general, I am optimistic about the potential and future technology improvements that can be made. I do have several concerns regarding the quality of the current experimental design. More pieces of evidence are needed to address these concerns.

1. RNase H based releasing of cDNAs may cause Gene length and detection bias. As shorter cDNAs (shorter genes) may be released much faster and efficiently compared with long cDNAs. Gene length bias analysis is needed here to understand the releasing gene profile. Besides, RNase H release may not be equally efficient in different tissue types, thus, more experiments with broader tissue types are needed here to demonstrate the advantage of this releasing process. It is also necessary to compare cDNA product quality directly with the enzymatic (Proteinase K) digestion-based method.

2. The manuscript did not provide enough validation for the possible outside ROI background labeling due to light scattering. In SI figure 2, there are strong Cy3 background signals remaining (outside ROI) after removing the non-crosslinked strands. Although this could be caused by insufficient washing, it may also be caused by light scattering-induced crosslinking. It also raised another question, what is exactly the highest spatial resolution (with enough Signal to noise ratio) Light-seq can reach?

3. It is necessary to demonstrate a higher number (>10) barcoding rounds experiment. The current manuscript only demonstrated 3 barcoding rounds which are not enough to show the robustness and consistency of this technique. 10 or 20 rounds of barcoding experiments are needed here to demonstrate the robustness of the whole experimental design, since repeated washing and light exposure may cause unpredictable tissue deformation at some stage.

Minor comments:

4. The photomasks and alignment to ROI are vitally important to the current experimental design, but there are not enough details in the experimental section. It may also be hard to address one entire single cell in a very dense tissue section when cell boundary information is missing or hard to collect.

Author Rebuttal, first revision:

[R1.0] We thank the reviewer for their kind words and careful review. We address each of the points individually below.

[C1.1]

1. To know the sensitivity of the methodology, it is important to know the efficiency of the barcode attachment reaction. One way to do this would be to take 10 different *in vitro* transcribed RNAs and to mix them in various ratios and abundances and then perform the reaction procedure and assess the resultant ratio of products. This ideally would be performed on RNAs that are immobilized such as on beads, embedded in agar or on nitrocellulose.
2. Likewise, it is important to know the efficiency of the “switch RT” reaction (what % of queried mRNAs are detected). This is particularly difficult but may be optimizable if the original efficiency can be assessed.

[R1.1] We agree with the reviewer that the barcode attachment reaction is one of several important steps that could limit the sensitivity of Light-Seq, including RT efficiency, polyadenylation efficiency, photo- crosslinking for barcode attachment, cDNA extraction, cross-junction synthesis efficiency, and associated background for each step, all of which are impacted by *in situ* conditions. In light of the reviewer's suggestion, we performed quantitative comparison to smFISH data, as the current gold

standard for RNA quantification *in situ*, to directly assess the cumulative sensitivity of the method in the *in situ* environment.

We previously developed the SABER-FISH method for amplifying smFISH signals and demonstrated very high sensitivity and specificity of individual RNA transcript detection, including in retina samples specifically¹. We have now added additional quantitative analysis comparing our detection efficiency to smFISH of several targets in previously performed for the same cell types, within the same sample type². Together, we estimate Light-Seq sensitivity to be $4.29 \pm 3.39\%$ (mean \pm std, $n=16$ genes, 4 replicates) and Drop-Seq $3.97 \pm 4.38\%$ ($n=16$ genes, 6 replicates) for these genes. We made several additions to the manuscript to emphasize the sensitivity of Light-Seq. We added sensitivity analysis in the new **Fig. 4e** and discussion of this comparison in the main text (Line 239-247), directly comparing gene detection for Light-Seq and Drop-Seq for a set of RNA markers benchmarked by smFISH. We also added gene-specific sensitivity estimations in **Extended Data Fig. 4h-j** and analysis for reads from genes of different lengths **Extended Data Fig. 5, 7**.

We acknowledge that this is an imperfect comparison because our degenerate RT primers (different from polyT priming used in Drop-Seq) could theoretically yield multiple reads on a single long RNA molecule (see response **[R3.1]** below for detailed description of length bias analysis). To clarify this, we have included a per-gene analysis for 16 RNAs measured by smFISH in the supplement, to show similar Light-Seq sensitivity compared to Drop-Seq for genes of different lengths (**Extended Data Fig. 5e-j, 7b-c**)

Regarding the photo-crosslinking efficiency itself, previous works have shown that the efficiency of 1 sec photo-crosslinking under a 365 nm wavelength of light is 90-97% *in vitro*³. To achieve a high level of photo-crosslinking *in situ* we empirically optimized the

Response to Reviewers

We thank the reviewers for their careful and thoughtful feedback on our work. To fully address your comments and upon editorial recommendation, we have revised the manuscript into the longer Article format, hence the previous figure numbering and the text layout changed substantially, although the fundamental content and data is not altered. Following the reviewer suggestions, we: (1) included additional experimental data and analyses that have been added to the as new figures and figure panels, (2) shared a preview of our ongoing work that will be incorporated into future publications in the response letter, (3) re-structured the supplementary information to have it more streamlined and comprehensive, (4) prepared detailed supplemental notes for detailing the experimental design considerations and implementation recommendations. We provide a list of these changes below, followed by inline responses to each specific comment. Together we believe these changes address the comments, and we're grateful for your feedback to improve the overall quality of the publication.

Reviewer 1

[C1.0] The authors have described an interesting method for ex situ RNA seq while preserving spatial identity of cells in a fixed biological sample through one-step photocross-linking of barcoding guide primers that permit barcodes to be added to the in situ transcribed cDNA. Potential advantages of using this method are cost-effectiveness, preservation of tissue context for transcriptomics, dispensable upstream processing of samples like dissociation of cells and sorting, to name a few. The authors use this approach to assess the partial transcriptomes from various brain and retina cellular regions including groups of identified TH⁺ cells. While the methodology is potentially useful, there are concerns that the authors should address to increase confidence that the technology is working as envisioned:

crosslinking efficiency for our imaging system and found that 10% 365 nm LED power output for 10-20s exposure was optimal for our system to get the best inside-ROI to out-of-ROI signal ratio. Of note, these parameters will depend on the optical setup (light source, light path, objective, etc.) and should be optimized for new optical setups. We have added user-friendly instructions for how to optimize these parameters in the new **Supplementary Note 4** and in protocols.io to assist future Light-Seq users in optimizing for their own setups.

[C1.2]

3. With regard to the switch RT reaction, is there sequencing evidence for “jumping” of the RT to other mRNAs at this step? This happens in standard RT reactions and likely would happen more frequently for this reaction. Does this confound the interpretation of results?

[R1.2] This is an excellent question and one that we’ve thought about deeply ourselves. We intentionally made several design choices to reduce the potential for this or related effects to lead to sequenceable reads, which we now outline extensively in a new **Supplementary Note 1**. In this note, we explain why we do not expect to see either of these background reads in our sequencing results and share the additional analysis we performed on our paired end reads to demonstrate that we do not see evidence of chimeric RNA reads in our sequencing results. To our main text, we added a sentence (Line 141-143) pointing to this new **Supplementary Note 1**) and added citations to several important papers that have

used and/or studied the template-switching behavior of reverse transcriptases, both to designated template switching oligos as well as to other RNA sequences.

[C1.3]

4. (a) There seems to be wide variation in the number of mapped sequenced reads from various samples. For example, on lines 140-143, 1959 (HEK cells) and 1170 (3T3) UMIs were reported per unit area. What does unit area mean? Is this per irradiated unit (multiple cells) or averaged to be per single cell?

[R1.3] We thank the reviewer for this point. Because the barcoding area of Light-Seq can be arbitrarily set by the user, we chose to normalize the number of transcripts that can be captured with Light-Seq as the number of unique molecular identifier (UMI) sequences per “unit area” that was roughly the size of a bead in Slide-Seq⁴ or a barcoded square in DBIT-Seq⁵, which we define as 10 μm x 10 μm . This metric was commonly used for reporting sensitivity and comparing spatial transcriptomics methods. Number of UMIs reported per unit area depend on the area of irradiation, and since cell shapes, sizes, cytoplasmic volumes, and RNA content are quite heterogeneous across cell types, it is expected that the read counts would not match across different cell types. For cultured cells, the unit area only includes segmented cells (not the surrounding empty area) and for tissues, the unit area only includes the outlined cells or layers and is not averaged per cell. We have now added this information to the in the main text (Lines 159-166) and clarified how the unit area was calculated in the legend of the new **Supplementary Table 1**.

We note, however, that this is not a good general metric for sensitivity, as RNA density is known to vary greatly with different cell types and 3D volumes⁶. Consistent with this, we observe wide variation in UMIs/unit area across cell types and in general, we observe that cells with larger cytoplasmic volumes correlate with higher UMIs/unit area (now shown in

Extended Data Fig. 3d and highlighted in the corresponding main text – Lines 209-214). In the retina, for example, the GCL has the largest cell volumes and cytoplasmic areas and the highest UMIs/unit area, while the ONL has the smallest cell/cytoplasm volumes and the smallest UMI/unit area. The discrepancy in UMIs per unit area for HEK versus 3T3 cells similarly correlates with differences in cytoplasmic volume per unit area. Although the same number of HEK and 3T3 cells were barcoded, the total barcoded cellular area for 3T3 cells is 2.3- 3-fold larger than the HEK cell area (**Supplementary Table 1**). This has now been emphasized in the table legend.

[C1.4]

4. (b) This is said to be subsaturating even though it was from 30 million reads, and that it almost doubles when going to 200 million reads. This suggests that there is a large amount of reads that didn't map to the genome (what percentage of the reads did map?) and that there are few reads/cell that can be detected. More clarification of this would be helpful.

[R1.4] As the reviewer observed, a significant number of reads didn't map uniquely to the genome. We now include an overview of our sequence processing pipeline (**Extended Data Fig. 4a**) and sunburst

plots depicting what fraction of reads are filtered out at which steps and why, including detailed methods about the pipeline (**Extended Data Fig. 4b**). As expected from our internal priming design, the majority of our reads are multimapping (e.g. rRNA). Future implementations of the technology could include rRNA depletion, as we now highlight in the Discussion (Lines 328-334), which would increase the proportion of uniquely mapped sequencing reads.

[C1.5]

5. The TH cell data appears to have many more UMIs (Supplementary Table 3, 8,428-to 10,446) than other cells or tissue areas that were examined in the study. Discussion of what accounts for these differences would be helpful in insuring that the technology is robust and working as envisioned.

[R1.5] This is a great observation and one that we have changed our text to better highlight (see also response **[R1.3]** above). We believe that the increase in UMIs for TH+ amacrine cells relative to other retinal cells and/or cells in culture is expected and likely driven by biological differences rather than technical ones. The TH+ cells are some of the biggest cells in the retina and have very large cytoplasmic areas. **Figure 5d** shows just how dramatic the cytoplasm difference is between these cells compared to the adjacent amacrine cells. We see similarly high UMIs per unit area in the retinal ganglion cells (RGCs) compared to the other cell types in the retina (**Extended Data Fig. 3 and [R1.3]**). RGCs, like TH+ amacrine cells, are large and sprawling in morphology and harbor long axons that reach into the brain, which is consistent with the idea that potentially these larger cells have more RNA than their smaller neighboring cell types. We now also emphasize in the main text (Lines 324-326) that this is a major caveat to interpreting the metric of “UMIs per unit area” for comparison across technologies for spatial transcriptomics, since there is large variability in these numbers across cell types or even across regions measured within a single cell.

[C1.6]

6. The authors only present exon data as their output, yet given the experimental procedure where whole cells are irradiated there must also be nuclear intron containing RNA detected in their sequencing runs. It would be informative to know how much nuclear RNA vs cytoplasmic RNA is present in the sample. Indeed, it might be that the nuclear RNA would be a relatively high fraction of the reads since it is possible that the nuclear RNA is less “fixed” than cytoplasmic RNA. Or alternatively there may be less due to less accessibility of the reagents. The authors should report these numbers and discuss the implications.

[R1.6] This is a great question, and the reviewer is correct that our current primary pipeline only maps to exons based on the annotated genome file for the appropriate genome (see new **Extended Data Fig. 4a-b**). We agree that the efficiency for recovering nuclear vs. cytoplasmic RNA would likely depend on the fixed *in situ* environment and accessibility. In light of this question, we have since performed mapping to full gene bodies and analysis of intronic reads, and have added details about this to our main text (Lines 257-260), supplement (**Supplementary Note 3**), Methods (**Intron analysis** section), and code base on Github. In brief, we do see substantial amounts of intronic reads in our sequencing results (>20% in all replicates) when we include them in the mapping, indicating we are able to successfully extract nuclear RNA at ratios consistent with similar methods using internal RNA priming⁷. As a result

of internal priming, we also see good representation of noncoding RNAs (**Extended Data Table 1**). Further optimization for fixation, permeabilization, and extraction could likely further improve the accessibility of nuclear RNA or change the ratios of RNA species recovered (see **[R1.7]** below).

[C1.7]

7. Have the authors explored methods for removing RNA binding proteins or RNA structure to increase the accessibility of the RNA to the reaction reagents including reverse transcriptase? Such data may expand the usefulness of the methodology to other types of fixatives or differentially prepared tissue sections especially pathological tissue specimens.

[R1.7] [Redacted]

[C1.8]

8. ROI dimensions and the number of cells in an ROI is estimated by the size of ROI is unclear. Authors should add this number to their respective figure legends.

[R1.8] We thank the reviewer for pointing this out, we made several changes to the figure legends to incorporate this information and also include more detailed information in our **Methods** section for how ROI areas and cell numbers were estimated. For the cell mixing experiments, we added the cell counts (~25 cells of each type) to the **Figure 3** legend (see **Extended Data Fig. 2** for masks), and ROI sizes are included in **Supplementary Table 1**. For the mouse retina experiments we have now added cell number estimations in the main text and methods, **Supplementary Table 3**, in addition to the **Figure 4** legend.

[C1.9]

9. While the barcoding strategy is light directed and can be specific, authors should show/comment/ discuss about the resolving power of light-seq barcoding to distinguish between adjacent cells.

[R1.9] Light-Seq uses 365 nm - 405 nm light for covalent barcode crosslinking, providing the basis for theoretically diffraction-limited barcoding. This could enable even subcellular barcoding, as demonstrated by the subcellular barcoding using laser-scanning in **Extended Data Fig. 1**. However, other practical considerations may affect the final resolving power. We have added a new panel to **Extended Data Fig. 1** showing the minimum feature size we can achieve using DMD illumination and the signal decay. We also included a line

scan showing fluorescence signal across and outside the ROI boundary for the rare amacrine cell type experiment (**Extended Data Fig. 6**). We now emphasized in the main text (Lines 179-189 and 300-305) how light-scattering and other sources of background can be introduced and mitigated. We further added a new section (**Supplementary Note 4**) with instructions for calibrating and optimizing the illumination, precisely focusing the illumination in XYZ, optimizing the illumination, measuring the sensitivity, drawing ROIs, and designing experiments to take into account light scattering. We also updated our protocols.io with these instructions.

[C1.10]

10. Spatial transcriptomics is advancing fast with newer assays and hence a close comparison of this method with other spatial methods (10X Visium, APEX-seq, slide-seq) becomes important. The Discussion section should be expanded to present a more detailed comparison of the data (not cost analysis) generated with this method as compared with other currently used spatial transcriptomic assays.

[R1.10] We appreciate this feedback. We have rewritten our Discussion to better compare the data to existing spatial transcriptomic assays and clarify key differences in workflow, flexibility, data types and sensitivity. We chose a conservative approach for our sequence analysis pipeline by filtering out reads

that multimap to the genome or to the transcriptome, and only considering exonic maps. This means that the true number of unique molecules we are actually labeling is much higher (see **Extended Data Figure 4a** for breakdown of sequence filtering). We compare these conservative estimates to smFISH data for 16 different markers (see **[R1.1]** above) to estimate sensitivity with comparison to a common standard. The Discussion section now contains a comparison of these sensitivity measurements to other spatial methods (Lines 315-326, 356-362, 387-392). We refrained from making a more direct/side-by-side comparison to alternative methods, because the quantitative metrics for some of the other methods are either not available or not analyzed in a transparent manner for us to evaluate direct comparisons. These could also vary significantly from tissue type to tissue type.

The low cost and convenience of the method is a critical advantage that makes Light-Seq accessible to scientists around the world, and we updated the cost estimates to better reflect non-promotional pricing (see also **[R1.12]** below) and added a caution note on potential price variations.

[C1.11]

11. Referencing in the paper was haphazard. There are many background papers that should be referenced as the particular subtechniques were previously described. For example, the in situ cDNA synthesis step should reference Tecott et al. "In Situ Transcription: Specific synthesis of cDNA in fixed tissue sections." Science 1988. There are many other examples as well and the authors are encouraged to be more appropriate and thorough in their referencing.

[R1.11] We thank the reviewer for pointing out this paper and have included it in our references. We also added additional references related to the cDNA synthesis, non-templated, and templated switching behavior of MMLV-type reverse transcriptases⁸⁻¹⁰ (see

[R1.3] above), and for other methods like PIC¹¹ and smFISH¹²), and RT optimizations^{13,14}. We welcome further suggestions for additional references that should be included.

[C1.12] There is a wealth of useful supplemental data but it is not well integrated into the body of the manuscript. While it is referenced, much of the supplemental data is not discussed. Also, if the data isn't directly relevant to the manuscript then it should be removed, e.g. the cost analysis table is superfluous and not representative of actual costs since most investigators get institutional or quantity discounts making the price comparisons unrealistic.

[R1.12] Thank you for this feedback. We have completely reworked the main text and switched to the Nature Methods Article format, allowing us to include more text, main figures and comprehensive references to our supplementary material. We also added a wealth of new data and analyses to the supplementary materials and re-formatted them to be more streamlined (in the form of Extended Data and Supplemental Information as the journal recommends).

The relatively low cost of the method is a huge part of making Light-Seq accessible to many academic labs across the world and this is important data for us to have and share with prospective users of the method. However, we agree that academic discounts and old price estimates before recent inflation could have unfairly lowered the cost estimate, so we updated the cost estimate table with non-discounted

publicly available list prices for all the reagents this was available for. After incorporating these updated commercial list prices, which combined account for the majority (~80%) of the total cost, our estimate went from \$30 to \$34.50 (potential pricing variations and cautions are now highlighted in **Supplementary Table 5**).

Reviewer 2

[C2.0] In this manuscript, Kishi et al. describe Light-Seq, a novel technique for in situ barcoding of cDNA followed by next-generation sequencing that allows for the characterization of transcriptomes from a defined group of cells or a single rare cell type. They provide excellent examples for the specificity and sensitivity of Light-Seq, sequencing transcriptomes from both cells in culture and fixed tissue sections. The ability of Light-Seq to resolve the transcriptome of a very rare cell type in the mouse retina is impressive. This kind of information is often not available from single-cell RNAseq experiments. The relative simplicity and affordability of the Light-Seq method are also impressive.

This manuscript is well written, the data presented is of high quality, and the experiments are well controlled. The authors provide more than sufficient evidence to support the applicability and the unique properties of their method.

I add only one cautionary note. The authors cite several papers posted on BioRxiv, which have not yet been peer-reviewed. I would suggest refraining from citing non-peer-reviewed papers.

[R2.0] We thank the reviewer for their kind words and careful review.

We have updated the citations for preprints that have now been published (original references 9, 14, 15, 27, 38) and we are hopeful that the few remaining preprint references (which we believe are important to cite considering the current dynamism of the field) will be published by the time this manuscript is finalized.

Reviewer 3

[C3.0] The Light_Seq technology is quite interesting and can be useful to the applications to study rare cell populations as demonstrated by the manuscript. Due to the limited throughput, it may not be able to be applied to study a large number of ROIs with high spatial resolution. In general, I am optimistic about the potential and future technology improvements that can be made. I do have several concerns regarding the quality of the current experimental design. More pieces of evidence are needed to address these concerns.

[R3.0] We thank the reviewer for their enthusiasm and careful review. Please see our answers to the specific concerns below.

[C3.1]

1. RNase H based releasing of cDNAs may cause Gene length and detection bias. As shorter cDNAs (shorter genes) may be released much faster and efficiently compared with long cDNAs. Gene length bias analysis is needed here to understand the releasing gene profile. Besides, RNase H release may not be equally efficient in different tissue types, thus, more experiments with broader tissue types are needed here to demonstrate the advantage of this releasing process. It is also necessary to compare cDNA product quality directly with the enzymatic (Proteinase K) digestion-based method.

[R3.1] These are very good points. We prefer the RNase H strategy as it offers high specificity in releasing cDNAs under mild conditions, but we agree that both the in situ RT with internal priming and RNase H release could create gene length-dependent biases. To assess this better, we have performed additional analysis on gene length bias as well as gene body coverage. We also additionally performed analysis on intronic reads to show we are able

to successfully extract nuclear sequences, see response [R1.6] above. These analyses are described in new supplementary material (**Extended Data Figures 5 and 7**, and **Supplementary Note 3**) and are referenced in the main text. The Methods section and code have also been updated to describe the analyses. In brief, we see good gene body coverage without significant 5' or 3' bias, as well consistent RPKM values across transcripts of varying length, as expected from the internal reverse transcription (RT) priming strategy that we use.

Regarding the Proteinase K treatment, in the current work we prioritized developing a workflow that preserved the integrity of proteins and morphology of the tissue but we agree that future modifications to the method such as alternative fixatives and protease digestion could help improve sequence accessibility and extraction for different tissue preparations, at the potential cost of disrupting morphology and protein epitopes for downstream analysis. We also have several additional tissue types that we have successfully performed Light-Seq on with further optimizations along this line, but those belong to separate follow-up publications that will be focused on the biological findings. While the current unmodified protocol has worked for several other tissues in our collaborator's hands, some applications such as FFPE preparations may require modifications. We have observed that optimal fixation and tissue preparation parameters do vary by tissue type, but that standard tissue-specific protocols that are compatible with RNA detection (for example FISH) tend to also work well for Light-Seq. We have added a note to protocols.io with suggestions for applying Light-Seq to different tissue

types. For some applications where preservation of proteins is not high priority, other harsher methods of extraction could in theory be applied. We have also added a note about this in protocols.io for future users.

Please refer to response [R1.7] above for additional information and a preview of other tissue data. We also added a sentence to the Discussion (Lines 328-334) about the potential application of proteases, antigen retrieval, and additional optimizations.

[C3.2]

2. The manuscript did not provide enough validation for the possible outside ROI background labeling due to light scattering. In SI figure 2, there are strong Cy3 background signals remaining (outside ROI) after removing the non-crosslinked strands. Although this could be caused by insufficient washing, it may also be caused by light scattering-induced crosslinking. It also raised another question, what is exactly the highest spatial resolution (with enough Signal to noise ratio) Light-seq can reach?

[R3.2] As the reviewer correctly notes, the resolution of Light-Seq barcoding is subject to background by light-scattering *in situ*, which does blur the ROI boundaries by ~1-3 μm with the DMD optical setup (see new **Extended Data Figures 1 and 6**, where we included linescans showing the fluorescence signal across and outside the ROI boundary and gives resolution estimates for subcellular labeling and for the rare amacrine cell type experiment). This light-scattering can be accounted for during ROI selection by eroding the ROI boundaries by a fixed distance, as demonstrated in **Extended Data Fig. 6** for the barcoding experiment described in **Fig. 5**. Please also see [R1.9] above for detailed descriptions of additions to the text, supplement, and protocols to address the light scattering and resolution question, as well as other potential sources of background.

[C3.3]

3. It is necessary to demonstrate a higher number (>10) barcoding rounds experiment. The current manuscript only demonstrated 3 barcoding rounds which are not enough to show the robustness and consistency of this technique. 10 or 20 rounds of barcoding experiments are needed here to demonstrate the robustness of the whole experimental design, since repeated washing and light exposure may cause unpredictable tissue deformation at some stage.

[R3.3] In this work we sought to develop a simple and accessible method for labs to profile a few different populations of cells that might otherwise be extremely difficult to achieve with existing methods. Three rounds of barcoding can comfortably fit within one working day with a single person preparing buffers and performing all (manual) pipetting operations. We also note in our online protocols.io at which steps the protocol can be paused, so that no protocols need to exceed a standard 8 hour day / 5 days a week schedule. While more barcoding rounds would be achievable manually, we strongly believe that a fully automated fluidics solution would be important for scaling up, which would in turn make the method less accessible and more tedious to implement for the general biology lab.

Scaling up the barcoding is of great interest to us for several future applications, but scaling will be the focus of future work.

Our cited previous publications, SABER-FISH¹ and Immuno-SABER¹⁵, which use very similar wash conditions were successfully applied for multiple (up to 7) rounds of iterative formamide-based washing, hybridization and imaging. Other published methods (e.g. CycIF¹⁶) perform much longer (up to 60) sequential workflows with repeated labeling, imaging, and photobleaching, and provide strong preceding evidence to expect good tissue preservation across more barcoding rounds. We now mention this also in the Discussion (Lines 389-395).

Minor comments:

[C3.4]

4. The photomasks and alignment to ROI are vitally important to the current experimental design, but there are not enough details in the experimental section. It may also be hard to address one entire single cell in a very dense tissue section when cell boundary information is missing or hard to collect.

[R3.4] Thank you for this feedback. We have included a new section (**Supplementary Note 4**) and our protocols.io now contains better instructions for aligning and calibrating the DMD field of view, focusing light in 3D on desired cells, and accommodating for light scattering by drawing ROIs slightly smaller than the desired sequencing boundaries (see also responses **[R1.9]** and **[R3.2]** above). This section also includes screenshots of the microscope software we use to calibrate the illumination and draw ROIs onto our samples (a couple examples shown below). We use a commercial DMD attachment on our microscope which comes with software for calibrating the illumination and drawing ROIs (see screenshots below) directly on an image of the sample while it sits on the microscope.

When calibration is done properly, the illumination is automatically aligned with the ROI. Other commercial microscopes have similar ROI selection and illumination tools and should not require additional alignment, as long as the sample remains on the microscope for pre-barcode imaging (e.g. of the antibody stain or bright field capture to identify cells of interest) and barcoding.

Calibration software on Nikon microscope.

Screenshot from ROI selection for cell mixing experiment (GFP overlaid on Bright-field).

Light-Seq offers the opportunity to customize the ROI selection which could be done using morphological features, target markers, or other general labels (like DAPI for nuclei, or WGA or ZO-1 immunostaining for membranes) that can be processed manually or via automated segmentation algorithms in a user-dependent manner. The cell boundary information can be addressed by pre-staining the tissue/cells with a marker specific to the cell type. This is in essence what was done to identify the TH⁺ amacrine cells; the cells were pre-stained with a TH antibody prior to the Light-seq barcoding workflow. This illustrates one of the advantages of Light-Seq, which also means that for single cell precision, having good labeling strategies for marking the boundaries would be important for accurate barcoding.

References

1. Kishi, J. Y. *et al.* SABER amplifies FISH: enhanced multiplexed imaging of RNA and DNA in cells and tissues. *Nat. Methods* **16**, 533–544 (2019).
 2. West, E. R. *et al.* Spatiotemporal patterns of neuronal subtype genesis suggest hierarchical development of retinal diversity. *Cell Rep.* **38**, 110191 (2022).
 3. Yoshimura, Y. & Fujimoto, K. Ultrafast reversible photo-cross-linking reaction: toward in situ DNA manipulation. *Org. Lett.* **10**, 3227–3230 (2008).
 4. Rodrigues, S. G. *et al.* Slide-seq: A scalable technology for measuring genome-wide expression at high spatial resolution. *Science* **363**, 1463–1467 (2019).
 5. Liu, Y. *et al.* High-Spatial-Resolution Multi-Omics Sequencing via Deterministic Barcoding in Tissue. *Cell* **183**, 1665–1681.e18 (2020).
 6. Berry, S., Müller, M. & Pelkmans, L. Nuclear RNA concentration coordinates RNA production with cell size in human cells. *bioRxiv* 2021.05.17.444432 (2021) doi:10.1101/2021.05.17.444432.
 7. Lee, S. *et al.* Covering all your bases: incorporating intron signal from RNA-seq data. *NAR Genom. Bioinform.* **2**, lqaa073 (2020).
 8. Cocquet, J., Chong, A., Zhang, G. & Varticovski, R. A. Reverse transcriptase template switching and false alternative transcripts. *Genomics* **88**, 127–131 (2006).
-
9. Zhu, Y. Y., Machleder, E. M., Chenchik, A., Li, R. & Siebert, P. D. Reverse transcriptase template switching: a SMART approach for full-length cDNA library construction. *Biotechniques* **30**, 892–897 (2001).
 10. Wulf, M. G. *et al.* Non-templated addition and template switching by Moloney murine leukemia virus (MMLV)-based reverse transcriptases co-occur and compete with each other. *J. Biol. Chem.* **294**, 18220–18231 (2019).
 11. Honda, M. *et al.* High-depth spatial transcriptome analysis by photo-isolation chemistry. *Nat. Commun.* **12**, 4416 (2021).
 12. Raj, A., van den Bogaard, P., Rifkin, S. A., van Oudenaarden, A. & Tyagi, S. Imaging individual mRNA molecules using multiple singly labeled probes. *Nat. Methods* **5**, 877–879 (2008).
 13. Lee, J. H. *et al.* Highly multiplexed subcellular RNA sequencing in situ. *Science* **343**, 1360–1363 (2014).
 14. Nuovo, G. J. The foundations of successful RT in situ PCR. *Front. Biosci.* **1**, c4–15 (1996).
 15. Saka, S. K. *et al.* Immuno-SABER enables highly multiplexed and amplified protein imaging in tissues. *Nat. Biotechnol.* **37**, 1080–1090 (2019).
 16. Lin, J.-R. *et al.* Highly multiplexed immunofluorescence imaging of human tissues and tumors using t-Cy3CF and conventional optical microscopes. *Elife* **7**, (2018).

Decision Letter, second revision:

Dear Sinem,

Thank you for submitting your revised manuscript "Light-Seq: Light-directed in situ barcoding of biomolecules in fixed cells and tissues for spatially indexed sequencing" (NMETH-A47917D). It has now been seen by the original referees and their comments are below. The reviewers find that the paper has

improved in revision, and therefore we'll be happy in principle to publish it in Nature Methods, pending minor revisions to comply with our editorial and formatting guidelines.

TRANSPARENT PEER REVIEW

Nature Methods offers a transparent peer review option for new original research manuscripts submitted from 17th February 2021. We encourage increased transparency in peer review by publishing the reviewer comments, author rebuttal letters and editorial decision letters if the authors agree. Such peer review material is made available as a supplementary peer review file. Please state in the cover letter 'I wish to participate in transparent peer review' if you want to opt in, or 'I do not wish to participate in transparent peer review' if you don't. Failure to state your preference will result in delays in accepting your manuscript for publication.

Thank you again for your interest in Nature Methods Please do not hesitate to contact me if you have any questions.

Sincerely,
Rita

Rita Strack, Ph.D.
Senior Editor
Nature Methods

ORCID

Reviewer #1 (Remarks to the Author):

The authors have done an admirable job of revising the manuscript. While there are a few things that I would still like to see as I am keenly interested in how the technology works I won't ask for further modifications. As this is the initial paper, with the revisions, it is certainly complete enough for others to use the technology and to see how it works in their hands. I personally don't like it when a reviewer asks for new experiments that are not germane to the paper being reviewed (as so many do) so I will simply look forward to future papers from this group.

Reviewer #3 (Remarks to the Author):

I am satisfied with the revised version and no further revision is needed.

Author Rebuttal, second revision:**Response to Reviewers**

We thank the reviewers for their careful and thoughtful feedback on our work. To fully address your comments and upon editorial recommendation, we have revised the manuscript into the longer Article format, hence the previous figure numbering and the text layout changed substantially, although the fundamental content and data is not altered. Following the reviewer suggestions, we: (1) included additional experimental data and analyses that have been added to the as new figures and figure panels, (2) shared a preview of our ongoing work that will be incorporated into future publications in the response letter, (3) re-structured the supplementary information to have it more streamlined and comprehensive, (4) prepared detailed supplemental notes for detailing the experimental design considerations and implementation recommendations. We provide a list of these changes below, followed by inline responses to each specific comment. Together we believe these changes address the comments, and we're grateful for your feedback to improve the overall quality of the publication.

Reviewer 1

[C1.0] The authors have described an interesting method for ex situ RNA seq while preserving spatial identity of cells in a fixed biological sample through one-step photocrosslinking of barcoding guide primers that permit barcodes to be added to the in situ transcribed cDNA. Potential advantages of using this method are cost-effectiveness, preservation of tissue context for transcriptomics, dispensable upstream processing of samples like dissociation of cells and sorting, to name a few. The authors use this approach to assess the partial transcriptomes from various brain and retina cellular regions including groups of identified TH+

cells. While the methodology is potentially useful, there are concerns that the authors should address to increase confidence that the technology is working as envisioned:

[R1.0] We thank the reviewer for their kind words and careful review. We address each of the points individually below.

[C1.1]

1. To know the sensitivity of the methodology, it is important to know the efficiency of the barcode attachment reaction. One way to do this would be to take 10 different *in vitro* transcribed RNAs and to mix them in various ratios and abundances and then perform the reaction procedure and assess the resultant ratio of products. This ideally would be performed on RNAs that are immobilized such as on beads, embedded in agar or on nitrocellulose.
2. Likewise, it is important to know the efficiency of the “switch RT” reaction (what % of queried mRNAs are detected). This is particularly difficult but may be optimizable if the original efficiency can be assessed.

[R1.1] We agree with the reviewer that the barcode attachment reaction is one of several important steps that could limit the sensitivity of Light-Seq, including RT efficiency, polyadenylation efficiency, photo-crosslinking for barcode attachment, cDNA extraction, cross-junction synthesis efficiency, and associated background for each step, all of which are impacted by *in situ* conditions. In light of the reviewer's suggestion, we performed quantitative comparison to smFISH data, as the current gold standard for RNA quantification *in situ*, to directly assess the cumulative sensitivity of the method in the *in situ* environment.

We previously developed the SABER-FISH method for amplifying smFISH signals and demonstrated very high sensitivity and specificity of individual RNA transcript detection, including in retina samples specifically¹. We have now added additional quantitative analysis comparing our detection efficiency to smFISH of several targets in previously performed for the same cell types, within the same sample type². Together, we estimate Light-Seq sensitivity to be $4.29 \pm 3.39\%$ (mean \pm std, $n=16$ genes, 4 replicates) and Drop-Seq $3.97 \pm 4.38\%$ ($n=16$ genes, 6 replicates) for these genes. We made several additions to the manuscript to emphasize the sensitivity of Light-Seq. We added sensitivity analysis in the new **Fig. 4e** and discussion of this comparison in the main text (Line 239-247), directly comparing gene detection for Light-Seq and Drop-Seq for a set of RNA markers benchmarked by smFISH. We also added gene-specific sensitivity estimations in **Extended Data Fig. 4h-j** and analysis for reads from genes of different lengths **Extended Data Fig. 5, 7**.

We acknowledge that this is an imperfect comparison because our degenerate RT primers (different from polyT priming used in Drop-Seq) could theoretically yield multiple reads on a single long RNA molecule (see response **[R3.1]** below for detailed description of length bias analysis). To clarify this, we have

included a per-gene analysis for 16 RNAs measured by smFISH in the supplement, to show similar Light-Seq sensitivity compared to Drop-Seq for genes of different lengths (**Extended Data Fig. 5e-j, 7b-c**)

Regarding the photo-crosslinking efficiency itself, previous works have shown that the efficiency of 1 sec photo-crosslinking under a 365 nm wavelength of light is 90-97% *in vitro*³. To achieve a high level of photo-crosslinking *in situ* we empirically optimized the crosslinking efficiency for our imaging system and found that 10% 365 nm LED power output for 10-20s exposure was optimal for our system to get the best inside-ROI to out-of-ROI signal ratio. Of note, these parameters will depend on the optical setup (light source, light path, objective, etc.) and should be optimized for new optical setups. We have added user-friendly instructions for how to optimize these parameters in the new **Supplementary Note 4** and in protocols.io to assist future Light-Seq users in optimizing for their own setups.

[C1.2]

3. With regard to the switch RT reaction, is there sequencing evidence for “jumping” of the RT to other mRNAs at this step? This happens in standard RT reactions and likely would happen more frequently for this reaction. Does this confound the interpretation of results?

[R1.2] This is an excellent question and one that we’ve thought about deeply ourselves. We intentionally made several design choices to reduce the potential for this or related effects to lead to sequenceable reads, which we now outline extensively in a new **Supplementary Note 1**. In this note, we explain why we do not expect to see either of these background reads in our sequencing results and share the additional analysis we performed on our paired end reads to demonstrate that we do not see evidence of chimeric RNA reads in our sequencing results. To our main text, we added a sentence (Line 141-143) pointing to this new **Supplementary Note 1** and added citations to several important papers that have used and/or studied the template-switching behavior of reverse transcriptases, both to designated template switching oligos as well as to other RNA sequences.

[C1.3]

4. (a) There seems to be wide variation in the number of mapped sequenced reads from various samples. For example, on lines 140-143, 1959 (HEK cells) and 1170 (3T3) UMIs were reported per unit area. What does unit area mean? Is this per irradiated unit (multiple cells) or averaged to be per single cell?

[R1.3] We thank the reviewer for this point. Because the barcoding area of Light-Seq can be arbitrarily set by the user, we chose to normalize the number of transcripts that can be captured with Light-Seq as the number of unique molecular identifier (UMI) sequences per “unit area” that was roughly the size of a bead in Slide-Seq⁴ or a barcoded square in DBIT-Seq⁵, which we define as 10 μm x 10 μm . This metric was commonly used for reporting sensitivity and comparing spatial transcriptomics methods. Number of UMIs

reported per unit area depend on the area of irradiation, and since cell shapes, sizes, cytoplasmic volumes, and RNA content are quite heterogeneous across cell types, it is expected that the read counts would not match across different cell types. For cultured cells, the unit area only includes segmented cells (not the surrounding empty area) and for tissues, the unit area only includes the outlined cells or layers and is not averaged per cell. We have now added this information to the in the main text (Lines 159-166) and clarified how the unit area was calculated in the legend of the new **Supplementary Table 1**.

We note, however, that this is not a good general metric for sensitivity, as RNA density is known to vary greatly with different cell types and 3D volumes⁶. Consistent with this, we observe wide variation in UMIs/unit area across cell types and in general, we observe that cells with larger cytoplasmic volumes correlate with higher UMIs/unit area (now shown in **Extended Data Fig. 3d** and highlighted in the corresponding main text – Lines 209-214). In the retina, for example, the GCL has the largest cell volumes and cytoplasmic areas and the highest UMIs/unit area, while the ONL has the smallest cell/cytoplasm volumes and the smallest UMI/unit area. The discrepancy in UMIs per unit area for HEK versus 3T3 cells similarly correlates with differences in cytoplasmic volume per unit area. Although the same number of HEK and 3T3 cells were barcoded, the total barcoded cellular area for 3T3 cells is 2.3-3-fold larger than the HEK cell area (**Supplementary Table 1**). This has now been emphasized in the table legend.

[C1.4]

4. (b) This is said to be subsaturating even though it was from 30 million reads, and that it almost doubles when going to 200 million reads. This suggests that there is a large amount of reads that didn't map to the genome (what percentage of the reads did map?) and that there are few reads/cell that can be detected. More clarification of this would be helpful.

[R1.4] As the reviewer observed, a significant number of reads didn't map uniquely to the genome. We now include an overview of our sequence processing pipeline (**Extended Data Fig. 4a**) and sunburst plots depicting what fraction of reads are filtered out at which steps and why, including detailed methods about the pipeline (**Extended Data Fig. 4b**). As expected from our internal priming design, the majority of our reads are multimapping (e.g. rRNA). Future implementations of the technology could include rRNA depletion, as we now highlight in the Discussion (Lines 328-334), which would increase the proportion of uniquely mapped sequencing reads.

[C1.5]

5. The TH cell data appears to have many more UMIs (Supplementary Table 3, 8,428-to 10,446) than other cells or tissue areas that were examined in the study. Discussion of what accounts for these differences would be helpful in insuring that the technology is robust and working as envisioned.

[R1.5] This is a great observation and one that we have changed our text to better highlight (see also response [R1.3] above). We believe that the increase in UMIs for TH⁺ amacrine cells relative to other retinal cells and/or cells in culture is expected and likely driven by biological differences rather than technical ones. The TH⁺ cells are some of the biggest cells in the retina and have very large cytoplasmic areas. **Figure 5d** shows just how dramatic the cytoplasm difference is between these cells compared to the adjacent amacrine cells. We see similarly high UMIs per unit area in the retinal ganglion cells (RGCs) compared to the other cell types in the retina (**Extended Data Fig. 3 and [R1.3]**). RGCs, like TH⁺ amacrine cells, are large and sprawling in morphology and harbor long axons that reach into the brain, which is consistent with the idea that potentially these larger cells have more RNA than their smaller neighboring cell types. We now also emphasize in the main text (Lines 324-326) that this is a major caveat to interpreting the metric of “UMIs per unit area” for comparison across technologies for spatial transcriptomics, since there is large variability in these numbers across cell types or even across regions measured within a single cell.

[C1.6]

6. The authors only present exon data as their output, yet given the experimental procedure where whole cells are irradiated there must also be nuclear intron containing RNA detected in their sequencing runs. It would be informative to know how much nuclear RNA vs cytoplasmic RNA is present in the sample. Indeed, it might be that the nuclear RNA would be a relatively high fraction of the reads since it is possible that the nuclear RNA is less “fixed” than cytoplasmic RNA. Or alternatively there may be less due to less accessibility of the reagents. The authors should report these numbers and discuss the implications.

[R1.6] This is a great question, and the reviewer is correct that our current primary pipeline only maps to exons based on the annotated genome file for the appropriate genome (see new **Extended Data Fig. 4a-b**). We agree that the efficiency for recovering nuclear vs. cytoplasmic RNA would likely depend on the fixed *in situ* environment and accessibility. In light of this question, we have since performed mapping to full gene bodies and analysis of intronic reads, and have added details about this to our main text (Lines 257-260), supplement (**Supplementary Note 3**), Methods (**Intron analysis** section), and code base on Github. In brief, we do see substantial amounts of intronic reads in our sequencing results (>20% in all replicates) when we include them in the mapping, indicating we are able to successfully extract nuclear RNA at ratios consistent with similar methods using internal RNA priming⁷. As a result of internal priming, we also see good representation of noncoding RNAs (**Extended Data Table 1**). Further optimization for fixation, permeabilization, and extraction could likely further improve the accessibility of nuclear RNA or change the ratios of RNA species recovered (see [R1.7] below).

[C1.7]

7. Have the authors explored methods for removing RNA binding proteins or RNA structure to increase the accessibility of the RNA to the reaction reagents including reverse transcriptase? Such

data may expand the usefulness of the methodology to other types of fixatives or differentially prepared tissue sections especially pathological tissue specimens.

[R1.7] We have so far focused on trying to preserve the *in situ* environment to the greatest extent possible throughout the Light-Seq barcoding and sequence extraction process, but these are great suggestions for future further improvements to the quality of sequencing data, at the potential cost of disrupting protein epitopes for further imaging analysis.

[Redacted]

[C1.8]

8. ROI dimensions and the number of cells in an ROI is estimated by the size of ROI is unclear. Authors should add this number to their respective figure legends.

[R1.8] We thank the reviewer for pointing this out, we made several changes to the figure legends to incorporate this information and also include more detailed information in our **Methods** section for how ROI areas and cell numbers were estimated. For the cell mixing experiments, we added the cell counts (~25 cells of each type) to the **Figure 3** legend (see **Extended Data Fig. 2** for masks), and ROI sizes are included in **Supplementary Table 1**. For the mouse retina experiments we have now added cell number estimations in the main text and methods, **Supplementary Table 3**, in addition to the **Figure 4** legend.

[C1.9]

9. While the barcoding strategy is light directed and can be specific, authors should show/comment/discuss about the resolving power of light-seq barcoding to distinguish between adjacent cells.

[R1.9] Light-Seq uses 365 nm - 405 nm light for covalent barcode crosslinking, providing the basis for theoretically diffraction-limited barcoding. This could enable even subcellular barcoding, as demonstrated by the subcellular barcoding using laser-scanning in **Extended Data Fig. 1**. However, other practical considerations may affect the final resolving power. We have added a new panel to **Extended Data Fig. 1** showing the minimum feature size we can achieve using DMD illumination and the signal decay. We also included a line scan showing fluorescence signal across and outside the ROI boundary for the rare amacrine cell type experiment (**Extended Data Fig. 6**). We now emphasized in the main text (Lines 179-189 and 300-305) how light-scattering and other sources of background can be introduced and mitigated. We further added a new section (**Supplementary Note 4**) with instructions for calibrating and optimizing the illumination, precisely focusing the illumination in XYZ, optimizing the illumination, measuring the sensitivity, drawing ROIs, and designing experiments to take into account light scattering. We also updated our protocols.io with these instructions.

[C1.10]

10. Spatial transcriptomics is advancing fast with newer assays and hence a close comparison of this method with other spatial methods (10X Visium, APEX-seq, slide-seq) becomes important. The Discussion section should be expanded to present a more detailed comparison of the data (not cost analysis) generated with this method as compared with other currently used spatial transcriptomic assays.

[R1.10] We appreciate this feedback. We have rewritten our Discussion to better compare the data to existing spatial transcriptomic assays and clarify key differences in workflow, flexibility, data types and sensitivity. We chose a conservative approach for our sequence analysis pipeline by filtering out reads that multimap to the genome or to the transcriptome, and only considering exonic maps. This means that the true number of unique molecules we are actually labeling is much higher (see **Extended Data Figure 4a** for breakdown of sequence filtering). We compare these conservative estimates to smFISH data for 16 different markers (see **[R1.1]** above) to estimate sensitivity with comparison to a common standard. The Discussion section now contains a comparison of these sensitivity measurements to other spatial methods (Lines 315-326, 356-362, 387-392). We refrained from making a more direct/side-by-side comparison to alternative methods, because the quantitative metrics for some of the other methods are either not available or not analyzed in a transparent manner for us to evaluate direct comparisons. These could also vary significantly from tissue type to tissue type.

The low cost and convenience of the method is a critical advantage that makes Light-Seq accessible to scientists around the world, and we updated the cost estimates to better reflect non-promotional pricing (see also **[R1.12]** below) and added a caution note on potential price variations.

[C1.11]

11. Referencing in the paper was haphazard. There are many background papers that should be referenced as the particular subtechniques were previously described. For example, the in situ cDNA synthesis step should reference Tecott et al. "In Situ Transcription: Specific synthesis of cDNA in fixed tissue sections." Science 1988. There are many other examples as well and the authors are encouraged to be more appropriate and thorough in their referencing.

[R1.11] We thank the reviewer for pointing out this paper and have included it in our references. We also added additional references related to the cDNA synthesis, non-templated, and templated switching behavior of MMLV-type reverse transcriptases⁸⁻¹⁰ (see **[R1.3]** above), and for other methods like PIC¹¹ and smFISH¹²), and RT optimizations^{13,14}. We welcome further suggestions for additional references that should be included.

[C1.12] There is a wealth of useful supplemental data but it is not well integrated into the body of the manuscript. While it is referenced, much of the supplemental data is not discussed. Also, if the data isn't directly relevant to the manuscript then it should be removed, e.g. the cost analysis table is superfluous and not representative of actual costs since most investigators get institutional or quantity discounts making the price comparisons unrealistic.

[R1.12] Thank you for this feedback. We have completely reworked the main text and switched to the Nature Methods Article format, allowing us to include more text, main figures and comprehensive references to our supplementary material. We also added a wealth of new data and analyses to the supplementary materials and re-formatted them to be more streamlined (in the form of Extended Data and Supplemental Information as the journal recommends).

The relatively low cost of the method is a huge part of making Light-Seq accessible to many academic labs across the world and this is important data for us to have and share with prospective users of the method. However, we agree that academic discounts and old price estimates before recent inflation could have unfairly lowered the cost estimate, so we updated the cost estimate table with non-discounted publicly available list prices for all the reagents this was available for. After incorporating these updated commercial list prices, which combined account for the majority (~80%) of the total cost, our estimate went from \$30 to \$34.50 (potential pricing variations and cautions are now highlighted in **Supplementary Table 5**).

Reviewer 2

[C2.0] In this manuscript, Kishi et al. describe Light-Seq, a novel technique for in situ barcoding of cDNA followed by next-generation sequencing that allows for the characterization of transcriptomes from a defined group of cells or a single rare cell type. They provide excellent examples for the specificity and sensitivity of Light-Seq, sequencing transcriptomes from both cells in culture and fixed tissue sections. The ability of Light-Seq to resolve the transcriptome of a very rare cell type in the mouse retina is impressive. This kind of information is often not available from single-cell RNAseq experiments. The relative simplicity and affordability of the Light-Seq method are also impressive.

This manuscript is well written, the data presented is of high quality, and the experiments are well controlled. The authors provide more than sufficient evidence to support the applicability and the unique properties of their method.

I add only one cautionary note. The authors cite several papers posted on BioRxiv, which have not yet been peer-reviewed. I would suggest refraining from citing non-peer-reviewed papers.

[R2.0] We thank the reviewer for their kind words and careful review.

We have updated the citations for preprints that have now been published (original references 9, 14, 15, 27, 38) and we are hopeful that the few remaining preprint references (which we believe are important to cite considering the current dynamism of the field) will be published by the time this manuscript is finalized.

Reviewer 3

[C3.0] The Light_Seq technology is quite interesting and can be useful to the applications to study rare cell populations as demonstrated by the manuscript. Due to the limited throughput, it may not be able to be applied to study a large number of ROIs with high spatial resolution. In general, I am optimistic about the potential and future technology improvements that can be made. I do have several concerns regarding the quality of the current experimental design. More pieces of evidence are needed to address these concerns.

[R3.0] We thank the reviewer for their enthusiasm and careful review. Please see our answers to the specific concerns below.

[C3.1]

1. RNase H based releasing of cDNAs may cause Gene length and detection bias. As shorter cDNAs (shorter genes) may be released much faster and efficiently compared with long cDNAs. Gene length bias analysis is needed here to understand the releasing gene profile. Besides, RNase H release may not be equally efficient in different tissue types, thus, more experiments with broader tissue types are needed here to demonstrate the advantage of this releasing process. It is also necessary to compare cDNA product quality directly with the enzymatic (Proteinase K) digestion-based method.

[R3.1] These are very good points. We prefer the RNase H strategy as it offers high specificity in releasing cDNAs under mild conditions, but we agree that both the in situ RT with internal priming and RNase H release could create gene length-dependent biases. To assess this better, we have performed additional analysis on gene length bias as well as gene body coverage. We also additionally performed analysis on intronic reads to show we are able to successfully extract nuclear sequences, see response [R1.6] above. These analyses are described in new supplementary material (**Extended Data Figures 5 and 7, and Supplementary Note 3**) and are referenced in the main text. The Methods section and code have also been updated to describe the analyses. In brief, we see good gene body coverage without significant 5' or 3' bias, as well consistent RPKM values across transcripts of varying length, as expected from the internal reverse transcription (RT) priming strategy that we use.

Regarding the Proteinase K treatment, in the current work we prioritized developing a workflow that preserved the integrity of proteins and morphology of the tissue but we agree that future modifications to the method such as alternative fixatives and protease digestion could help improve sequence accessibility and extraction for different tissue preparations, at the potential cost of disrupting morphology and protein epitopes for downstream analysis. We also have several additional tissue types that we have successfully performed Light-Seq on with further optimizations along this line, but those belong to separate follow-up publications that will be focused on the biological findings. While the current unmodified protocol has

worked for several other tissues in our collaborator's hands, some applications such as FFPE preparations may require modifications. We have observed that optimal fixation and tissue preparation parameters do vary by tissue type, but that standard tissue-specific protocols that are compatible with RNA detection (for example FISH) tend to also work well for Light-Seq. We have added a note to protocols.io with suggestions for applying Light-Seq to different tissue types. For some applications where preservation of proteins is not high priority, other harsher methods of extraction could in theory be applied. We have also added a note about this in protocols.io for future users.

Please refer to response [R1.7] above for additional information and a preview of other tissue data. We also added a sentence to the Discussion (Lines 328-334) about the potential application of proteases, antigen retrieval, and additional optimizations.

[C3.2]

2. The manuscript did not provide enough validation for the possible outside ROI background labeling due to light scattering. In SI figure 2, there are strong Cy3 background signals remaining (outside ROI) after removing the non-crosslinked strands. Although this could be caused by insufficient washing, it may also be caused by light scattering-induced crosslinking. It also raised another question, what is exactly the highest spatial resolution (with enough Signal to noise ratio) Light-seq can reach?

[R3.2] As the reviewer correctly notes, the resolution of Light-Seq barcoding is subject to background by light-scattering *in situ*, which does blur the ROI boundaries by ~1-3 μm with the DMD optical setup (see new **Extended Data Figures 1 and 6**, where we included linescans showing the fluorescence signal across and outside the ROI boundary and gives resolution estimates for subcellular labeling and for the rare amacrine cell type experiment). This light-scattering can be accounted for during ROI selection by eroding the ROI boundaries by a fixed distance, as demonstrated in **Extended Data Fig. 6** for the barcoding experiment described in **Fig. 5**. Please also see [R1.9] above for detailed descriptions of additions to the text, supplement, and protocols to address the light scattering and resolution question, as well as other potential sources of background.

[C3.3]

3. It is necessary to demonstrate a higher number (>10) barcoding rounds experiment. The current manuscript only demonstrated 3 barcoding rounds which are not enough to show the robustness and consistency of this technique. 10 or 20 rounds of barcoding experiments are needed here to demonstrate the robustness of the whole experimental design, since repeated washing and light exposure may cause unpredictable tissue deformation at some stage.

[R3.3] In this work we sought to develop a simple and accessible method for labs to profile a few different populations of cells that might otherwise be extremely difficult to achieve with existing methods. Three rounds of barcoding can comfortably fit within one working day with a single person preparing buffers and performing all (manual) pipetting operations. We also note in our online protocols.io at which steps the protocol can be paused, so that no protocols need to exceed a standard 8 hour day / 5 days a week schedule. While more barcoding rounds would be achievable manually, we strongly believe that a fully automated fluidics solution would be important for scaling up, which would in turn make the method less accessible and more tedious to implement for the general biology lab. Scaling up the barcoding is of great interest to us for several future applications, but scaling will be the focus of future work.

Our cited previous publications, SABER-FISH¹ and Immuno-SABER¹⁵, which use very similar wash conditions were successfully applied for multiple (up to 7) rounds of iterative formamide-based washing, hybridization and imaging. Other published methods (e.g. CycIF¹⁶) perform much longer (up to 60) sequential workflows with repeated labeling, imaging, and photobleaching, and provide strong preceding evidence to expect good tissue preservation across more barcoding rounds. We now mention this also in the Discussion (Lines 389-395).

Minor comments:

[C3.4]

4. The photomasks and alignment to ROI are vitally important to the current experimental design, but there are not enough details in the experimental section. It may also be hard to address one entire single cell in a very dense tissue section when cell boundary information is missing or hard to collect.

[R3.4] Thank you for this feedback. We have included a new section (**Supplementary Note 4**) and our protocols.io now contains better instructions for aligning and calibrating the DMD field of view, focusing light in 3D on desired cells, and accommodating for light scattering by drawing ROIs slightly smaller than the desired sequencing boundaries (see also responses **[R1.9]** and **[R3.2]** above). This section also includes screenshots of the microscope software we use to calibrate the illumination and draw ROIs onto our samples (a couple examples shown below). We use a commercial DMD attachment on our microscope which comes with software for calibrating the illumination and drawing ROIs (see screenshots below) directly on an image of the sample while it sits on the microscope. When calibration is done properly, the illumination is automatically aligned with the ROI. Other commercial microscopes have similar ROI selection and illumination tools and should not require additional alignment, as long as the sample remains on the microscope for pre-barcode imaging (e.g. of the antibody stain or bright field capture to identify cells of interest) and barcoding.

Calibration software on Nikon microscope.

Screenshot from ROI selection for cell mixing experiment (GFP overlaid on Brightfield).

Light-Seq offers the opportunity to customize the ROI selection which could be done using morphological features, target markers, or other general labels (like DAPI for nuclei, or WGA or ZO-1 immunostaining for membranes) that can be processed manually or via automated segmentation algorithms in a user-dependent manner. The cell boundary information can be addressed by pre-staining the tissue/cells with a marker specific to the cell type. This is in essence what was done to identify the TH⁺ amacrine cells; the cells were prestained with a TH antibody prior to the Light-seq barcoding workflow. This illustrates one of the advantages of Light-Seq, which also means that for single cell precision, having good labeling strategies for marking the boundaries would be important for accurate barcoding.

References

1. Kishi, J. Y. *et al.* SABER amplifies FISH: enhanced multiplexed imaging of RNA and DNA in cells and tissues. *Nat. Methods* **16**, 533–544 (2019).
2. West, E. R. *et al.* Spatiotemporal patterns of neuronal subtype genesis suggest hierarchical development of retinal diversity. *Cell Rep.* **38**, 110191 (2022).
3. Yoshimura, Y. & Fujimoto, K. Ultrafast reversible photo-cross-linking reaction: toward in situ DNA manipulation. *Org. Lett.* **10**, 3227–3230 (2008).
4. Rodrigues, S. G. *et al.* Slide-seq: A scalable technology for measuring genome-wide expression at high spatial resolution. *Science* **363**, 1463–1467 (2019).
5. Liu, Y. *et al.* High-Spatial-Resolution Multi-Omics Sequencing via Deterministic Barcoding in Tissue. *Cell* **183**, 1665–1681.e18 (2020).
6. Berry, S., Müller, M. & Pelkmans, L. Nuclear RNA concentration coordinates RNA production with cell size in human cells. *bioRxiv* 2021.05.17.444432 (2021) doi:10.1101/2021.05.17.444432.
7. Lee, S. *et al.* Covering all your bases: incorporating intron signal from RNA-seq data. *NAR Genom Bioinform* **2**, lqaa073 (2020).
8. Cocquet, J., Chong, A., Zhang, G. & Veitia, R. A. Reverse transcriptase template switching and false alternative transcripts. *Genomics* **88**, 127–131 (2006).
9. Zhu, Y. Y., Machleder, E. M., Chenchik, A., Li, R. & Siebert, P. D. Reverse transcriptase template switching: a SMART approach for full-length cDNA library construction. *Biotechniques* **30**, 892–897 (2001).
10. Wulf, M. G. *et al.* Non-templated addition and template switching by Moloney murine leukemia virus (MMLV)-based reverse transcriptases co-occur and compete with each other. *J. Biol. Chem.* **294**, 18220–18231 (2019).
11. Honda, M. *et al.* High-depth spatial transcriptome analysis by photo-isolation chemistry. *Nat. Commun.* **12**, 4416 (2021).
12. Raj, A., van den Bogaard, P., Rifkin, S. A., van Oudenaarden, A. & Tyagi, S. Imaging individual mRNA molecules using multiple singly labeled probes. *Nat. Methods* **5**, 877–879 (2008).
13. Lee, J. H. *et al.* Highly multiplexed subcellular RNA sequencing in situ. *Science* **343**, 1360–1363 (2014).
14. Nuovo, G. J. The foundations of successful RT in situ PCR. *Front. Biosci.* **1**, c4-15 (1996).
15. Saka, S. K. *et al.* Immuno-SABER enables highly multiplexed and amplified protein imaging in tissues. *Nat. Biotechnol.* **37**, 1080–1090 (2019).
16. Lin, J.-R. *et al.* Highly multiplexed immunofluorescence imaging of human tissues and tumors using t-CyCIF and conventional optical microscopes. *Elife* **7**, (2018).

Final Decision Letter:

Dear Sinem,

I am pleased to inform you that your Article, "Light-Seq: Light-directed in situ barcoding of biomolecules in fixed cells and tissues for spatially indexed sequencing", has now been accepted for publication in Nature Methods. Your paper is tentatively scheduled for publication in our October print issue, and will be published online prior to that. The received and accepted dates will be Feb 19, 2022 and August 10, 2022. This note is intended to let you know what to expect from us over the next month or so, and to let you know where to address any further questions.

Your paper will now be copyedited to ensure that it conforms to Nature Methods style. Once proofs are generated, they will be sent to you electronically and you will be asked to send a corrected version within 24 hours. It is extremely important that you let us know now whether you will be difficult to contact over the next month. If this is the case, we ask that you send us the contact information (email, phone and fax) of someone who will be able to check the proofs and deal with any last-minute problems.

If, when you receive your proof, you cannot meet the deadline, please inform us at rjsproduction@springernature.com immediately.

Once your manuscript is typeset and you have completed the appropriate grant of rights, you will receive a link to your electronic proof via email with a request to make any corrections within 48 hours. If, when you receive your proof, you cannot meet this deadline, please inform us at rjsproduction@springernature.com immediately.

Once your paper has been scheduled for online publication, the Nature press office will be in touch to confirm the details.

Content is published online weekly on Mondays and Thursdays, and the embargo is set at 16:00 London time (GMT)/11:00 am US Eastern time (EST) on the day of publication. If you need to know the exact publication date or when the news embargo will be lifted, please contact our press office after you have submitted your proof corrections. Now is the time to inform your Public Relations or Press Office about your paper, as they might be interested in promoting its publication. This will allow them time to

prepare an accurate and satisfactory press release. Include your manuscript tracking number NMETH-A47917E and the name of the journal, which they will need when they contact our office.

About one week before your paper is published online, we shall be distributing a press release to news organizations worldwide, which may include details of your work. We are happy for your institution or funding agency to prepare its own press release, but it must mention the embargo date and Nature Methods. Our Press Office will contact you closer to the time of publication, but if you or your Press Office have any inquiries in the meantime, please contact press@nature.com.

If you are active on Twitter, please e-mail me your and your coauthors' Twitter handles so that we may tag you when the paper is published.

Please note that Nature Methods is a Transformative Journal (TJ). Authors may publish their research with us through the traditional subscription access route or make their paper immediately open access through payment of an article-processing charge (APC). Authors will not be required to make a final decision about access to their article until it has been accepted. Find out more about Transformative Journals

Authors may need to take specific actions to achieve compliance with funder and institutional open access mandates. If your research is supported by a funder that requires immediate open access (e.g. according to Plan S principles) then you should select the gold OA route, and we will direct you to the compliant route where possible. For authors selecting the subscription publication route, the journal's standard licensing terms will need to be accepted, including self-archiving policies. Those licensing terms will supersede any other terms that the author or any third party may assert apply to any version of the manuscript.

To assist our authors in disseminating their research to the broader community, our SharedIt initiative provides you with a unique shareable link that will allow anyone (with or without a subscription) to read the published article. Recipients of the link with a subscription will also be able to download and print the PDF. As soon as your article is published, you will receive an automated email with your shareable link.

Please note that you and your coauthors may order reprints and single copies of the issue containing your article through Nature Portfolio's reprint website, which is located at

<http://www.nature.com/reprints/author-reprints.html>. If there are any questions about reprints please send an email to author-reprints@nature.com and someone will assist you.

Best regards,
Rita